_Article_

# mTORC1 senses glutamine and other amino acids through GCN2

Gianluca Figlia [ID] [1,2,3,5][✉], Sandra Müller[1,2,3], Fabiola Garcia-Cortizo [ID] [1,2,3], Marilena Neff[1,2,3], Glynis Klinke[4], Gernot Poschet [ID] [4] & Aurelio A Teleman [ID] [1,2,3][✉]

## Abstract

mTORC1 promotes cell growth when nutrients such as amino acids are available. While dedicated sensors relaying availability of leucine, arginine and methionine to mTORC1 have been identified, it is still unclear how mTORC1 senses glutamine, one of its most potent inducers. Here, we find that glutamine is entirely sensed through the protein kinase GCN2, whose initial activation is not triggered by depletion of glutamine itself, but by the concomitant depletion of asparagine. In turn, GCN2 leads to a succession of events that additively inhibit mTORC1: within 1 h, GCN2 inhibits mTORC1 through the Rag GTPases, independently of its function as an eIF2α kinase. Later, GCN2-mediated induction of ATF4 upregulates Ddit4 followed by Sestrin2, which together cause additional mTORC1 inhibition. Additionally, we find that depletion of virtually any other amino acid also inhibits mTORC1 through GCN2. GCN2 and the dedicated amino acid sensors thus represent two independent systems that enable mTORC1 to perceive a wide spectrum of amino acids.

**Keywords** Glutamine; Asparagine; mTORC1; GCN2; Amino Acid Sensors
**Subject Category** Metabolism

## Introduction

Cells are able to sense nutrient levels and to accordingly adjust their growth to prevent potentially disastrous imbalances between nutrient usage and availability. In eukaryotes, this is critically mediated by the mechanistic Target Of Rapamycin Complex 1 (mTORC1), a protein kinase complex that senses and integrates inputs from multiple nutrients as well as from growth factors (Battaglioni et al, 2022; Goul et al, 2023; Kim and Guan, 2019; Liu and Sabatini, 2020). In turn, mTORC1 regulates many important anabolic pathways to promote protein, lipid and nucleotide synthesis, while inhibiting catabolic pathways such as autophagy. Consistent with this central coordinating role, dysregulation of mTORC1 is involved in a broad range of diseases, including cancer,

diabetes and obesity (Liu and Sabatini, 2020), and it is a major driver of ageing (Mannick and Lamming, 2023). Knowledge of the mechanisms enabling mTORC1 to sense and respond to nutrients and other inputs is thus pivotal for understanding what causes its dysregulation in disease and for targeting it therapeutically.

Activation of mTORC1 requires its simultaneous interaction with two small GTPases on the surface of lysosomes, Rheb and the Rag GTPases. On the one hand, interaction of mTORC1 with Rheb allosterically promotes its kinase activity (Yang et al, 2017). On the other hand, binding to the Rag GTPases is necessary to recruit mTORC1 to lysosomes to allow interaction with Rheb (Kim et al, 2008; Sancak et al, 2008). Since Rheb is activated by growth factors (Inoki et al, 2002; Manning et al, 2002; Potter et al, 2002) and the Rag GTPases by nutrients (Kim et al, 2008; Sancak et al, 2008), this mechanism causes maximal activation of mTORC1 only when both mitogenic and nutrient inputs are simultaneously available. In contrast to monomeric Rheb, the Rag GTPases form obligate heterodimers of RagA or RagB bound to RagC or RagD (Kim et al, 2008; Sancak et al, 2008). Particularly critical for the interaction with mTORC1 is the nucleotide loading state of the A/B subunits (Anandapadamanaban et al, 2019; Napolitano et al, 2020), which is regulated by the GATOR (GTPase Activity TOward Rags) complex (Bar-Peled et al, 2013; Valenstein et al, 2024). This is a large protein complex consisting of the subcomplex GATOR1, which acts as a GAP for RagA/B, the subcomplex GATOR2, which inhibits the activity of GATOR1 through a still largely unclear mechanism, and the subcomplex KICSTOR, which probably stabilizes the interaction between GATOR1 and GATOR2 (Peng et al, 2017; Valenstein et al, 2024; Wolfson et al, 2017). Differential sensitivity of the RagA/B isoforms to GATOR1 combined with tissue-specific patterns of RagA/B expression allows organisms to fine-tune the regulation of mTORC1 by nutrient inputs in different tissues (Figlia et al, 2022).

According to the current model, amino acid availability feeds into this large molecular machinery through dedicated sensors (Wolfson and Sabatini, 2017), proteins that bind to one specific amino acid and in turn regulate the function of GATOR2 (Chantranupong et al, 2016; Liu et al, 2024; Wolfson et al, 2016) or the interaction between GATOR1-KICSTOR and GATOR2 (Gu et al, 2017; Tang et al, 2022). Among amino acids, leucine, arginine and glutamine are considered the most important ones for mTORC1 (Duran et al, 2012; Hara et al, 1998; Jewell et al, 2015; Nicklin et al, 2009). Leucine is sensed through the sensors Sestrin1

[1]Signal Transduction in Cancer and Metabolism, German Cancer Research Center (DKFZ), Heidelberg 69120, Germany. [2]Faculty of Medicine, Heidelberg University, Heidelberg 69120, Germany. [3]Faculty of Biosciences, Heidelberg University, Heidelberg 69120, Germany. [4]Center for Organismal Studies, Heidelberg University, Heidelberg 69120, Germany. [5]Present address: Institute of Neuropathology, RWTH Aachen University Hospital, Aachen, Germany. [✉]E-mail: gfiglia@ukaachen.de; a.teleman@dkfz.de

and Sestrin2 (and to a lesser extent Sestrin3), which bind and inhibit GATOR2 when leucine levels are low (Wolfson et al, 2016). Two distinct mechanisms are instead responsible for sensing arginine. While SLC38A9 senses the lysosomal pool of arginine and directly interacts with the Rag GTPases (Fromm et al, 2020; Jung et al, 2015; Rebsamen et al, 2015; Wang et al, 2015), CASTOR1 senses cytosolic arginine either as a homodimer or as a heterodimer with CASTOR2 and it binds and regulates GATOR2 through a mechanism analogous to Sestrin1/2/3 (Chantranupong et al, 2016). A dedicated sensor for methionine has also been identified. In contrast to Sestrin1/2/3 and CASTOR1/2, the methionine sensor SAMTOR does not bind methionine directly but rather its downstream metabolite S-adenosylmethionine (SAM) and interacts with GATOR1 rather than GATOR2 to promote its GAP activity (Gu et al, 2017).

Although glutamine is one of the most important amino acids for mTORC1, how glutamine is sensed, and whether the mechanism involves a dedicated sensor analogous to those identified for leucine, arginine and methionine, is still unclear. Besides its role in protein synthesis, glutamine serves as an amino-group donor for asparagine, nucleotide and amino sugar biosynthesis (Zhang et al, 2017). Glutamine can also be hydrolyzed to glutamate for energy production through the TCA cycle, a pathway known as glutaminolysis (Fig. 1A). Whether mTORC1 senses glutamine directly or one of these many downstream metabolites is also unclear. While initial evidence pointed to a role for glutaminolysis in glutamine sensing through a mechanism dependent on the Rag GTPases (Duran et al, 2012), subsequent studies concluded that glutamine is sensed independently of the Rag GTPases (Jewell et al, 2015; Meng et al, 2020), thereby leaving open questions regarding the mechanism of glutamine sensing and the role of glutamine metabolism in this process.

By systematically testing the role of downstream metabolites of glutamine, of the GATOR-Rag machinery, and of other signaling pathways converging on mTORC1, we provide here new insights into the mechanism of glutamine sensing. We find that upon brief removal of glutamine, mTORC1 senses the drop in the downstream metabolite asparagine rather than in glutamine itself. Depletion of glutamine and asparagine, however, is not sensed through a dedicated sensor as for leucine, arginine and methionine, but rather through the stress-activated kinase GCN2, which broadly responds to accumulation of uncharged tRNAs (Masson, 2019) and/or ribosome collisions (Ishimura et al, 2016; Wu et al, 2020) when amino acids are scarce. Upon activation, GCN2 initiates multiple sequential mechanisms to inhibit mTORC1, both Rag-dependent and Rag-independent, thus reconciling the conflicting results on the role of the Rag GTPases in glutamine sensing (Duran et al, 2012; Jewell et al, 2015). Interestingly, these mechanisms rely only in part on the canonical function of GCN2 as an eIF2α kinase and on downstream induction of the integrated stress response (ISR) (Costa-Mattioli and Walter, 2020), pointing to a novel function of GCN2 in mTORC1 regulation independent of the ISR. We further provide evidence that GCN2 represents a general amino acid sensing system to couple mTORC1 activity not only to the availability of glutamine, but also to the availability of virtually any amino acid, including those also sensed through dedicated sensors. When leucine is re-supplied to cells, mTORC1 inhibition by GCN2 is released more slowly than mTORC1 inhibition through the dedicated sensors Sestrin1/2/3, providing a potential explanation for the co-existence of these two independent amino acid sensing systems.

# Results

## Asparagine is the main limiting metabolite sensed by mTORC1 upon acute glutamine depletion

Glutamine serves as an amino-group donor for the synthesis of asparagine, glucosamine-6-phosphate (GlcN6P), purines, pyrimidines and nicotinamide adenine dinucleotide (NAD) (Fig. 1A) (Zhang et al, 2017), generating glutamate as a byproduct. Glutamate is also generated from glutamine after hydrolysis of its gamma amino group to free ammonium in the first reaction of glutaminolysis (Zhang et al, 2017). In addition, intracellular glutamine can be exchanged for extracellular leucine through the SLC7A5/SLC3A2 antiporter to promote leucine uptake (Nicklin et al, 2009). As a first step towards understanding how mTORC1 senses glutamine, we sought to determine whether mTORC1 senses glutamine itself or one of its many downstream metabolites. To this end, we tested if replenishing any of these metabolites would be sufficient to rescue mTORC1 activity upon glutamine removal. To replenish cell-permeable metabolites (shown in orange, Fig. 1A), we added these directly into the culture medium. To replenish non-cell-permeable metabolites, we provided cell-permeable precursors (shown in blue, Fig. 1A) that are converted through alternative metabolic pathways into the metabolites of interest. To test the role of glutamine-dependent leucine uptake, we provided an ester derivative of leucine that freely diffuses through the plasma membrane and is subsequently hydrolyzed to leucine (methyl-leucine) (Reeves, 1979). We could not however assess the role of phosphoribosylamine and phosphoribosylformylglycinamidine (FGAM), two metabolites in the de novo purine biosynthesis pathway, since active or facilitated transport through the plasma membrane has not been reported, their negatively charged phospho-group predicts low diffusion across the membrane and no other known metabolic pathway can replenish these metabolites. We confirmed that supplementation of the culture medium with each metabolite caused a corresponding increase in its intracellular level (Fig. EV1A–H). In order to minimize the contribution of indirect metabolic effects, we focused our analysis on cells acutely deprived of glutamine and supplemented with each metabolite for 1 h. Consistent with the importance of glutamine for mTORC1, acute depletion for 1 h was sufficient to cause marked mTORC1 inhibition in HEK293T cells, indicated by reduced phosphorylation of its direct substrate S6K1 (lane 2, Fig. 1B,C). Strikingly, among the metabolites tested, supplementation with asparagine ("N") completely prevented this effect (lane 3, Fig. 1B,C), suggesting that asparagine rather than glutamine itself is the limiting metabolite sensed by mTORC1 in this timeframe. Supplementation with asparagine also prevented mTORC1 inhibition upon glutamine removal in three other commonly used cell lines of different origins (HeLa, U2OS and HepG2) (Fig. EV2A–C), thus ruling out cell-line-specific effects. Consistent with glutamine being sensed through its metabolism to asparagine, treatment with 6-diazo-5-oxo-L-norleucine (DON), an inhibitor of all glutamine-utilizing enzymes, reduced mTORC1 activity in glutamine-replete conditions, and this effect could be rescued by supplementation with asparagine but not glutamine (Fig. EV2D). Titration of asparagine in glutamine-starved cells showed that as little as 25 µM asparagine was sufficient to maintain high mTORC1 activity (Fig. 1D,E), corresponding to

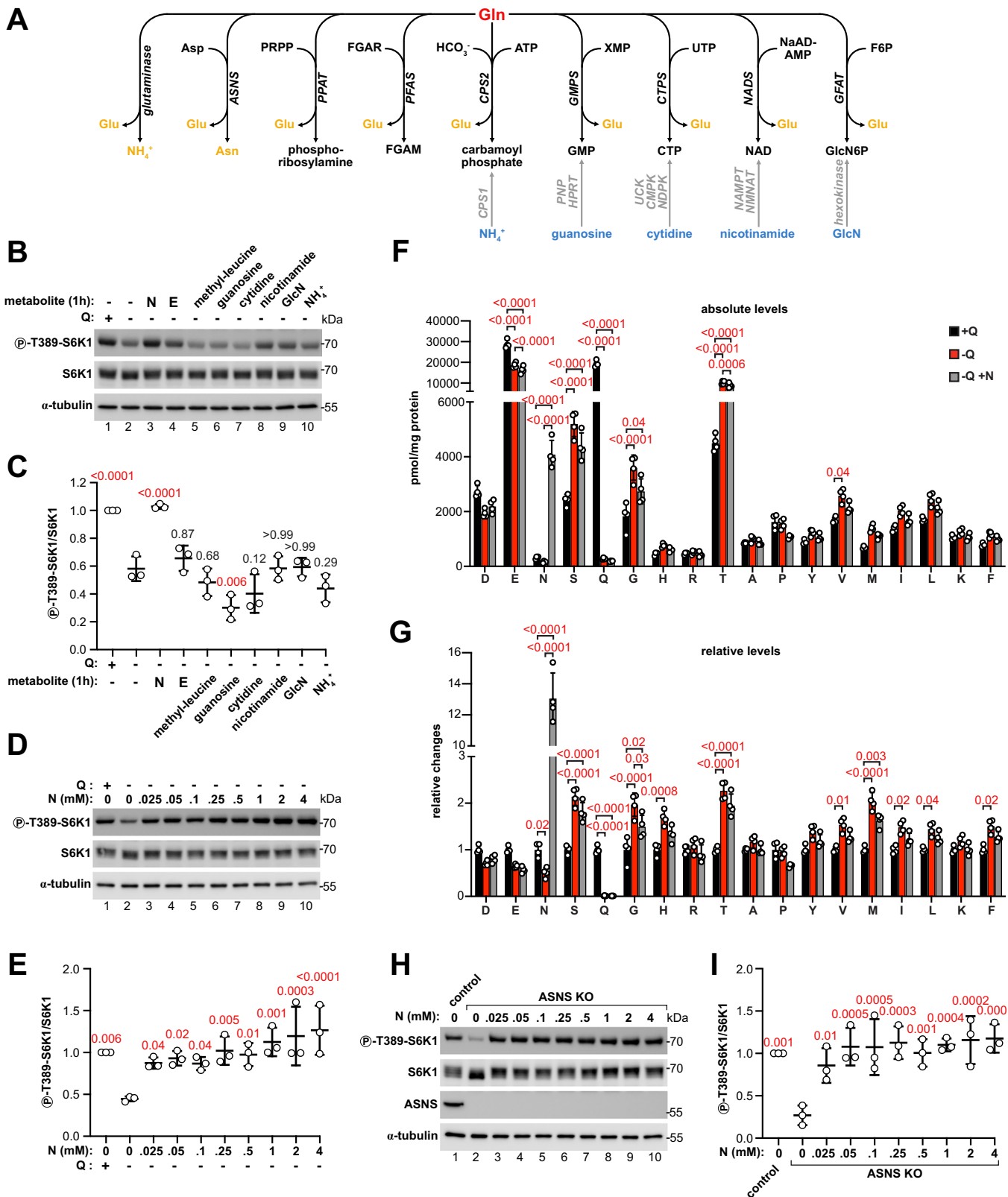

**Figure 1.  Asparagine is the limiting downstream metabolite sensed by mTORC1 upon acute glutamine depletion.**

(A) Graphical summary of the metabolic reactions that use glutamine (Gln) as a substrate in mammalian cells. Additional substrates, the main reaction products and the enzymes catalyzing these reactions are indicated. Metabolites in orange are cell-permeable. Metabolites in blue can serve as cell-permeable precursors to replenish non-cell-permeable downstream metabolites of glutamine through the metabolic reactions shown in gray. Glu glutamate, Asp asparatate, Asn asparagine, PRPP phosphoribosyl pyrophosphate, FGAR phosphoribosyl-N-formylglycinamide, FGAM phosphoribosylformylglycinamidine, XMP xanthosine monophosphate, GMP guanosine monophosphate, UTP uridine triphosphate, CTP cytidine triphosphate, deamido-NAD deamido-nicotinamide adenine dinucleotide, NAD nicotinamide adenine dinucleotide, F6P fructose-6-phosphate, GlcN6P glucosamine-6-phosphate, GlcN glucosamine. (B, C) Supplementation of HEK293T cells with asparagine rescues the drop in mTORC1 activity caused by glutamine removal. Immunoblot from cells incubated for 1 h in regular DMEM, glutamine-free DMEM or glutamine-free DMEM supplemented with the indicated metabolites at 4 mM concentration. Representative example (B) of three biological replicates quantified in (C). The same samples were loaded on different gels and blotted in parallel with the indicated antibodies. Protein concentration was controlled by blotting α-tubulin on a separate gel. Glutamine-starved cells without any metabolite supplementation are set to 1. Q: glutamine, N: asparagine, E: glutamate, GlcN: glucosamine, line: average, error bars: standard deviation. One-way ANOVA and Dunnett's post hoc test. Exact $P$ values relative to lane 2 are indicated in the figure and highlighted in red when significant (< 0.05). (D, E) In all, 25 μM asparagine is sufficient to rescue mTORC1 activity upon glutamine removal. Immunoblot from HEK293T cells incubated for 1 h in regular DMEM, glutamine-free DMEM or glutamine-free DMEM containing the indicated concentrations of asparagine. Representative example (D) of three biological replicates quantified in (E). The same samples were loaded on different gels and blotted in parallel with the indicated antibodies. Protein concentration was controlled by blotting α-tubulin on a separate gel. Glutamine-starved cells without asparagine supplementation are set to 1. Q: glutamine, N: asparagine, line: average, error bars: standard deviation. One-way ANOVA and Dunnett's post hoc test. Exact $P$ values relative to lane 2 are indicated in the figure and highlighted in red when significant (<0.05). (F, G) Re-supplementation of asparagine does not rescue glutamine levels nor those of other amino acids depleted upon glutamine removal. Measurement of amino acid levels in HEK293T cells incubated for 1 h in regular DMEM, glutamine-free DMEM or glutamine-free DMEM supplemented with 250 μM asparagine. Data are expressed as either pmol amino acid per mg protein (F) or relative changes after setting each amino acid concentration in glutamine-rich conditions to 1 (G). D: aspartate, E: glutamate, N: asparagine, S: serine, Q: glutamine, G: glycine, H: histidine, R: arginine, T: threonine, A: alanine, P: proline, Y: tyrosine, V: valine, M: methionine, I: isoleucine, L: leucine, K: lysine, F: phenylalanine. Bar height: average, error bars: standard deviation, $n = 4$ biological replicates. Two-way ANOVA and Tukey's post hoc test. Exact $P$ values are indicated in the figure and highlighted in red when significant (< 0.05). (H, I) Removal of asparagine from ASNS-knockout cells causes mTORC1 inhibition. Immunoblot from control or ASNS KO HEK293T cells incubated for 8 h in regular DMEM with or without supplementation of asparagine (N) at the indicates concentrations. Representative example (H) of three biological replicates quantified in (I). The same samples were loaded on different gels and blotted in parallel with the indicated antibodies. Protein concentration was controlled by blotting α-tubulin on a separate gel. Asparagine-starved ASNS-knockout cells are set to 1. Line: average, error bars: standard deviation. One-way ANOVA and Dunnett's post hoc test. Exact $P$ values relative to lane 2 are indicated in the figure and highlighted in red when significant (<0.05). Source data are available online for this figure.

approximately one third of the asparagine concentration in human plasma after food intake (Steginks et al, 1991).

In contrast to glutamine, asparagine undergoes very little, if any, downstream metabolism (Ubuka and Meister, 1971). Asparagine has been shown, however, to promote expression of glutamine synthetase in glutamine-depleted cells (Pavlova et al, 2018) and to serve as an exchange factor to promote the uptake of other amino acids (Krall et al, 2016). To exclude that asparagine supplementation rescues mTORC1 activity by replenishing intracellular glutamine or any other amino acid, we measured the levels of intracellular amino acids upon acute glutamine removal for 1 h with or without supplementation of asparagine. We used a tenfold excess of asparagine (250 μM) to ensure detection of any potential effect of asparagine on the abundance of other amino acids. Glutamine removal caused a 50% reduction in asparagine levels and a 30% reduction in aspartate and glutamate levels (Fig. 1F,G), while all other amino acids were unaffected or even increased, possibly as a result of reduced protein synthesis due to mTORC1 inhibition. Neither supplementation with aspartate nor with glutamate, however, rescued mTORC1 activity as strongly as asparagine (Fig. EV2E). Of note, glutamine removal caused an increase rather than a decrease in leucine levels, further indicating that this amino acid is unlikely to play a role in glutamine sensing. As expected, supplementation with 250 μM asparagine in glutamine-starved cells caused a large, more than tenfold increase in intracellular asparagine levels compared to unstarved cells. Despite the excess availability of asparagine, however, we could not detect any significant increase in glutamine, aspartate, glutamate or any other amino acid. Thus, asparagine itself is sufficient to maintain mTORC1 active in the absence of glutamine. Consistent with this conclusion, asparagine supplementation also rescued mTORC1 activity in glutamine synthetase (GLUL)-knockout cells acutely deprived of glutamine (Fig. EV2F), in which no metabolism of asparagine to glutamine would be possible.

Previous studies indicated a role for glutaminolysis in glutamine sensing through generation of glutamate and subsequent conversion of glutamate to the TCA metabolite α-ketoglutarate (Duran et al, 2012). We therefore asked if asparagine could regulate mTORC1 by indirectly affecting α-ketoglutarate levels. However, α-ketoglutarate levels did not change significantly in cells acutely deprived of glutamine, and they also did not increase upon supplementation with asparagine (Fig. EV2G), indicating that this metabolite is unlikely to signal to mTORC1 upon acute depletion of glutamine.

To determine whether metabolism to asparagine is not only sufficient, but also necessary to maintain mTORC1 activity, we then generated HEK293T cells with a knockout for asparagine synthetase (ASNS), the enzyme synthesizing asparagine from glutamine. As DMEM does not contain asparagine, ASNS-deleted cells required asparagine supplementation in the medium for survival and proliferation (Fig. EV2H). Acute removal of asparagine in ASNS-knockout cells for 1 h caused marked mTORC1 inhibition despite the presence of glutamine in the medium (Fig. EV2I), while addition of as little as 25 μM asparagine was sufficient to maintain high levels of mTORC1 activity (Fig. 1H,I).

Together, these results show that asparagine is both necessary and sufficient to maintain mTORC1 activity, indicating that asparagine is the main limiting metabolite sensed by mTORC1 during acute glutamine depletion.

## Both Rag-dependent and Rag-independent mechanisms are involved in glutamine sensing

Having defined which downstream metabolite signals glutamine sufficiency, we then turned to the underlying molecular mechanisms. According to the current model, nutrients are sensed by mTORC1 through dedicated sensors specifically recognizing each nutrient and signaling to the GATOR complex to control the

nucleotide loading state of the Rag GTPases (Goul et al, 2023; Wolfson and Sabatini, 2017). However, in contrast to leucine (Wolfson et al, 2016), arginine (Chantranupong et al, 2016) and methionine (Gu et al, 2017), no such sensor has been identified for glutamine. Moreover, whether the GATOR-Rag machinery is at all involved in glutamine sensing is unclear, with some studies indicating a Rag-dependent mechanism (Duran et al, 2012) and others a Rag-independent one (Jewell et al, 2015; Meng et al, 2020).

To better understand the contribution of the Rag GTPases in glutamine sensing, we studied the response of mTORC1 to either acute or prolonged removal of this amino acid in cells where the Rag GTPases were either constitutively active or deleted. In control HEK293T cells, glutamine removal from 1 to 8 h caused progressive mTORC1 inhibition, as indicated by decreased phosphorylation of three direct mTORC1 substrates, S6K1, 4EBP1, and ULK1 (Figs. 2A,B and EV3A,B). Comparable dynamics of mTORC1 inhibition could also be observed in ASNS-knockout cells deprived of asparagine (Fig. EV3C,D). To generate cell lines with constitutive activation of the Rag GTPases, we either overexpressed a GTP-locked RagB$^{short}$ mutant protein that abolishes its GTPase activity (Sancak et al, 2008) (Fig. EV3E,F) or disrupted the GATOR1 subcomplex by deleting its subunit DEPDC5 (Bar-Peled et al, 2013) (Fig. 2C,D). Strikingly, in both cases, constitutive activation of the Rag GTPases completely abolished mTORC1 inhibition upon acute glutamine removal for 1 h, but did not prevent progressive mTORC1 inhibition at later time points. As a control, constitutive activation of the Rag GTPases also blunted mTORC1 inhibition upon a short course of leucine starvation (Figs. 2C,D and EV3E,F), consistent with the established role of the GATOR-Rag machinery in leucine sensing (Wolfson et al, 2016). In a converse approach, we then tested the response of mTORC1 to glutamine removal in the absence of functional Rag GTPases using a cell line lacking the two paralogous isoforms RagA and RagB. As previously reported (Figlia et al, 2022; Jewell et al, 2015), RagAB double-knockout cells growing in control, glutamine-replete conditions have mTORC1 activity that is strongly reduced but still detectable (Fig. EV3G). Glutamine removal did not, however, cause further reduction of mTORC1 activity for the first 1–2 h (Fig. EV3G,H), after which we observed progressive inhibition of mTORC1, thus recapitulating the response of cells with constitutively active Rag GTPases (Figs. 2C,D and EV3E,F).

Together, these results indicate that both Rag-dependent and Rag-independent mechanisms are involved in glutamine sensing, although with different dynamics: while sensing of acute glutamine depletion largely relies on the Rag GTPases, additional mechanisms independent of the GATOR-Rag axis play an important role when glutamine depletion persists longer than 1–2 h.

## mTORC1 inhibition upon glutamine depletion is entirely mediated by GCN2

Since glutamine is sensed by mTORC1 in a partially Rag-independent manner, we tested the involvement of other signaling pathways known to regulate mTORC1. GCN2 and AMPK are two evolutionarily conserved nutrient-sensing pathways that respond to starvation of proteogenic amino acids and to energy stress, respectively (Herzig and Shaw, 2018; Masson, 2019; Steinberg and Hardie, 2023) and have been shown to impinge on mTORC1 (Dai et al, 2023; Gwinn et al, 2008; Inoki et al, 2003; Sinha et al, 2024; Ye et al, 2015). Since both glutamine and asparagine are proteogenic amino acids, activation of GCN2 could

play a role in glutamine sensing. Additionally, since glutamine is an important energy substrate (Zhang et al, 2017), removal of glutamine could cause activation of AMPK. Interestingly, a potential role of asparagine in regulating AMPK has also been reported (Krall et al, 2016). We failed, however, to observe any appreciable increase in AMPK activity upon glutamine removal, as indicated by phosphorylation of two direct substrates of AMPK, ACC and Raptor, in contrast to glucose starvation, which strongly activated AMPK (Fig. 2A). Instead, glutamine removal promptly and potently increased GCN2 activity, as indicated by autophosphorylation on T899, phosphorylation of its substrate eIF2α and accumulation of ATF4, to a similar extent as that caused by complete amino acid starvation or treatment with tunicamycin (Fig. 2A), a potent activator of the unfolded protein response (UPR). Consistent with no detectable activation of AMPK, the response of mTORC1 to glutamine deprivation still occurred normally in cells where both AMPK catalytic subunits α1 and α2 were simultaneously knocked out (Fig. EV4A,B). As expected, AMPK-knockout cells showed resistance to mTORC1 inhibition by glucose starvation, consistent with glucose being partially sensed through this signaling pathway (Efeyan et al, 2013; Orozco et al, 2020).

Strikingly, in contrast to AMPK, deletion of GCN2 rendered mTORC1 completely resistant to glutamine starvation both during acute depletion for 1 h and at later time points (Fig. 2E,F), suggesting that both the Rag-independent mechanism during prolonged glutamine depletion, as well as the Rag-dependent one, act downstream of GCN2. GCN2-knockout cells nonetheless responded to complete amino acid starvation, indicating that they are still capable of sensing other amino acids, for instance via the leucine, arginine and methionine dedicated sensors. Resistance of mTORC1 to glutamine deprivation could also be observed in control cells acutely treated with GCN2iB, a pharmacological inhibitor of GCN2 (Fig. EV5A,B), arguing against the possibility that the lack of a response in GCN2-knockout cells is due to chronic disruption of this signaling pathway. Acute treatment with GCN2iB could also prevent mTORC1 inhibition in all other cell lines we tested (Fig. EV5C–H), indicating a general role of GCN2 in glutamine sensing.

Given the importance of asparagine downstream of glutamine for mTORC1 regulation, we next tested how deprivation or supplementation of asparagine affects GCN2 activity. Similar to glutamine removal in control cells, removal of asparagine in ASNS-knockout cells also strongly activated GCN2 within 1 h (Fig. 2G). Deletion of GCN2 in ASNS-knockout cells was sufficient to completely block mTORC1 inhibition at all time points upon asparagine removal (Fig. 2G,H), consistent with asparagine regulating mTORC1 in an entirely GCN2-dependent mechanism. Conversely, supplementation of asparagine prevented GCN2 activation in control cells deprived of glutamine for 1 h (Fig. EV5I, lanes 1–3), confirming that asparagine is the most limiting amino acid upon acute glutamine depletion. However, we also reasoned that prolonged removal of glutamine should eventually lead to glutamine levels becoming limiting and consequently to activation of GCN2 and inhibition of mTORC1 despite asparagine supplementation, since GCN2 senses the lack of any amino acid. Indeed, that is what we observed when cells were depleted of glutamine and supplemented with asparagine for longer than 1–2 h (Fig. EV5I, lanes 4–6, Fig. EV5J). Thus, metabolism to asparagine is most important during the acute response to glutamine removal, but less so when glutamine levels are persistently low.

Together, these results indicate that glutamine sensing by mTORC1 is entirely mediated by the general amino acid sensing

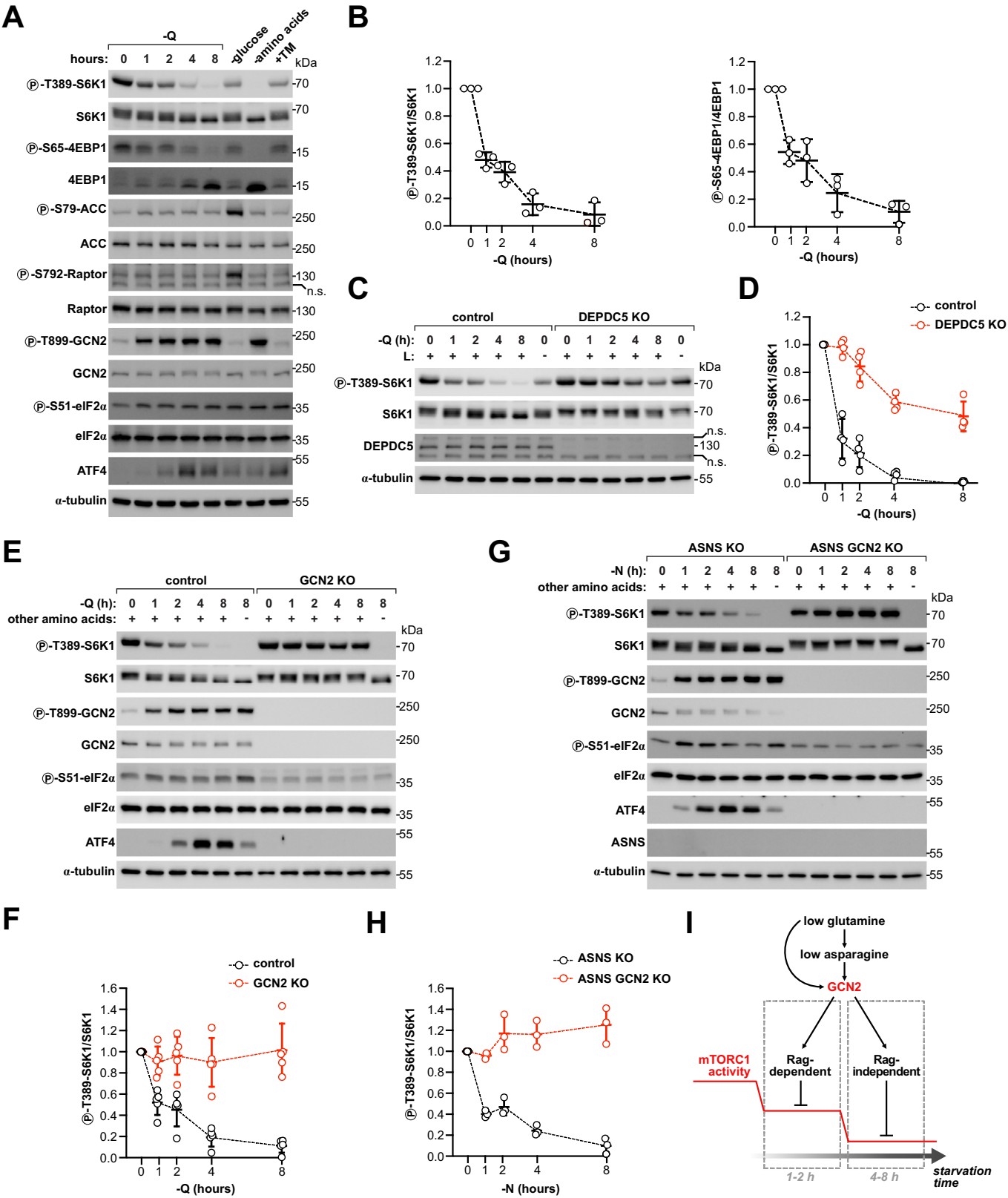

**Figure 2. Glutamine is sensed through both Rag-dependent and Rag-independent mechanisms downstream of GCN2.**

(A, B) Glutamine removal activates GCN2 but not AMPK. Immunoblot from HEK293T cells incubated in regular or glutamine-free DMEM for the indicated time points. As a control, cells were incubated in glucose-free DMEM for 2 h, amino acid-free DMEM for 8 h or treated with tunicamycin (TM, 1 μg/ml) for 8 h. n.s. non-specific band. Representative example (A) of four biological replicates quantified in (B). The same samples were loaded on different gels and blotted in parallel with the indicated antibodies. Protein concentration was controlled by blotting α-tubulin on a separate gel. Glutamine-rich condition for each genotype is set to 1. Q: glutamine, line: average, error bars: standard deviation. (C, D) Acute mTORC1 inhibition upon glutamine removal is GATOR1-dependent. Immunoblot from control or DEPDC5 KO HEK293T cells incubated in regular or glutamine-free DMEM for the indicated time points. As a control, cells were incubated in leucine-free DMEM for 2 h. Q glutamine, L leucine. n.s.: non-specific bands. Representative example (C) of four biological replicates quantified in (D). The same samples were loaded on different gels and blotted in parallel with the indicated antibodies. Protein concentration was controlled by blotting α-tubulin on a separate gel. Glutamine-rich condition for each genotype is set to 1. line: average, error bars: standard deviation. (E, F) Both acute and late inhibition of mTORC1 to glutamine removal is GCN2-dependent. Immunoblot from control or GCN2 KO HEK293T cells incubated in regular or glutamine-free DMEM for the indicated time points. As a control, cells were incubated in amino acid-free DMEM for 8 h. Q glutamine. Representative example (E) of five biological replicates quantified in (F). The same samples were loaded on different gels and blotted in parallel with the indicated antibodies. Protein concentration was controlled by blotting α-tubulin on a separate gel. Glutamine-rich condition for each genotype is set to 1. line: average, error bars: standard deviation. (G, H) mTORC1 senses asparagine in a GCN2-dependent manner. Immunoblot from ASNS KO or ASNS GCN2 double KO HEK293T cells incubated in DMEM supplemented with 250 μM asparagine (N) or in regular DMEM, which does not contain asparagine, for the indicated time points. As a control, cells were incubated in amino acid-free DMEM for 8 h. Representative experiment (G) of three biological replicates quantified in (H). The same samples were loaded on different gels and blotted in parallel with the indicated antibodies. Protein concentration was controlled by blotting α-tubulin on a separate gel. Cells supplemented with asparagine (N) for each genotype are set to 1. Line: average, error bars: standard deviation. (I) The response of mTORC1 to acute glutamine depletion is mediated by the Rag GTPases, while the response at later time points is Rag-independent. Both Rag-dependent and Rag-independent mechanisms are however downstream of GCN2. Source data are available online for this figure.

pathway GCN2, both during the initial response to acute glutamine withdrawal, as well as later in response to prolonged glutamine starvation (Fig. 2I). This argues against the presence of a dedicated sensor for glutamine/asparagine analogous to those reported for leucine, arginine and methionine.

## Rag-independent inhibition of mTORC1 is mediated by Ddit4 downstream of the ISR

Next, we investigated which mechanisms downstream of GCN2 mediate Rag-dependent and Rag-independent inhibition of mTORC1. Upon activation of GCN2 by amino acid deprivation or ribosomal stalling/collisions, GCN2 phosphorylates the translation initiation factor eIF2α at serine 51 (Costa-Mattioli and Walter, 2020). This causes a global drop in mRNA translation while specifically up-regulating translation of ATF4, a master transcription factor that promotes expression of stress response genes such as ASNS (Fig. 3A) (Siu et al, 2002). This sequence of events is known as the integrated stress response (ISR), because it is also induced by three other eIF2α kinases, PERK, HRI and PRK, which respond to different cellular stresses. One of the ATF4 targets is *Sesn2* (Ye et al, 2015), the gene coding for the leucine sensor Sestrin2 (Wolfson et al, 2016). GCN2-dependent induction of Sestrin2 downstream of the ISR was previously shown to contribute to mTORC1 inhibition during long-term (>20 h) deprivation of multiple amino acids (Ye et al, 2015), representing the best-established mechanistic connection between GCN2 and mTORC1. To test whether Sestrin2 could also be involved in mTORC1 inhibition during the initial 8 h of glutamine deprivation, we first measured the induction of Sestrin2 by qPCR. While *Sesn2* levels and those of other ATF4-induced transcripts all showed a time-dependent response to glutamine removal, indicating activation of the ISR, substantial accumulation was not observed until at least 2 h (Fig. 3B). Moreover, at the protein level, Sestrin2 did not increase significantly until 8 h (Fig. EV6A), indicating that induction of Sestrin2 expression downstream of the ISR is unlikely to play a role within the first 8 h of glutamine deprivation. Similarly, we could observe substantial accumulation of Sestrin2 in ASNS-knockout

cells only after 8 h of asparagine deprivation (Fig. EV6B). Consistent with Sestrin2 being induced through activation of GCN2 and the ISR (Ye et al, 2015), removal of glutamine or asparagine failed to increase Sestrin2 expression in the absence of GCN2 (Fig. EV6A,B). We then generated a cell line lacking Sestrin2 and the two related leucine-sensing proteins Sestrin1 and Sestrin3 (Wolfson et al, 2016), so as to exclude potential compensatory effects. Despite a mild increase in mTORC1 activity at all time points, deletion of all three isoforms did not affect the dynamics of mTORC1 inhibition, confirming that neither Sestrin2 nor the other two related proteins play a major role in the response to glutamine deprivation during the first 8 h (Fig. EV6C,D). As a control, Sestrin triple-knockout cells showed less mTORC1 inhibition upon leucine starvation (Fig. EV6C), consistent with the function of these proteins as leucine sensors for the mTORC1 pathway (Wolfson et al, 2016). In line with previous data (Ye et al, 2015), however, Sestrin triple-knockout cells did show resistance to mTORC1 inhibition after overnight glutamine removal, indicating that while initially dispensable, induction of Sestrin2 is required to inhibit mTORC1 upon long-term starvation (Fig. EV6E). In line with this conclusion, Sestrin2 levels were more strongly elevated after overnight glutamine removal than at 8 h (Fig. EV6F).

We next tested the contribution of other known ATF4 targets. Among these, to our knowledge, only Ddit4 has been previously characterized as an inhibitor of mTORC1. Ddit4 was shown to act through the TSC complex (Brugarolas et al, 2004; Corradetti et al, 2005; Sofer et al, 2005), which is the GAP for Rheb (Battaglioni et al, 2022; Goul et al, 2023; Kim and Guan, 2019; Liu and Sabatini, 2020), and to be induced by unfolded proteins and mitochondrial stress through PERK (Whitney et al, 2009) and HRI (Condon et al, 2021), respectively. Whether Ddit4 can also be induced by amino acid starvation in a GCN2-dependent manner is however unknown. Indeed, we observed that Ddit4 mRNA levels increased strongly during glutamine starvation, with a twofold increase as early as 1 h after glutamine removal and thus with faster dynamics than *Sesn2* or other ATF4 targets (Fig. 3B). Correspondingly, we detected a progressive accumulation of Ddit4 protein starting from 4 h after glutamine or asparagine removal (Figs. 3C and EV6B). Consistent with

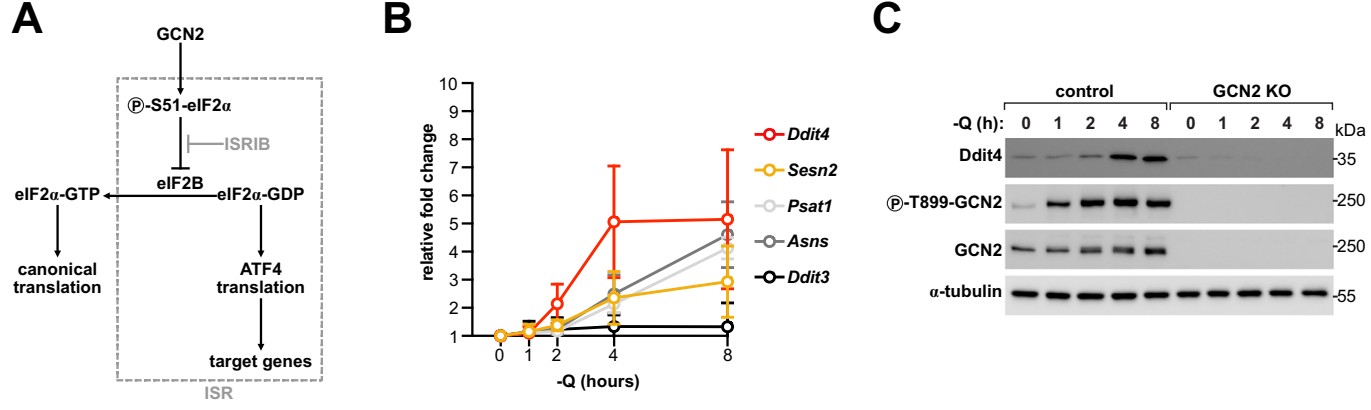

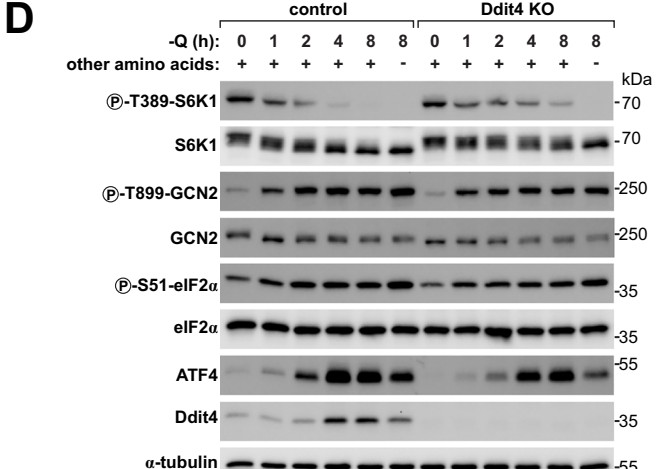

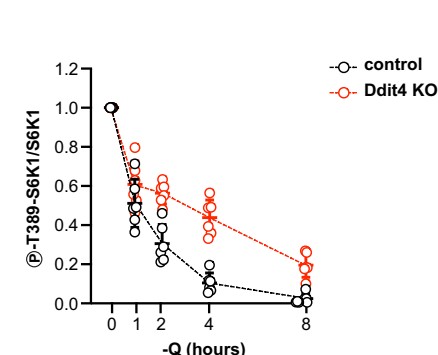

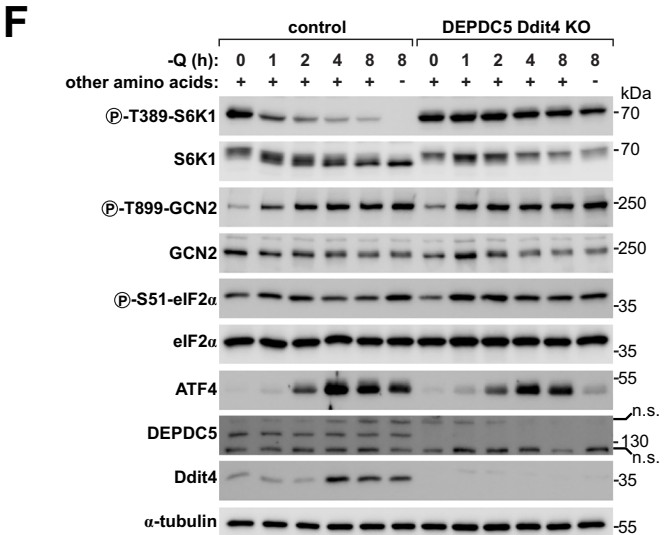

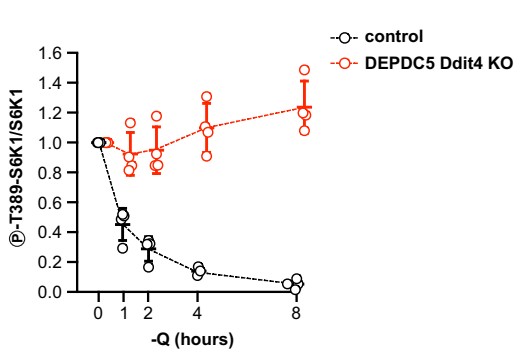

**Figure 3.   Rag-independent inhibition of mTORC1 is mediated by Ddit4 downstream of the ISR.**

(A) Graphical scheme of the integrated stress response (ISR) downstream of GCN2. Phosphorylation of eIF2α by GCN2 causes inhibition of eIF2B and, as a consequence, inhibition of canonical translation and increased translation of the transcription factor ATF4. (B) qPCR analysis of the transcript levels of selected ATF4 targets from HEK293T cells incubated in regular or glutamine-free DMEM for the indicated time points. Q: glutamine, circle: average, error bars: standard deviation, $n = 3$ biological replicates. (C) Ddit4 upregulation upon glutamine removal is mediated by GCN2. Immunoblot from control or GCN2 KO HEK293T cells incubated in regular or glutamine-free DMEM for the indicated time points. Q glutamine. Representative of three biological replicates. The same samples were loaded on different gels and blotted in parallel with the indicated antibodies. Protein concentration was controlled by blotting α-tubulin on a separate gel. (D, E) Ddit4 contributes to the late, but not the acute response to glutamine removal. Immunoblot from control or Ddit4 KO HEK293T cells incubated in regular or glutamine-free DMEM for the indicated time points. As a control, cells were incubated in amino acid-free DMEM for 8 h. Q glutamine. Representative example (D) of six biological replicates quantified in (E). The same samples were loaded on different gels and blotted in parallel with the indicated antibodies. Protein concentration was controlled by blotting α-tubulin on a separate gel. Glutamine-rich condition for each genotype is set to 1. Line: average, error bars: standard deviation. (F, G) Ddit4 mediates Rag-independent mTORC1 inhibition by glutamine deprivation. Immunoblot from control or DEPDC5 Ddit4 double KO HEK293T cells incubated in regular or glutamine-free DMEM for the indicated time points. As a control, cells were incubated in amino acid-free DMEM for 8 h. Q glutamine. Representative example (F) of three biological replicates quantified in (G). The same samples were loaded on different gels and blotted in parallel with the indicated antibodies. Protein concentration was controlled by blotting α-tubulin on a separate gel. Glutamine-rich condition for each genotype is set to 1. Line: average, error bars: standard deviation. Source data are available online for this figure.

a mechanism depending on ISR induction downstream of GCN2, Ddit4 expression failed to increase in cells lacking GCN2 (Figs. 3C and EV6B). While deletion of Ddit4 did not prevent mTORC1 inhibition after acute glutamine removal for 1 h, it did however blunt the inhibition at subsequent time points and in particular between 2 and 4 h (Fig. 3D,E), in line with the dynamics of Ddit4 accumulation (Fig. 3C). We noticed that these time points correspond to the onset of Rag-independent mTORC1 inhibition (Figs. 2C,D and EV3E–H). Since Ddit4 acts through the TSC complex and thus independently of the Rag GTPases, we reasoned that induction of Ddit4 could be the underlying Rag-independent mechanism causing mTORC1 inhibition. To test this, we deleted Ddit4 in cells lacking the GATOR1 subunit DEPDC5. Knockout of DEPDC5 removes the Rag-dependent signaling branch downstream of glutamine, leaving only the Rag-independent branch to regulate mTORC1. If Ddit4 is responsible for this Rag-independent branch, then the double-knockout cells should be completely insensitive to glutamine removal. Indeed, cells lacking both Ddit4 and DEPDC5 showed no detectable decrease in mTORC1 activity upon glutamine deprivation up to the last time point analyzed (Fig. 3F,G), similar to GCN2-knockout cells and in contrast to cells lacking only DEPDC5 (Fig. 2C,D). This indicates that Ddit4 is responsible for the entire Rag-independent regulation of mTORC1 during the first 8 h of glutamine removal. Consistent with a mechanism dependent on the TSC complex, glutamine depletion failed to inhibit mTORC1 in TSC2-knockout MEFs, despite even stronger induction of Ddit4 as compared to control cells (Fig. EV6G). Disruption of the TSC complex also abolished mTORC1 inhibition after acute glutamine removal for 1 h (Fig. EV6G), as recruitment of the TSC complex by inactive Rag GTPases contributes to mTORC1 inhibition (Demetriades et al, 2014).

Together, these results indicate that GCN2-mediated induction of Ddit4 through the ISR is the main Rag-independent mechanism inhibiting mTORC1 during the first 8 h of glutamine deprivation. If glutamine deprivation persists, slower induction of Sestrin2 contributes subsequently to additional mTORC1 inhibition.

## Inhibition of mTORC1 upon acute glutamine depletion does not require induction of ATF4

As neither deletion of Sestrin2 nor of Ddit4 could prevent mTORC1 inhibition upon acute glutamine depletion for 1 h, we asked whether this could be mediated by another uncharacterized

target of ATF4. To this end, we first tested whether ATF4 is at all involved in mTORC1 inhibition upon acute glutamine depletion using ATF4-knockout cells. While generating these lines, we noticed that ATF4 deletion severely compromised cell viability and proliferation, and that this could be rescued by supplementing the medium with asparagine. Indeed, ATF4-knockout cells showed dramatically reduced levels of ASNS protein (Fig. EV7A,B), indicating a role for ATF4 not only in stress-induced expression, but also in the basal expression of this enzyme, in agreement with previous observations in cancer cell lines (Ye et al, 2010). Consistent with reduced intracellular asparagine levels, ATF4-knockout cells showed basal elevation of GCN2 activity and partial inhibition of mTORC1 when not supplemented with asparagine (lanes 1 vs 7, Fig. EV7A). Nonetheless, glutamine removal was able to further inhibit mTORC1 at 1 h (Fig. EV7A), indicating that the acute response to glutamine deprivation is ATF4-independent. The response to glutamine removal at later time points, however, was ATF4-dependent, in agreement with it being mediated by Ddit4 (above). To see this effect more clearly, we circumvented the reduced mTORC1 activity in ATF4-knockout cells by supplementing the medium of both control and ATF4-knockout cells with asparagine. We then assessed their response to combined glutamine and asparagine removal (Fig. EV7B,C). As expected, supplementation of asparagine to ATF4-knockout cells prevented basal activation of GCN2 and increased basal mTORC1 activity back to control levels. Upon combined glutamine/asparagine deprivation, ATF4-knockout cells again showed a monophasic response with mTORC1 inhibition after 1 h, but no further inhibition at later time points, confirming that the initial response is ATF4-independent. As an additional approach to circumvent the lower mTORC1 activity in ATF4-knockout cells, we restored ASNS expression through stable transfection. ASNS-expressing ATF4-knockout cells did not depend any more on asparagine supplementation for growth and showed basal mTORC1 activity comparable with control cells (Fig. 4A). Consistent with the data above, glutamine removal caused an initial drop in mTORC1 activity comparable to control cells, but no additional inhibition at later time points (Fig. 4A,B).

Together, these data indicate that induction of ATF4 target genes downstream of the ISR is required during prolonged glutamine deprivation, but not for the response to acute depletion for 1 h.

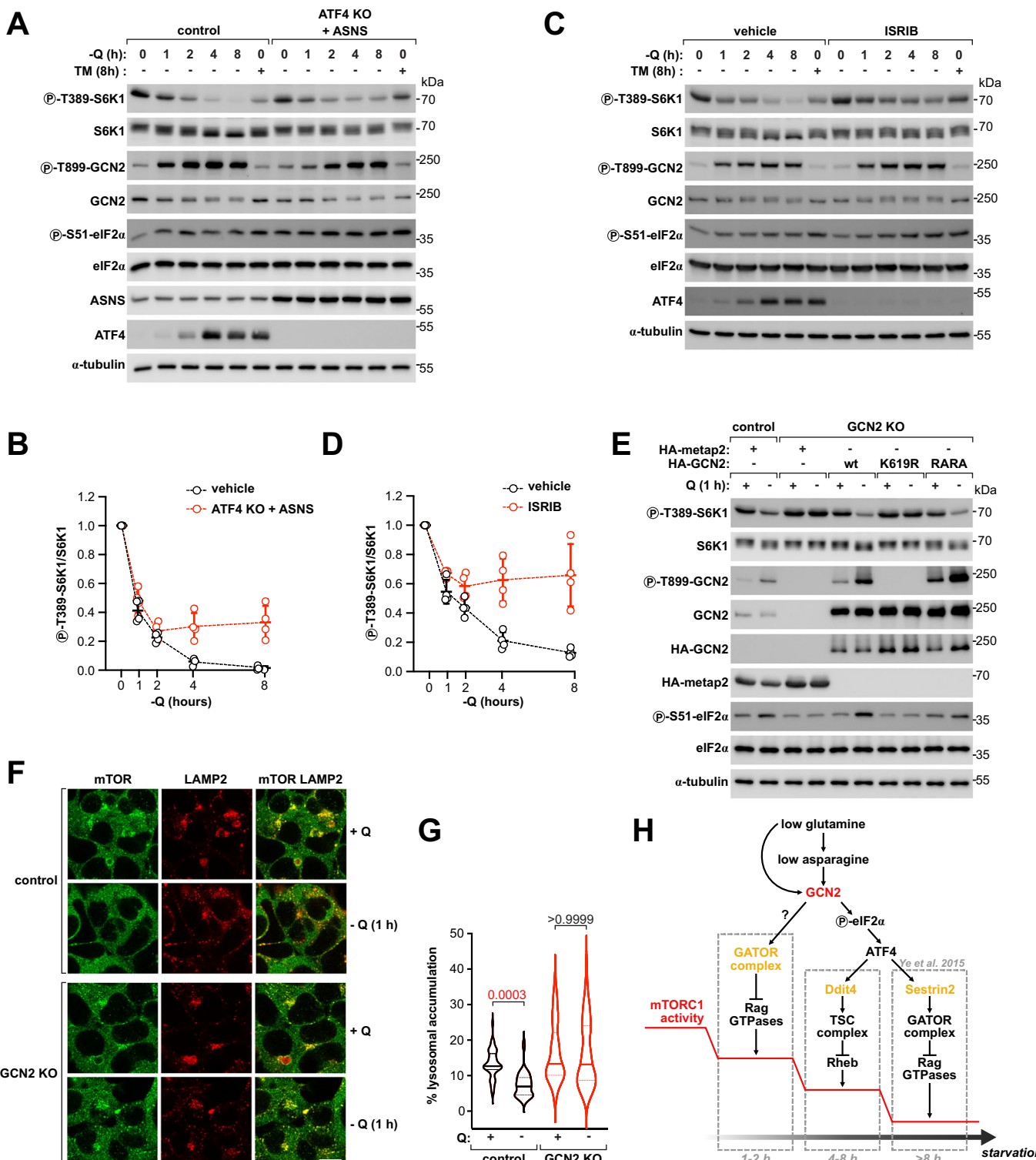

## Inhibition of mTORC1 upon acute glutamine depletion does not require eIF2α phosphorylation

The data presented above indicate that the inhibition of mTORC1 upon glutamine removal for 1 h is GCN2-dependent but ATF4-independent. We therefore asked whether the step in between—

inhibition of eIF2B by phosphorylated eIF2α (Fig. 3A)—is required for mTORC1 inhibition. Indeed, phosphorylation of eIF2α has the capacity to promote translation of other transcripts besides ATF4 (Sidrauski et al, 2015), raising the possibility that these could mediate mTORC1 inhibition. To test this, we took advantage of ISRIB, a compound which relieves the inhibition of eIF2B by

◄ **Figure 4.  mTORC1 inhibition upon acute glutamine depletion does not require induction of the ISR.**

(A, B) Acute mTORC1 inhibition after glutamine removal is ATF4-independent. Immunoblot from control or ATF4 KO HEK293T cells stably transfected with ASNS were incubated in regular or glutamine-free DMEM for the indicated time points. As a control, cells were treated with tunicamycin (TM, 1 μg/ml) for 8 h. Q glutamine. Representative example (A) of three biological replicates quantified in (B). Glutamine-rich condition for each genotype is set to 1. The same samples were loaded on different gels and blotted in parallel with the indicated antibodies. Protein concentration was controlled by blotting α-tubulin on a separate gel. Line: average, error bars: standard deviation. (C, D) Restoring eIF2B function with ISRIB does not prevent acute mTORC1 inhibition after glutamine removal. Immunoblot from HEK293T cells treated with vehicle (DMSO) or 200 nM ISRIB and incubated in regular or glutamine-free DMEM for the indicated time points. As a control, cells were treated with tunicamycin (TM, 1 μg/ml) for 8 h. Q glutamine. Representative example (C) of four biological replicates quantified in (D). The same samples were loaded on different gels and blotted in parallel with the indicated antibodies. Protein concentration was controlled by blotting α-tubulin on a separate gel. Glutamine-rich condition for vehicle or ISRIB are set to 1. Line: average, error bars: standard deviation. (E) Acute mTORC1 inhibition upon glutamine removal requires GCN2 catalytic activity. Immunoblot from control or GCN2 KO HEK293T cells stably transfected with a HA-tagged negative control protein (HA-metap2), HA-tagged wild-type or kinase-dead (K619R) GCN2 or a GCN2 mutant unable to bind to PP1 (RARA). Cells were incubated in regular or glutamine-free DMEM for 1 h. Q glutamine. Representative of three biological replicates. The same samples were loaded on different gels and blotted in parallel with the indicated antibodies. Protein concentration was controlled by blotting α-tubulin on a separate gel. (F, G) Glutamine removal displaces mTOR from lysosomes in a GCN2-dependent manner. Immunostaining for mTOR and the lysosomal marker LAMP2 in control or GCN2 KO HEK293T cells incubated in regular or glutamine-free DMEM for 1 h. Q glutamine. Representative example (F) of three biological replicates quantified in (G). Data are shown as violin plots of the percentage of LAMP2 signal overlapping with the mTOR signal in each field of view. $n = 48$ fields of view per condition from three biological replicates (16 fields of view per replicate). Continuous line: median, dashed line: first and third quartiles. Two-way ANOVA and Tukey's post hoc test. Exact P values are indicated in the figure and highlighted in red when significant (<0.05). Scale bar: 20 μm. (H) Graphical scheme of the sequential mechanisms inhibiting mTORC1 in response to glutamine deprivation. At early time points (1–2 h), GCN2 causes inhibition of the Rag GTPases independent of the ISR, likely through phosphorylation of the GATOR complex or of a substrate which in turn impinges on this complex. At intermediate (4–8 h) and late (>8 h) time points, mTORC1 inhibition by GCN2 is instead dependent on induction of the ISR, first through increased expression of Ddit4, which inhibits mTORC1 through the TSC complex in a Rag-independent manner, and then through the leucine sensor Sestrin2, which acts through the GATOR-Rag machinery. Source data are available online for this figure.

phosphorylated eIF2α (Fig. 3A) and thus prevents all the consequences of eIF2α phosphorylation on mRNA translation (Sidrauski et al, 2013; Sidrauski et al, 2015). As expected, treatment with ISRIB blocked ATF4 accumulation upon both glutamine removal and induction of the UPR by tunicamycin, but did not alter eIF2α phosphorylation itself (Fig. 4C). Analogous to deletion of ATF4, mTORC1 inhibition upon acute glutamine depletion for 1 h was largely preserved in ISRIB-treated cells, while no additional decrease in mTORC1 activity could be detected at later time points (Fig. 4C,D), indicating that the initial response to glutamine removal is independent of eIF2B inhibition. As expected, treatment with ISRIB did not affect mTORC1 activity in GCN2-knockout cells, consistent with eIF2α acting downstream of GCN2 (Fig. EV7D). Consistent with the ISRIB results, also MEFs expressing an alanine point mutant of eIF2α that disrupts phosphorylation at S51 remained sensitive to mTORC1 inhibition upon acute glutamine deprivation for 1 h (Fig. EV7E,F), indicating that this response must be mediated by GCN2 independently of eIF2α phosphorylation. This conclusion was further confirmed by the observation that cells treated with tunicamycin, which also induces the ISR but does not activate GCN2, did not show mTORC1 inhibition at 1 or 2 h of treatment, but only at later time points when ATF4 is induced (Fig. EV7G,H).

In sum, acute inhibition of mTORC1 by glutamine deprivation is GCN2-dependent, but does not require phosphorylation of eIF2α nor induction of ATF4. Hence, it is mediated by GCN2 independent of its canonical function in the ISR.

## Acute glutamine depletion inhibits mTORC1 through a non-eIF2α substrate of GCN2

Although eIF2α has long been considered the only substrate of GCN2, a phosphoproteomics analysis in yeast has recently shown that also the β subunit of the eIF2 complex as well as GCN20, a protein regulating GCN2 in yeast, are substrates of GCN2 (Dokladal et al, 2021), raising the possibility that acute glutamine depletion could act through a substrate of GCN2 other than eIF2α.

Alternatively, a mechanism independent of GCN2 function as a protein kinase could be envisaged, for instance through interaction with and recruitment of other proteins. Indeed, GCN2 has been reported to bind to the α and γ catalytic subunits of protein phosphatase 1 (PP1) through a RVXF motif typically present in the regulatory subunits of PP1 (Stonyte et al, 2023). Reciprocal binding between PP1α/γ and GCN2 are also reported in the BioPlex database of protein–protein interactions (Huttlin et al, 2021). To test whether the kinase activity of GCN2 or rather the interaction with PP1α/γ are required for mTORC1 inhibition, we then reconstituted GCN2-knockout cells with wild-type GCN2, a kinase-dead mutant GCN2 (K619R) or a mutant lacking the RVXF motif (RARA). Consistent with a mechanism depending on the kinase activity of GCN2, expression of wild-type or the RVXF-mutant GCN2 completely rescued mTORC1 inhibition by acute glutamine deprivation, but not expression of kinase-dead GCN2 (Fig. 4E). These results are also in line with the observation above that mTORC1 inhibition can be prevented by using a pharmacological inhibitor of GCN2 (Fig. EV5A–H). As mTORC1 inhibition upon acute glutamine depletion requires the kinase activity of GCN2 but not phosphorylation of eIF2α, we conclude that it is likely mediated by phosphorylation of an additional substrate.

## mTORC1 inhibition upon acute glutamine depletion does not involve ARF1 or FBXO22

In search of the GCN2 substrate that regulates mTORC1 upon acute glutamine depletion, we first considered proteins that have already been implicated in glutamine sensing or in mTORC1 regulation downstream of GCN2. The Golgi-localized ADP-ribosylation factor 1 (ARF1) GTPase has previously been proposed to mediate glutamine sensing through a mechanism independent of the Rag GTPases, based on the evidence that its knockdown or pharmacological inhibition with Brefeldin A in RagAB-knockout cells inhibited mTORC1 (Jewell et al, 2015). Although our data point to a mechanism that is dependent on the Rag GTPases (Figs. 2C,D and EV3E–H), we nevertheless tested whether ARF1

could be involved. However, in contrast to acute glutamine depletion for 1 h, treatment with Brefeldin A for the same time at the concentration previously shown to inhibit mTORC1 in RagAB-knockout cells (Jewell et al, 2015) did not detectably affect mTORC1 activity in control or GCN2-knockout cells (Fig. EV8A).

We next turned to FBXO22, a SCF E3 ubiquitin ligase component recently reported to be phosphorylated by GCN2. Upon phosphorylation, FBXO22 was proposed to promote the ubiquitination of mTOR in the vicinity of the catalytic pocket to prevent the recruitment and phosphorylation of mTORC1 substrates (Ge et al, 2023). As for ARF1, also this mechanism would however be Rag-independent, as it would affect the intrinsic catalytic activity of mTORC1 rather than its lysosomal localization. Indeed, genetic deletion of FBXO22 did not prevent mTORC1 inhibition upon acute glutamine depletion (Fig. EV8B,C). To exclude potential compensatory effects caused by long-term deletion of FBXO22, we also acutely treated control cells with TAK-243, a pharmacological inhibitor of the ubiquitin activating enzyme (Hyer et al, 2018), to block all new ubiquitination events. TAK-243 caused lower basal mTORC1 activity, possibly due to reduced availability of amino acids released by the ubiquitin-proteasome system, as indicated by higher basal phosphorylation of GCN2, but it did not prevent additional inhibition upon glutamine removal (Fig. EV8D). Thus, neither ARF1 nor FBXO22 are likely involved in the response to acute glutamine depletion. Instead, consistent with a Rag-dependent mechanism (Figs. 2C,D and EV3E–H), we observed that acute glutamine removal for 1 h reduced the lysosomal localization of mTOR and that this effect was abolished in GCN2-knockout cells (Fig. 4F,G). We thus hypothesized that GCN2 directly phosphorylates the GATOR-Rag machinery or, alternatively, an intermediate protein which in turn impinges on the function of this machinery.

To test whether GCN2 directly phosphorylates one or more components of the GATOR-Rag machinery, we analyzed the phosphoproteome of control and GCN2-knockout cells in glutamine-rich conditions or after acute glutamine depletion for 1 h (Fig. EV9A–D; Dataset EV1). As expected, glutamine deprivation reduced the phosphorylation of multiple proteins downstream of mTORC1 in control cells (Fig. EV9B) but not in GCN2-knockout cells (Fig. EV9C), consistent with a GCN2-dependent mechanism of mTORC1 inhibition. Among the proteins with elevated phosphorylation in control but not in GCN2-knockout cells was the GCN2 substrate eIF2β, confirming activation of GCN2 by glutamine deprivation. We scrutinized our dataset for components of GATOR1 (DEPDC5, Nprl2, Nprl3), GATOR2 (Mios, Wdr24, Wdr59, Sec13, Seh1l) and KICSTOR (KPTN, ITFG2, C12orf66, Szt2), the three subcomplexes forming together the GATOR complex (Valenstein et al, 2024). Although we could detect phosphorylation of DEPDC5, Mios, Wdr24, Wdr59 and Szt2, none of these phosphorylations responded to glutamine removal and/or showed GCN2-dependency (Fig. EV9B–D). Among the other known components of the mTORC1 pathway detected in our screen, the only protein showing a pattern consistent with GCN2-dependent phosphorylation was Raptor (Fig. EV9B–D), the mTORC1 subunit directly interacting with the Rag GTPases (Sancak et al, 2008). To test whether phosphorylation of Raptor could be involved in mTORC1 inhibition upon acute glutamine depletion, we generated a cell line with strongly reduced levels of Raptor and no detectable phosphorylation of S6K1 and we reconstituted it with wild-type Raptor or with mutant versions of Raptor abolishing the phosphorylation site detected in our analysis (S722A), phosphorylation of an adjacent residue (S721A) predicted to be phosphorylated by GCN2 in an in vitro kinome analysis (Johnson et al, 2023) or both (S721A, S722A). However, while all mutant constructs restored S6K1 phosphorylation to comparable levels as the wild-type protein, none of them prevented mTORC1 inhibition upon acute glutamine depletion (Fig. EV9E). Thus, while we cannot completely exclude that GCN2 phosphorylates other known components of the mTORC1 pathway not detected in our screen, these results rather point to the possibility that GCN2 controls the GATOR-Rag machinery indirectly through phosphorylation of a substrate not previously implicated in mTORC1 signaling. Further work will be required to identify this substrate.

In sum, we propose a model whereby, upon activation, GCN2 phosphorylates a non-eIF2α substrate, which in turn impinges on the GATOR-Rag machinery to cause release of mTORC1 from lysosomes and thus its inactivation as the earliest event upon glutamine deprivation (Fig. 4H). If low glutamine levels persist, induction of the ISR through eIF2α phosphorylation then causes additional mTORC1 inhibition first through a Rag-independent mechanism mediated by Ddit4 and then through slower accumulation of Sestrin2. We could not observe a role for FBXO22, Arf1, or glutaminolysis in glutamine sensing by mTORC1.

## Depletion of most amino acids individually inhibits mTORC1 through GCN2

Since GCN2 is activated by deprivation of any amino acid (Dong et al, 2000), we reasoned that this could potentially provide a general mechanism to relay the availability of any amino acid to mTORC1. To test this, we depleted control and GCN2-knockout cells of each of the 15 amino acids present in DMEM. Indeed, depletion of each amino acid in control cells acutely for 1 h resulted in activation of GCN2 and mTORC1 inhibition, with the exception of glycine, cysteine and histidine (Fig. 5A,B). Acute depletion of glycine or cysteine neither inhibited mTORC1 nor strongly activated GCN2, indicating that intracellular levels of these amino acids can be preserved in the short term without uptake from the medium. A time course of histidine depletion showed that depletion of this amino acid does lead to mTORC1 inhibition, albeit with somewhat slower kinetics compared to most other amino acids (Fig. EV10A,B). Among all amino acids, leucine and arginine were those whose depletion caused the strongest mTORC1 inhibition, in line with previous results (Hara et al, 1998). In stark contrast to control cells, mTORC1 was completely resistant to amino acid depletion in GCN2-knockout cells (Figs. 5C,D and EV10A,B), with the exception of arginine, leucine, methionine and serine, for which we still observed a partial response. Thus, GCN2 contributes toward the inhibition of mTORC1 in response to the removal of any amino acid.

For arginine, leucine, and methionine, dedicated sensors acting through the GATOR-Rag machinery have been identified (Chantranupong et al, 2016; Gu et al, 2017; Rebsamen et al, 2015; Wang et al, 2015; Wolfson et al, 2016). The observation that mTORC1 inhibition is only partially GCN2-dependent for these amino acids suggests that GCN2 and the dedicated amino acid sensors function as two independent mechanisms, with the dedicated sensors

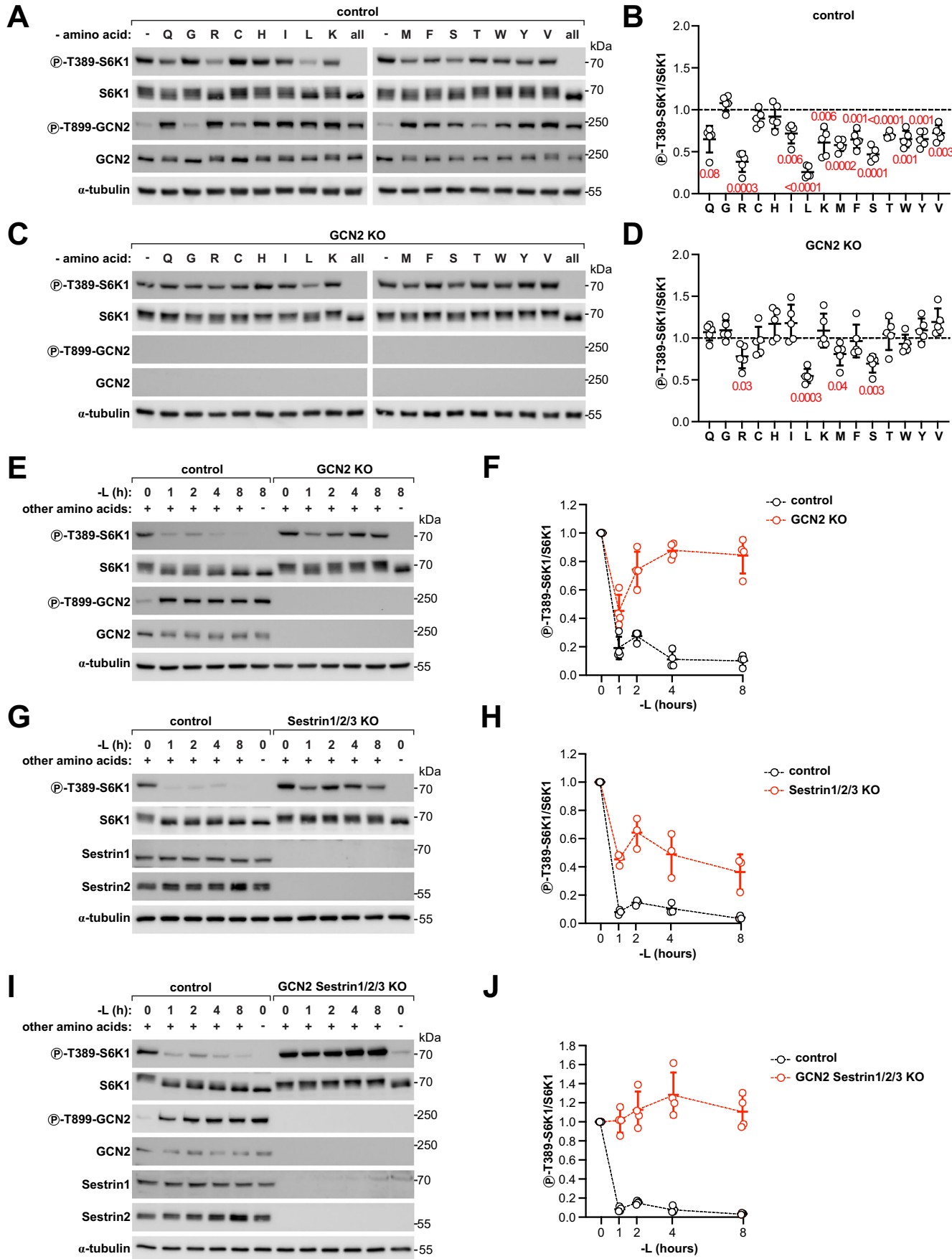

◀ **Figure 5.   GCN2 mediates inhibition of mTORC1 upon deprivation of most amino acids.**

(**A–D**) Removal of most amino acids individually inhibits mTORC1 in a GCN2-dependent manner. Immunoblot from control (**A, B**) or GCN2 KO (**C, D**) HEK293T cells incubated for 1 h in regular DMEM or in DMEM lacking each of the indicated amino acids or all amino acids. Representative examples in (**A, C**) of five independent biological replicates quantified in (**B, D**), respectively. The same samples were loaded on different gels and blotted in parallel with the indicated antibodies. Protein concentration was controlled by blotting α-tubulin on a separate gel. Quantifications are expressed as fold change compared to cells incubated in regular DMEM containing all amino acids. Line: average, error bars: standard deviation, $n = 5$ biological replicates. Exact $P$ values by one-sample $t$ and Wilcoxon test are indicated in the figure and highlighted in red when significant (<0.05). Q glutamine, G glycine, R arginine, C cysteine, H histidine, I isoleucine, L leucine, K lysine, M methionine, F phenylalanine, S serine, T threonine, W tryptophan, Y tyrosine, V valine. (**E–H**) GCN2 and Sestrin1/2/3 contribute both to mTORC1 inhibition upon leucine deprivation. Immunoblot from control, GCN2 KO (**E, F**) or Sestrin1/2/3 triple KO (**G, H**) HEK293T cells incubated in regular DMEM or leucine-free DMEM for the indicated time points. As a control, cells were incubated in amino acid-free DMEM for 8 h. Deletion of Sestrin3 was determined through sequencing of the corresponding genomic locus, given the poor quality of the Sestrin3 antibodies tested. L leucine. Representative examples (**E, G**) of multiple biological replicates quantified in (**F, H**), respectively. The same samples were loaded on different gels and blotted in parallel with the indicated antibodies. Protein concentration was controlled by blotting α-tubulin on a separate gel. $n = 4$ biological replicates for (**E, F**) or three biological replicates for (**G, H**). Leucine-rich condition for each genotype is set to 1. Line: average, error bars: standard deviation. (**I, J**) mTORC1 no longer senses leucine removal in GCN2-Sestrin1/2/3 quadruple-knockout cells. Immunoblot from control or GCN2-Sestrin1/2/3 KO HEK293T cells incubated in regular or leucine-free DMEM for the indicated time points. As a control, cells were incubated in amino acid-free DMEM for 8 h. L leucine. Representative example (**I**) of four biological replicates quantified in (**J**). The same samples were loaded on different gels and blotted in parallel with the indicated antibodies. Protein concentration was controlled by blotting α-tubulin on a separate gel. Leucine-rich condition for each genotype is set to 1. Line: average, error bars: standard deviation. Source data are available online for this figure.

sensing one specific amino acid and GCN2 responding to the lack of any amino acid. To confirm this possibility, we studied in more detail the relative contribution of these two sensing mechanisms in the case of leucine, an amino acid with well-defined dedicated sensors, namely Sestrin1/2/3. To this end, we used cells knockout for GCN2, for Sestrin1/2/3 or for both GCN2 and Sestrin1/2/3 and tested their response to a time course of leucine deprivation. Consistent with the results above and with previous results in MEFs (Averous et al, 2016; Ye et al, 2015), GCN2-knockout cells showed partial inhibition of mTORC1 after acute leucine depletion for 1 h, followed by a paradoxical increase in mTORC1 activity at later time points (Fig. 5E,F), indicating that GCN2 is required for both complete and persistent mTORC1 inhibition. Sestrin1/2/3-knockout cells also showed substantial mTORC1 inhibition upon leucine deprivation (Fig. 5G,H), indicating that these dedicated sensors are not the only mechanism responsible for leucine sensing. Indeed, consistent with simultaneous sensing through both Sestrin1/2/3 and GCN2, it was only upon combined disruption of both sensing mechanisms in GCN2-Sestrin1/2/3 quadruple-knockout cells that we observed a complete lack of mTORC1 inhibition upon leucine deprivation (Fig. 5I,J).

We then considered whether the same would hold true also for arginine, another important amino acid for mTORC1 for which dedicated sensors have been identified (Chantranupong et al, 2016; Rebsamen et al, 2015; Wang et al, 2015). Indeed, analogous to leucine, deprivation of arginine caused only partial and transient inhibition of mTORC1 in the absence of GCN2 (Fig. EV10C,D), suggesting that also in this case sensing through the dedicated sensors alone is insufficient to ensure a normal mTORC1 response. However, the presence of multiple distinct sensors for arginine, namely CASTOR1/2 (Chantranupong et al, 2016) and SLC38A9 (Jung et al, 2015; Rebsamen et al, 2015; Wang et al, 2015), and the role of SLC38A9 in both arginine and cholesterol sensing (Castellano et al, 2017) prevented us from studying in more detail the relative contribution of these sensors and of GCN2.

Together, these results indicate that two sets of sensing mechanisms are functioning simultaneously upon amino acid removal: activation of GCN2 enables mTORC1 to sense the limitation of virtually any amino acid, while dedicated sensors that of specific amino acids.

## GCN2 and the dedicated sensors Sestrin1/2/3 respond with different dynamics to amino acid addback

We wondered why two distinct amino acid sensing mechanisms, GCN2 and the dedicated sensors, are present in cells, given that both are capable of quickly responding to amino acid deprivation. While sensing some amino acids through both mechanisms at the same time could serve to ensure stronger mTORC1 inhibition, an additional possibility could be that GCN2 and the dedicated sensors differ in some other aspect of their responses, for instance, in how fast they release mTORC1 inhibition when amino acids become available again. We thus turned to an experimental setting in which we first depleted amino acids acutely for 1 h and then added them back for 15 min to 2 h. We studied the dynamics of mTORC1 reactivation following addback of leucine, an amino acid sensed through both GCN2 and dedicated sensors, or glutamine, an amino acid entirely sensed through GCN2. Consistent with previous observations (Wolfson et al, 2016), mTORC1 responded promptly to addback of leucine, with substantial reactivation already after 15 min (Fig. 6A,B) and full reactivation to the levels of unstarved cells within 1–2 h. Instead, reactivation of mTORC1 upon addback of glutamine followed much slower dynamics, with little appreciable increase in mTORC1 activity before 1 h (Fig. 6C,D). This delay was not the consequence of glutamine metabolism to asparagine, because the addition of both asparagine and glutamine together did not substantially accelerate mTORC1 reactivation (Fig. EV11A,B). Consistent with the importance of asparagine for glutamine sensing, ASNS-knockout cells deprived of both glutamine and asparagine failed to reactivate mTORC1 when re-stimulated with glutamine (Fig. EV11C,D). Also in this case, mTORC1 activity did not appreciably increase before 1–2 h of amino acid addback. Thus, the release of mTORC1 inhibition upon amino acid addback seems to be much slower for GCN2 than for the dedicated sensors, as suggested by the strikingly different dynamics of mTORC1 reactivation between glutamine and leucine. Consistent with this interpretation, we observed that both autophosphorylation of GCN2 and phosphorylation of eIF2α remained high until 1–2 h of amino acid addback (Fig. 6A–D). To exclude that the different dynamics of mTORC1 reactivation between leucine and glutamine only reflect how fast the intracellular levels of these amino acids are

replenished upon addback, we tested how mTORC1 responded to leucine addback in the absence of Sestrin1/2/3, i.e., in a condition in which this amino acid is entirely sensed through GCN2. Strikingly, in contrast to control cells upon leucine addback (Fig. 6A,B), cells lacking the dedicated sensors Sestrin1/2/3 did not show any appreciable increase in mTORC1 activity until 1–2 h of leucine re-addition (Fig. 6E,F), thus closely recapitulating the slower dynamics of the response to glutamine addback (Fig. 6C,D).

Together, these results support a model in which mTORC1 is additively regulated by two sets of mechanisms—GCN2 and the dedicated amino acid sensors. Although both mechanisms inhibit mTORC1 rapidly when amino acids are removed, they result in different kinetics of mTORC1 reactivation when amino acids become available again, with GCN2 being slow and the dedicated amino acid sensors fast. As a result, depletion and then addback of leucine elicits a two-phase response consisting of an initial fast reactivation of mTORC1, as the inhibition from Sestrin1/2/3 is released, followed by a slower additional increase corresponding to inactivation of GCN2 (Fig. 6G). In contrast, the response to glutamine addback lacks a fast component and is entirely determined by the slower dynamics of GCN2.

## Discussion

Glutamine is long known to be among the most important amino acids sensed by mTORC1, but the underlying mechanism has remained unclear, with previous studies pointing to either a role for the Rag GTPases (Duran et al, 2012) downstream of glutaminolysis or to a mechanism completely independent of the Rag GTPases (Jewell et al, 2015; Meng et al, 2020). Here, we systematically tested the roles of various downstream metabolites of glutamine, of the Rag GTPases and of other signaling pathways converging on mTORC1. We conclude that (1) asparagine is the primary metabolite sensed by mTORC1 during acute glutamine deprivation, (2) mTORC1 regulation in response to glutamine is entirely mediated by the general amino acid sensing kinase GCN2, and (3) both Rag-dependent and Rag-independent mechanisms are involved downstream of GCN2. Thus, in contrast to other important amino acids sensed by mTORC1, glutamine signals through a mechanism that does not require a dedicated amino acid sensor.

Despite being a non-essential amino acid, glutamine derived from endogenous synthesis does not satisfy the high demand of proliferating cells, as shown by the observation that in the absence of exogenous glutamine cells undergo cell cycle arrest and/or apoptosis, and suppress mTORC1 accordingly (Huang et al, 2017; Krall et al, 2016; Pavlova et al, 2018; Zhang et al, 2014). Supplementation of asparagine to cells cultured in glutamine-free medium was shown to be sufficient to rescue these metabolic effects (Huang et al, 2017; Krall et al, 2016; Pavlova et al, 2018; Zhang et al, 2014). Since asparagine is not metabolized further in mammalian cells, this has been attributed to the requirement of asparagine for protein synthesis (Pavlova et al, 2018) or to the uptake of arginine and other amino acids signaling to mTORC1 in exchange for intracellular asparagine (Krall et al, 2016). Our findings shed new light on these observations by showing that asparagine also plays a direct role in mTORC1 regulation through GCN2, pointing to these two signaling pathways as primary drivers of cell cycle arrest in the

absence of glutamine and of its rescue upon asparagine supplementation. Suppression of mTORC1 through this mechanism is also likely to contribute substantially to the antitumoral effect of asparagine depletion by L-asparaginase, a central component of the therapy of acute lymphoblastic leukemia and other cancers expressing low levels of ASNS (Hunger and Mulligan, 2015; Van Trimpont et al, 2022).

Although acute glutamine deprivation reduces cellular glutamate, aspartate and asparagine, besides glutamine itself, supplementation of asparagine alone maintains mTORC1 activity without affecting the abundance of the other amino acids. This observation seems at odds with the notion that depletion of any amino acid is sufficient to activate GCN2 (Masson, 2019). This particular dependency on asparagine can probably be explained by the very low basal intracellular levels of asparagine, approximately ten times less than aspartate, 100 times less than glutamate and glutamine, and lower than any other non-essential amino acid, as previously reported by others (Pavlova et al, 2018; Zhang et al, 2014) and also confirmed by our measurements. Reduction of this already limited pool could thus have a much stronger impact on cells than a comparable reduction of the levels of the other amino acids. Besides being the amino acid with the lowest abundance, asparagine is also the only amino acid whose synthesis strictly depends on glutamine. The small pool of intracellular asparagine is thus poised to function as a gauge for glutamine availability, ensuring that mTORC1 is promptly suppressed as soon as glutamine availability drops. This may enable cells to inhibit protein synthesis and maintain glutamine levels before they become critically low. Despite this safeguard mechanism, persistent deprivation of glutamine would be expected to eventually deplete intracellular levels of glutamine to the point that both low asparagine and low glutamine, and possibly also lower levels of other amino acids whose synthesis depends on glutamine, drive activation of GCN2 and suppression of mTORC1. Indeed, we observe that supplementation of asparagine can prevent GCN2 activation and mTORC1 inhibition only short term, indicating that the role of asparagine in glutamine sensing is mainly focused on detecting rapid fluctuations of glutamine levels rather than chronic changes.

Although both GCN2 and mTORC1 are amino acid-responsive pathways, they have long been considered to have distinct regulation and downstream functions, the former responding to deprivation of any amino acid to initiate a stress response and the latter to availability of selected amino acids to promote cell growth. The discovery that prolonged amino acid starvation induces Sestrin2 expression through GCN2 and the ISR (Ye et al, 2015) provided the first, and thus far best-established connection between these two signaling pathways. By defining two additional mechanisms whereby GCN2 regulates mTORC1, we show here that the cross-talk between these two pathways is more extensive than previously thought. Interestingly, not all mechanisms require induction of the ISR, typically assumed to be the main signaling output of GCN2. We find that GCN2 is capable of inhibiting mTORC1 already within 1 h, thus in a timeframe in which no major change in the expression of ISR-induced proteins is expected. Consistent with this, acute inhibition by GCN2 is independent of either eIF2α phosphorylation or ATF4, but still requires GCN2 kinase activity. We do not find any evidence for a role of FBXO22 in this context, a substrate recognition component for the SCF E3 ubiquitin ligase complex recently proposed to be phosphorylated by

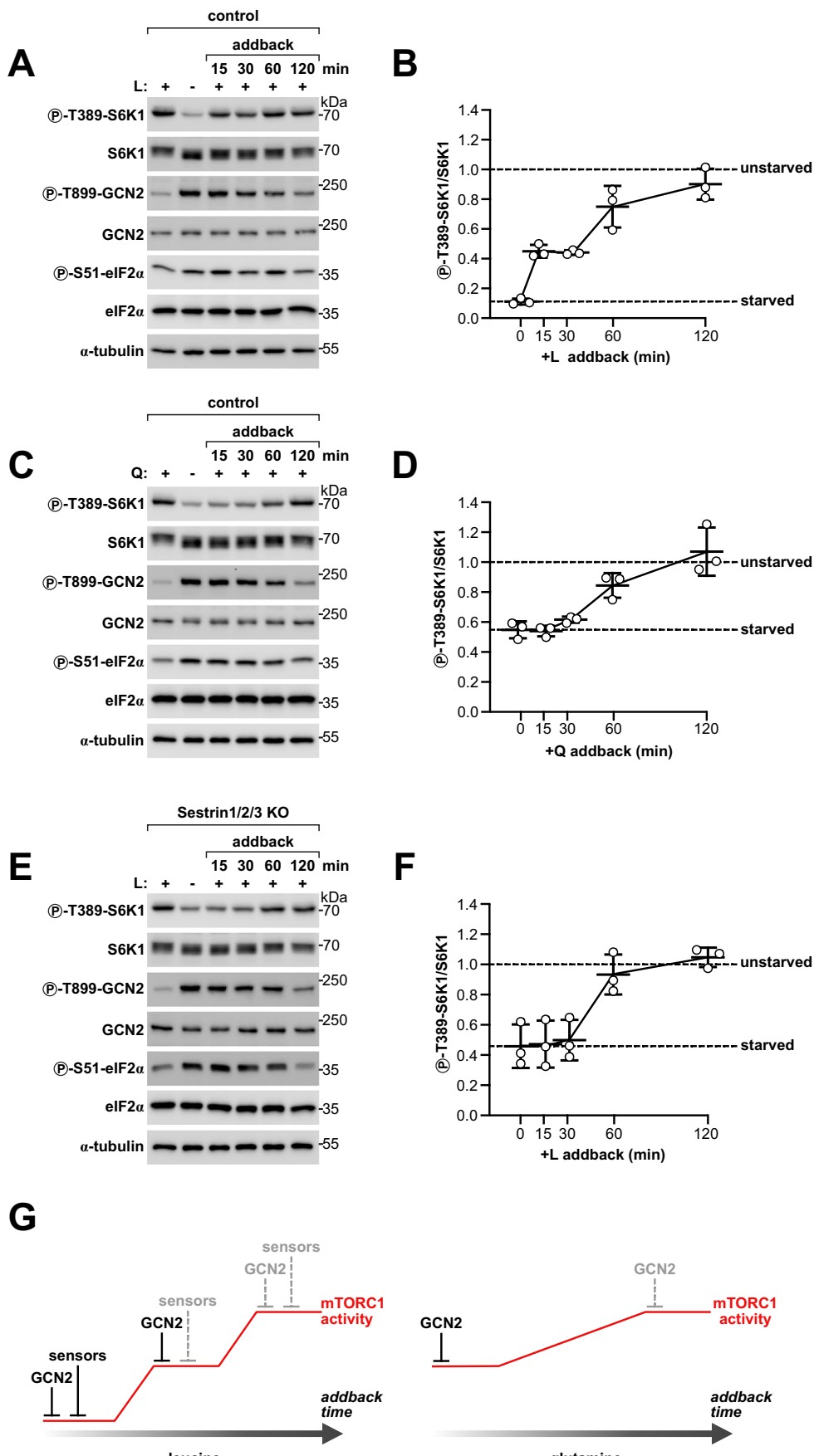

**Figure 6.   GCN2 responds more slowly to amino acid addback than the dedicated sensors Sestrin1/2/3.**

(A–D) mTORC1 is reactivated more rapidly upon addback of leucine (A, B) than of glutamine (C, D). Immunoblot from control HEK293T cells treated either with leucine removal and re-addition (A, B) or glutamine removal and re-addition (C, D). L leucine, Q glutamine. Representative examples (A, C) of three biological replicates quantified in (B, D), respectively. The same samples were loaded on different gels and blotted in parallel with the indicated antibodies. Protein concentration was controlled by blotting α-tubulin on a separate gel. Unstarved cells are set to 1. Line: average, error bars: standard deviation. (E, F) mTORC1 also is reactivated more slowly upon leucine addback in cells lacking Sestrin1/2/3. Sestrin1/2/3 KO HEK293T cells were incubated in regular or leucine-free DMEM for 1 h, or leucine-free DMEM for 1 h and then regular DMEM (with leucine) for the indicated time points. L leucine. Representative example (E) of three biological replicates quantified in (F). The same samples were loaded on different gels and blotted in parallel with the indicated antibodies. Protein concentration was controlled by blotting α-tubulin on a separate gel. Unstarved cells are set to 1. Line: average, error bars: standard deviation. (G) Graphical scheme of the response of mTORC1 to leucine and glutamine re-addition and model of the underlying mechanisms. After addback of leucine, fast release of the inhibition by Sestrin1/2/3 causes partial reactivation of mTORC1, followed by full reactivation only after slower inactivation of GCN2. In contrast, after addback of glutamine, mTORC1 response is entirely driven by inactivation of GCN2 and thus lacks a fast component as in the case of leucine. Source data are available online for this figure.

GCN2 and to promote ubiquitination of mTOR in response to amino acid starvation (Ge et al, 2023). Although we cannot exclude that FBXO22 plays a role upon deprivation of other amino acids, this would however imply that its phosphorylation by GCN2, or the functional consequences of its phosphorylation, differ depending on whether GCN2 is activated by deprivation of glutamine or of another amino acid. Instead, although the exact substrate phosphorylated by GCN2 remains elusive, we favor the possibility that GCN2 phosphorylates a protein acting on the GATOR-Rag machinery, based on the observation that (1) constitutive activation of the Rag GTPases prevents acute mTORC1 inhibition upon glutamine removal, and (2) glutamine removal does not cause displacement of mTORC1 from lysosomes in the absence of GCN2. While we cannot completely exclude that components of the Rag-GATOR machinery are directly phosphorylated by GCN2, the results of our phosphoproteomics analysis seem to indicate that GCN2 controls the Rag-GATOR machinery through an intermediate protein likely not previously implicated in mTORC1 signaling. The identification of this substrate will require further study.

Unlike the acute response, persistent and progressive inhibition of mTORC1 when cells experience prolonged glutamine deprivation requires induction of the ISR. In addition to the reported role of Sestrin2 in this context (Ye et al, 2015), we find here that activation of GCN2 by prolonged glutamine deprivation also increases the expression of Ddit4, another inhibitor of mTORC1 previously shown to be upregulated during the UPR (Whitney et al, 2009) or upon mitochondrial stress downstream of ATF4 (Condon et al, 2021). Ddit4 accumulates with much faster dynamics than Sestrin2 and, accordingly, contributes to mTORC1 inhibition already after 2–4 h of glutamine deprivation, whereas Sestrin2 increases significantly at the protein level only after 8 h and is thus not majorly involved in mTORC1 inhibition before this time point. Although the mechanistic details are still unclear, Ddit4 seems to inhibit mTORC1 by stabilizing and/or potentiating the GAP activity of the TSC complex and thus through Rheb (Brugarolas et al, 2004; Corradetti et al, 2005; Sofer et al, 2005), unlike Sestrin2 or the acute ISR-independent mechanism, which instead act through the Rag GTPases. Consistent with this, our results show that, while constitutive activation of the Rag GTPases alone does not prevent mTORC1 inhibition when glutamine deprivation persists more than 2–4 h, constitutive activation of the Rag GTPases combined with deletion of Ddit4 blocks this effect and renders mTORC1 persistently active until the last time point analyzed. Ddit4 thus represents the most important Rag-

independent mechanism downstream of GCN2. The existence of both Rag-dependent and Rag-independent mechanisms probably explains why previous studies concluded that glutamine is sensed through the Rag GTPases (Duran et al, 2012) whereas others found evidence for a mechanism independent of these small GTPases (Jewell et al, 2015). Activation of GCN2 by glutamine deprivation thus triggers at least three sequential and additive mechanisms to inhibit mTORC1: a fast, ISR-independent mechanism acting within 1 h through the Rag GTPases (mechanism 1) followed by induction of the ISR, which first causes accumulation of Ddit4 within 2–4 h (mechanism 2) and then of Sestrin2 after 8 h (mechanism 3).

The identification of dedicated sensors for the amino acids leucine (Wolfson et al, 2016), arginine (Chantranupong et al, 2016; Rebsamen et al, 2015; Wang et al, 2015) and methionine (Gu et al, 2017), as well as for cholesterol (Castellano et al, 2017; Shin et al, 2022), has led to the assumption that analogous dedicated sensors likely exist also for every other major nutrient sensed by mTORC1. We were thus surprised to find that glutamine sensing is entirely mediated by GCN2, arguing against the presence of a dedicated sensor for this amino acid. Consistent with this conclusion, mTORC1 remained sensitive to deprivation of leucine, arginine and methionine also in the absence of GCN2, but not to deprivation of glutamine. As activation of GCN2 is not unique to glutamine, it follows that the GCN2-dependent mechanisms discussed above should also be similarly triggered whenever any other amino acid is missing. Indeed, we find that deprivation of most amino acids individually causes inhibition of mTORC1 in a GCN2-dependent manner already within 1 h. Activation of GCN2 thus represents a largely overlooked mechanism coupling mTORC1 activity to availability of virtually any amino acid, not only through the slower ISR-dependent mechanisms, but also acutely, i.e., in a comparable timeframe as the responses mediated by the dedicated amino acid sensors. We propose that GCN2 and the dedicated sensors represent two independent amino acid sensing systems that regulate mTORC1 additively. These two systems show however one important functional difference, which emerged when comparing the response of mTORC1 to addback of leucine and glutamine: while the inhibition of mTORC1 by the dedicated sensors is released within a few minutes as soon as the corresponding amino acid becomes again available, the inhibition by GCN2 persists much longer. We do not currently understand what accounts for the slow dynamics of GCN2 inactivation. It has now become clear that GCN2 responds not only to the accumulation of uncharged tRNAs, but also to ribosomal collisions, which can be caused by deprivation of one or more amino acids, among other stresses (Wu

et al, 2020). Thus, an intriguing possibility could be that GCN2 activity persists until ribosomal collisions are resolved, which requires a complex series of events coordinated by the ribosomal quality control pathway (Joazeiro, 2019). Interestingly, ribosomal collisions have also been recently shown to occur upon UV exposure and to lead to mTORC1 inhibition downstream of GCN2 (Sinha et al, 2024), confirming the importance of GCN2 as a negative regulator of mTORC1 and expanding further the range of stresses converging on mTORC1.

While GCN2 and most interacting proteins and effectors are conserved in all eukaryotic organisms, most fungi lack orthologues of the dedicated amino acid sensors identified so far (Wolfson and Sabatini, 2017), and it is unclear how mTORC1 is able to sense amino acids in these species. It is tempting to speculate that, in the absence of dedicated sensors, all amino acid inputs to mTORC1 in lower eukaryotes are relayed by GCN2, which could represent the most "basic" and evolutionary ancient sensing mechanism utilized by mTORC1. Consistent with this possibility, GCN2 has been previously implicated in the response of mTORC1 to amino acid starvation in the yeast species *S. cerevisiae* (Yuan et al, 2017). The emergence of dedicated amino acid sensors later during evolution might have allowed higher eukaryotes to adapt better to variable environmental conditions, thanks to the faster dynamics of this amino acid sensing system.

# Methods

### Reagents and tools table

| Reagent/resource | Reference or source | Identifier or catalog number |
|---|---|---|
| **Experimental models** | | |
| HEK293T | ATCC | CRL-3216 |
| HEK293T GCN2 KO | This study | N/A |
| HEK293T AMPKα1/α2 KO | This study | N/A |
| HEK293T DEPDC5 KO | This study | N/A |
| HEK293T RagA/RagB KO | This study | N/A |
| HEK293T ASNS KO | This study | N/A |
| HEK293T ASNS GCN2 KO | This study | N/A |
| HEK293T Sesn1/2/3 KO | This study | N/A |
| HEK293T Ddit4 KO | This study | N/A |
| HEK293T DEPDC5 Ddit4 KO | This study | N/A |
| HEK293T ATF4 KO | This study | N/A |
| HEK293T Raptor KO | This study | N/A |
| HEK293T Sesn1/2/3 GCN2 KO | This study | N/A |
| HEK293T stably expressing HA-RagB$^{Q99L}$ | This study | N/A |
| HEK293T stably expressing HA-rap2a | This study | N/A |
| HEK293T ATF4 KO stably expressing ASNS | This study | N/A |
| HEK293T stably expressing HA-metap2 | This study | N/A |

| Reagent/resource | Reference or source | Identifier or catalog number |
|---|---|---|
| HEK293T GCN2 KO stably expressing HA-metap2 | This study | N/A |
| HEK293T GCN2 KO stably expressing HA-GCN2 | This study | N/A |
| HEK293T GCN2 KO stably expressing HA-GCN2 K619R | This study | N/A |
| HEK293T GCN2 KO stably expressing HA-GCN2 RARA | This study | N/A |
| HEK293T Raptor KO stably expressing HA-metap2 | This study | N/A |
| HEK293T Raptor KO stably expressing HA-Raptor | This study | N/A |
| HEK293T Raptor KO stably expressing HA-Raptor S721A | This study | N/A |
| HEK293T Raptor KO stably expressing HA-Raptor S722A | This study | N/A |
| HEK293T Raptor KO stably expressing HA-Raptor S721A/S722A | This study | N/A |
| HeLa | ATCC | CCL-2 |
| U2OS | ATCC | HTB-96 |
| HepG2 | ATCC | HB-8065 |
| eIF2α$^{S51S/S}$ knock-in MEF | Randal Kaufman and Georg Stoecklin | N/A |
| eIF2α$^{S51A/A}$ knock-in MEF | Randal Kaufman and Georg Stoecklin | N/A |
| p53$^{-/-}$ TSC2$^{+/+}$ MEF | David Kwiatkowski and Michael Hall | N/A |
| p53$^{-/-}$ TSC2$^{-/-}$ MEF | David Kwiatkowski and Michael Hall | N/A |
| **Recombinant DNA** | | |
| pSpCas9(BB)-2A-Puro (px459) | Addgene | 62988 |
| piggyBac-puro | This study | N/A |
| **Antibodies** | | |
| phospho-S6K1 T389 | Cell Signaling Technology | 9205 |
| S6K1 | Cell Signaling Technology | 2708 |
| phospho-4EBP1 S65 | Cell Signaling Technology | 9451 |
| 4EBP1 | Cell Signaling Technology | 9452 |
| α-tubulin | Sigma-Aldrich | T9026 |
| ASNS | Cell Signaling Technology | 20843 |
| phospho-ACC S79 | Cell Signaling Technology | 11818 |
| ACC | Cell Signaling Technology | 3662 |
| phospho-Raptor S792 | Cell Signaling Technology | 2083 |
| Raptor | Cell Signaling Technology | 2280 |

| Reagent/resource | Reference or source | Identifier or catalog number |
|---|---|---|
| phospho-GCN2 T899 | Boster | P01172 |
| GCN2 | Cell Signaling Technology | 3302 |
| phospho-eIF2α S51 | Cell Signaling Technology | 3398 |
| eIF2α | Cell Signaling Technology | 5324 |
| ATF4 | Cell Signaling Technology | 11815 |
| DEPDC5 | Abcam | ab185565 |
| HA tag | Cell Signaling Technology | 2367 |
| AMPKα1/α2 | Cell Signaling Technology | 2532 |
| Redd1/Ddit4 | Abcam | ab191871 |
| Sestrin1 | Proteintech | 21668-1-AP |
| Sestrin2 | Proteintech | 10795-1-AP |
| FBXO22 | Proteintech | 13606-1-AP |
| mTOR | Cell Signaling Technology | 2983 |
| GLUL | Cell Signaling Technology | 80636 |
| LAMP2 | Hybridoma Bank | H4B4 |
| Anti-rabbit HRP | Jackson ImmunoResearch | 111-035-003 |
| Anti-mouse HRP | Jackson ImmunoResearch | 115-035-003 |
| Anti-rabbit Alexa488 | Life Technologies | A11008 |
| Anti-mouse TRITC | Jackson ImmunoResearch | 715-025-151 |
| **Oligonucleotides and other sequence-based reagents** | | |
| gRNA oligonucleotides | This study | Dataset EV2 |
| PCR primers | This study | Dataset EV2 |
| qPCR primers | This study | Dataset EV2 |
| **Chemicals, enzymes, and other reagents** | | |
| Q5 High Fidelity DNA Polymerase | NEB | M0491 |
| High glucose, high glutamine DMEM | Life Technologies | 41965-062 |
| FBS Superior | Sigma-Aldrich | S0615 |
| Penicillin/streptomycin | Life Technologies | 15140-122 |
| L-asparagine | Sigma-Aldrich | A4159 |
| Trypsin-EDTA 0.25% | Life Technologies | 25200-056 |
| Lipofectamine 2000 | Life Technologies | 11668500 |
| Puromycin | Sigma-Aldrich | P9620 |
| Glutamine-free DMEM | Life Technologies | 31053028 |
| Dialyzed FBS | Life Technologies | A33820-01 |
| L-glutamine | Life Technologies | 25030024 |
| L-glutamate | Sigma-Aldrich | G5889 |
| Methyl-leucine | Sigma-Aldrich | L1002 |

| Reagent/resource | Reference or source | Identifier or catalog number |
|---|---|---|
| Guanosine | Sigma-Aldrich | G6264 |
| Cytidine | Sigma-Aldrich | C122106 |
| Nicotinamide | Sigma-Aldrich | N0636 |
| D-glucosamine | Sigma-Aldrich | G4875 |
| Ammonium chloride | Sigma-Aldrich | A9434 |
| Tunicamycin | Sigma-Aldrich | 654380 |
| 6-diazo-5-oxo-L-norleucine (DON) | Sigma-Aldrich | D2141 |
| GCN2iB | Selleckchem | S8929 |
| TAK-243 | MedChemExpress | HY-100487 |
| DMSO | Carl Roth | 4720.2 |
| DPBS | Life Technologies | 14040174 |
| Ammonia assay kit | Sigma-Aldrich | MAK310 |
| 10 kDa MWCO PES columns | Sartorius | VS0102 |
| Formaldehyde | ThermoFisher | 28908 |
| Saponin | Sigma-Aldrich | 47036 |
| Bovine serum albumin | AppliChem | A1391 |
| DAPI | Applichem | A1001 |
| Vectashield | Vector Labs | H-1000 |
| Trizol | Life Technologies | 15596018 |
| Maxima H Minus reverse transcriptase | ThermoFisher | EP0753 |
| primaQUANT CYBR 2x qPCR SYBRGreen Master Mix with LOW ROX | Streinbrenner | SL-9913 |
| CellTiter-Glo Luminescent Cell Viability Assay | Promega | G7570 |
| benzonase | Millipore | 70746 |
| **Software** | | |
| GraphPad Prism v10.0.0 | GraphPad | N/A |
| Excel v16.16.27 | Microsoft | N/A |
| CellProfiler v4.2.5 | Broad Institute | N/A |
| Fragpipe v20 | Kong et al, 2017 | N/A |
| MSFragger v3.8 | Kong et al, 2017 | N/A |
| **Other** | | |

## Cell culture

HEK293T (ATCC #CRL-3216), HeLa (ATCC #CCL-2), U2OS (ATCC #HTB-96) and HepG2 (ATCC #HB-8065) cells were obtained commercially. Control eIF2α$^{S51S/S}$ and eIF2α$^{S51A/A}$ knock-in MEF cells were kindly provided by Randal Kaufman and Georg Stoecklin. Control (p53$^{-/-}$) and TSC2-knockout (p53$^{-/-}$ TSC2$^{-/-}$) MEF cells were kindly provided by David Kwiatkowski and Michael Hall. All cell lines were cultured in high glucose, high glutamine DMEM (Life Technologies #41965-062) supplemented with 10% FBS Superior (Sigma-Aldrich #S0615) and 1% penicillin/streptomycin (Life Technologies #15140-122), with the exception of ASNS-knockout and

ATF4-knockout HEK293T cell lines, which were additionally supplemented with 250 µM asparagine (Sigma-Aldrich #A4159). Cells were kept in 5% $CO_2$ at 37 °C and passaged every other day with trypsin-EDTA 0.25% (Life Technologies #25200-056). All cell lines were regularly tested for *Mycoplasma* to exclude contamination. No cell authentication was performed.

## Plasmids

The pSpCas9(BB)-2A-Puro (px459) plasmid (Addgene #62988) was a kind gift of Feng Zhang (Ran et al, 2013) and was used to generate all genetically edited cell lines after cloning of oligonucleotides coding for appropriate gRNAs. For stable transfections, a modified piggyBac vector (piggyBac-puro) was generated from a piggyBac-T2A-mNeonGreen plasmid by replacing the T2A-mNeonGreen sequence with a sequence containing the multiple cloning site and BGH polyadenylation signal from pcDNA3.1(+), followed by an expression cassette for puromycin resistance under control of the SV40 promoter. The piggyBac-T2A-mNeonGreen plasmid was a kind gift of Moritz Mall (DKFZ Heidelberg). The coding sequence of GCN2 was amplified from HEK293T cDNA using Q5 High Fidelity DNA Polymerase (NEB #M0491) and cloned into the piggyBac-puro vector. PiggyBac-puro plasmids expressing metap2, rap2a, RagB$^{shortQ99L}$ and Raptor were generated from previously described plasmids (Figlia et al, 2022) after subcloning of the corresponding coding sequences. Point mutations were introduced through PCR-based site-directed mutagenesis. The GCN2 mutant unable to interact with PP1α/γ was generated by mutating the RVRF sequence (amino acids 463-466) into RARA. All plasmids were verified by Sanger sequencing.

## Generation of knockout and stable cell lines

HEK293T knockout cell lines were generated using the CRISPR/Cas9 technology. Cells were transiently transfected with px459 plasmids expressing appropriate gRNAs selected from the Brunello library (Doench et al, 2016) (Dataset EV2) using Lipofectamine 2000 (Life Technologies #11668500) as per the manufacturer's instructions. Two days after transfection, cells were seeded in 96-well plates at a concentration of one cell per well to allow individual cell clones to grow for ~2–3 weeks. After passaging, each clone was screened by immunoblot to confirm loss of the protein of interest and genotyped by Sanger sequencing of the target locus amplified by PCR using appropriate oligonucleotide primers (Dataset EV2) and cloned into TOPO vectors (Life Technologies #450640). Double-, triple- or quadruple-knockout cell lines were generated sequentially (ASNS GCN2 KO, DEPDC5 Ddit4 KO, Sestrin1/2/3 GCN2 KO) or by simultaneous transfection of multiple gRNAs targeting the genes of interest (AMPKα1/α2 KO, Sestrin1/2/3 KO).

Stable HEK293T cell lines were generated using the piggyBac transposon system (Woodard and Wilson, 2015). Cells were transiently transfected with piggyBac-puro plasmids expressing the protein of interest and a plasmid expressing piggyBac transposase at a 1:1 ratio using Lipofectamine 2000 as per the manufacturer's instructions. Starting from two days after transfection, cells were cultured in growth medium supplemented with 1 µg/ml puromycin (Sigma-Aldrich #P9620) for approximately two weeks, until a polyclonal resistant population emerged.

## Cell treatments

Amino acid starvation was performed in DMEM-based media prepared by omitting the amino acid of interest, as described below. As unstarved control or for amino acid re-stimulation after starvation, the starvation medium was supplemented with the amino acid of interest at the same concentration as in DMEM, unless indicated otherwise. For glutamine starvation, glutamine-free DMEM (Life Technologies #31053028) was supplemented with 10% dialyzed FBS (Life Technologies #A33820-01), with or without additional supplementation with 4 mM L-glutamine (Life Technologies #25030024). For asparagine starvation, regular DMEM (Life Technologies #41965-062), which does not contain asparagine, was supplemented with 10% dialyzed FBS (Life Technologies #A33820-01), with or without additional supplementation with L-asparagine (Sigma-Aldrich #A4159) at 250 µM concentration, unless indicated otherwise. For starvation of other amino acids individually, amino acid-free homemade DMEM was prepared according to the commercial formulation (Life Technologies #41965-062) and supplemented with 10% dialyzed FBS, 4 mM L-glutamine and 50x homemade amino acid mixture containing all other amino acids but the amino acid of interest. For complete amino acid starvation, amino acid-free homemade DMEM was only supplemented with 10% dialyzed FBS.

To test the role of downstream glutamine metabolism, the following metabolites were added at 4 mM concentration to glutamine-free medium prepared as described above: L-asparagine (Sigma-Aldrich #A4159), L-glutamate (Sigma-Aldrich #G5889), methyl-leucine (Sigma-Aldrich #L1002), guanosine (Sigma-Aldrich #G6264), cytidine (Sigma-Aldrich #C122106), nicotinamide (Sigma-Aldrich #N0636), D-glucosamine (Sigma-Aldrich #G4875), ammonium chloride (Sigma-Aldrich #A9434).

The following chemical inhibitors were used at the indicated concentrations and dissolved in growth medium: tunicamycin (Sigma-Aldrich #654380), 6-diazo-5-oxo-L-norleucine (DON, Sigma-Aldrich #D2141), GCN2iB (Selleckchem #S8929), TAK-243 (MedChemExpress #HY-100487). The same volume of DMSO (Carl Roth #4720.2) was used as a control.

Cells were washed once with DPBS (Life Technologies #14040174) before each treatment.

## Cell lysis and immunoblot

Cells were seeded one day before experiments in 6-well plates at a density of $1.7 \times 10^6$ (HEK293T) or $0.2 \times 10^6$ (MEF) cells per well in growth medium. After the above-described treatments, cells were washed once in ice-cold DPBS and scraped on ice in 100 µl lysis buffer (40 mM HEPES pH 7.4, 1% Tergitol, 10 mM β-glycerophosphate, 10 mM sodium pyrophosphate, 30 mM NaF, 2× Roche Complete protease inhibitor cocktail with EDTA, 1× Roche PhosSTOP phosphatase inhibitors). After incubation on ice for 5 min, lysates were cleared by maximal speed centrifugation on a tabletop centrifuge for 10 min and used immediately or stored at −80 °C until needed. For each sample, 10–20 µg of protein in roughly 10 µL of volume was loaded on multiple SDS-PAGE gels and blotted on nitrocellulose membranes following a standard immunoblot procedure. The same samples were loaded in parallel on several different gels, transferred, and blotted in parallel with the indicated antibodies. Protein concentration was controlled by

blotting α-tubulin on a separate gel. Sample volumes were controlled by subsequent Ponceau stain. Some membranes were cut to hybridize with different antibodies detecting proteins of different sizes, as indicated in the Source Data with dotted black lines. No membranes were stripped and rehybridized. After blocking in 5% skimmed milk in PBS-T for 1 h, membranes were incubated overnight with primary antibodies diluted at the indicated concentration in 5% BSA or 5% skimmed milk in PBS-T, followed by 1-h incubation in HRP-conjugated secondary antibodies. Chemiluminescent signals were detected on a Chemi-Doc Imager (Biorad). Membranes were not reblotted.

## Metabolite measurements

HEK293T cells were seeded in six-well plates at a density of $1.7 \times 10^6$ cells per well. The following day, cells were incubated in growth medium containing the metabolites used in Fig. 1B at 4 mM concentration or in glutamine-rich, glutamine-free medium, or glutamine-free medium supplemented with 250 µM asparagine for 1 h. After treatment, cells were washed once with ice-cold 0.9% NaCl solution, scraped in 1 ml 0.9% NaCl on ice, spun down at $300 \times g$ for 4 min at 4 °C and either analyzed immediately, in the case of ammonium, or snap frozen in liquid nitrogen, for all other metabolites. The levels of each metabolite were determined as described below and subsequently normalized for the total protein amount in each sample.

### Amino acids, glucosamine, and α-ketoglutarate

Metabolites were extracted from cell pellets with 0.3 ml of 0.1 M HCl in an ultrasonic ice bath for 10 min. Insoluble material was removed by centrifugation for 10 min at $25{,}000 \times g$. For derivatization with DMB (1,2-diamino-4,5-methylenedioxybenzene), 30 µl extract were mixed with 30 µl DMB reagent (5 mM DMB, 20 mM sodium hydrosulfite, 1 M 2-mercaptoethanol, 1.2 M HCl) and incubated at 100 °C for 45 min. After 10 min centrifugation, the reaction was diluted with 240 µl 10% acetonitrile, and the derivatized ketoacids were separated by reversed-phase chromatography on an Acquity HSS T3 column (100 mm × 2.1 mm, 1.7 µm, Waters) connected to an Acquity H-class UPLC system. Prior to separation, the column was heated to 40 °C and equilibrated with 5 column volumes of solvent A (0.1% formic acid in 10% acetonitrile) at a flow rate of 0.55 ml/min. Baseline separation of DMB derivates was achieved by increasing the concentration of acetonitrile (B) in buffer A as follows: 2 min 2% B, 4.5 min 15% B, 10.5 min 38% B, 10.6 min 90% B, hold for 2 min, and return to 2% B in 3.5 min. The separated derivatives were detected by fluorescence (Acquity FLR detector, Waters, excitation: 367 nm, emission: 446 nm). Amino acids were derivatized with AccQ-Tag reagent (Waters) and determined as previously described (Weger et al, 2016).

### Nucleotides

Cell pellets were processed following an adjusted protocol targeting energy carriers such as NAD/NADH and NADP/NADPH (Rost et al, 2020). Cell pellets were extracted using 250 µl cooled extraction buffer (Acetonitrile: Methanol: 15 mM ammonium acetate in H2O (3:1:1), pH 10) and incubated in an ultrasonic ice bath for 5 min. Afterwards, samples were centrifuged for 15 min at 4 °C and $13{,}000 \times g$, and the resulting supernatant was transferred to a new LC-MS-grade autosampler vial (Waters) for measuring. For

metabolite separation and detection, an ACQUITY I-class PLUS UPLC system (Waters) coupled to a QTRAP 6500+ (AB SCIEX) mass spectrometer with electrospray ionization (ESI) source was used. In detail, metabolites were separated on an ACQUITY Premier BEH Amide Vanguard Fit column (100 mm × 2.1 mm, 1.7 µm, Waters) with constant column temperature of 35 °C. Separation of nucleotides was achieved by the following LC gradient scheme using mobile phase A (50/50; Acetonitrile/Water with 5 mM ammonium acetate + 0.05% (v/v) ammonium hydroxide, pH 10) and mobile phase B (90/10; Acetonitrile: Water with 5 mM ammonium acetate + 0.05% (v/v) ammonium hydroxide, pH 10):

| Time (min) | Flow (mL/min) | %A | %B |
|---|---|---|---|
| 0 | 0.400 | 5 | 95 |
| 0.5 | 0.400 | 5 | 95 |
| 0.51 | 0.350 | 5 | 95 |
| 5 | 0.350 | 90 | 10 |
| 5.1 | 0.300 | 90 | 10 |
| 5.2 | 0.300 | 100 | 0 |
| 10 | 0.300 | 100 | 0 |
| 10.3 | 0.400 | 5 | 95 |
| 15 | 0.400 | 5 | 95 |

Data acquisition was performed using Analyst 1.7.2 (AB SCIEX) and processed using the OS software suite 2.0.0 (AB SCIEX).

### Ammonium

To measure intracellular ammonium, a commercial kit based on reactivity with o-phthalaldehyde was used as per the manufacturer's instructions (Sigma-Aldrich #MAK310). Briefly, cell pellets were resuspended in ice-cold distilled water, incubated on ice for 10 min and centrifuged to remove cell debris. Supernatants were deproteinized using 10 kDa MWCO PES columns (Sartorius #VS0102) and then immediately used in the assay using 10 µl per sample. The resulting fluorescent signal was detected on a SPECTROstar Omega plate reader (BMG LABTECH).

## Immunocytochemistry and image analysis

One day before experiments, HEK293T cells were seeded on fibronectin-coated 12-mm glass coverslips in 24-well plates at a density of $1.8 \times 10^5$ cells per well. After treatment, cells were washed once in ice-cold DPBS and fixed with 4% formaldehyde (Thermo-Fisher #28908) for 15 min at room temperature. Cells were permeabilized with 0.1% saponin (Sigma-Aldrich #47036) for 10 min, blocked with 0.25% BSA (AppliChem #A1391) for 1 h and then incubated overnight with primary antibodies diluted in blocking solution. Fluorophore-conjugated secondary antibodies were added for 1 h at room temperature, followed by counterstaining with 5 µg/ml DAPI (Applichem #A1001) for 5 min. Coverslips were mounted on glass slides with Vectashield (Vector Labs #H-1000) and imaged on a confocal microscope (Leica TCS SP8) using a ×63 objective. Eight random fields per coverslip for a total of two coverslips per condition were acquired. Colocalization analysis was performed using CellProfiler. Briefly, the mTOR and

LAMP2 signals were segmented using an Otsu-based adaptive thresholding algorithm and the percentage of LAMP2 area colocalizing with the mTOR-positive area was calculated.

## Antibodies

The following primary antibodies were used for immunoblots at 1:1000 dilution: phospho-S6K1 T389 (Cell Signaling Technology #9205), S6K1 (Cell Signaling Technology #2708), phospho-4EBP1 S65 (Cell Signaling Technology #9451), 4EBP1 (Cell Signaling Technology #9452), α-tubulin (Sigma-Aldrich #T9026, 1:5000), ASNS (Cell Signaling Technology #20843), phospho-ACC S79 (Cell Signaling Technology #11818), ACC (Cell Signaling Technology #3662), phospho-Raptor S792 (Cell Signaling Technology #2083), Raptor (Cell Signaling Technology #2280), phospho-GCN2 T899 (Boster #P01172), GCN2 (Cell Signaling Technology #3302), phospho-eIF2α S51 (Cell Signaling Technology #3398), eIF2α (Cell Signaling Technology #5324), ATF4 (Cell Signaling Technology #11815), DEPDC5 (Abcam #ab185565), HA tag (Cell Signaling Technology #2367), AMPKα1/α2 (Cell Signaling Technology #2532), Redd1/Ddit4 (Abcam #ab191871), Sestrin1 (Proteintech #21668-1-AP), Sestrin2 (Proteintech #10795-1-AP), FBXO22 (Proteintech #13606-1-AP), mTOR (Cell Signaling Technology #2983), GLUL (Cell Signaling Technology #80636). The following secondary antibodies were used for immunoblots at 1:10000 dilution: anti-rabbit HRP (Jackson ImmunoResearch #111-035-003, used 1:10000), anti-mouse HRP (Jackson ImmunoResearch #115-035-003, used 1:10000). The following primary antibodies were used for immunostainings: mTOR (Cell Signaling Technology #2983, 1:200), LAMP2 (Hybridoma Bank #H4B4, 1:50). The following secondary antibodies were used for immunostainings: anti-rabbit Alexa488 (Life Technologies #A11008, 1:500), anti-mouse TRITC (Jackson ImmunoResearch #715-025-151, 1:200).

## RNA extraction and qPCR

Cellular total RNA was extracted with Trizol (Life Technologies #15596018) as per the manufacturer's instructions. In total, 2 μg total RNA per sample were reverse-transcribed using an oligo-dT primer and the Maxima H Minus reverse transcriptase (Thermo-Fisher #EP0753). Expression of the genes of interest was determined by qPCR using a SYBRGreen-based master mix (primaQUANT CYBR 2x qPCR SYBRGreen Master Mix with LOW ROX, Streinbrenner, #SL-9913) and appropriate forward and reverse oligonucleotide primers annealing to exons separated by at least one intron, so as to minimize amplification of potential contaminating genomic DNA (Dataset EV2). Relative transcript abundance was quantified with the $2^{-\Delta\Delta Ct}$ method.

## Cell proliferation analysis

HEK293T cells were seeded in multiple 96-well plates at a density of $2 \times 10^3$ cells per well. Relative cell numbers were determined using the CellTiter-Glo Luminescent Cell Viability Assay (Promega #G7570) over 5 days as per the manufacturer's instructions. To account for counting and/or seeding errors, the chemiluminescent signal measured for each condition on each day was normalized to the chemiluminescent signal measured 3 h after seeding (day 0).

## Phosphoproteomics analysis

### Treatments and sample preparation

One day before the experiment, HEK293T cells were seeded in 10-cm dishes at a density of $10^7$ cells per dish. After the treatments, samples were prepared as previously described (Potel et al, 2018). Cells were washed once in ice-cold DPBS and scraped on ice in 500 μl lysis buffer (100 mM Tri-HCl pH 8.5, 7 M urea, 1% Triton X-100, 5 mM TCEP, 30 mM chloroacetamide, 10 U/ml DNAse I, 1 mM MgCl₂, 1 mM sodium orthovanadate, 1× Roche Complete EDTA-free protease inhibitor cocktail, 1× Roche PhosSTOP phosphatase inhibitors). The lysates were then sonicated, cleared by maximal speed centrifugation on a tabletop centrifuge for 10 min and treated with 1% benzonase (Millipore #70746) for 30 min. Proteins were precipitated by adding four volumes of methanol, one volume of chloroform and three volumes of water to each lysate, followed by centrifugation for 15 min at 3500 rpm. After removing the upper phase, the precipitated proteins were washed by adding three volumes of methanol, pelleted by centrifugation for 15 min at 3500 rpm, air-dried for 5 min and stored at −80 °C.

Precipitated proteins were resuspended in digestion buffer (100 mM Tris-HCl pH 8.5, 1% sodium deoxycholate, 5 mM Tris(2-carboxyethyl) phosphin-hydrochloride and 30 mM chloroacetamide). Trypsin was added to a 1:50 ratio (w/w), and protein digestion was performed overnight at room temperature. The next day, digestion was stopped by the addition of TFA to a final concentration of 1% in the sample. The sodium deoxycholate was precipitated for 15 min at room temperature, and samples were centrifuged for 10 min at $17,000 \times g$ at room temperature. The supernatant was desalted by using Oasis HLB 96-well plates 30 μM (Waters). Thereby, buffer A was composed of MS-grade water (Chemsolute) with 0.1% formic acid and buffer B 80% acetonitrile (Chemsolute) in MS-grade water with 0.1% formic acid. Eluted peptides were dried in a vacuum centrifuge.

Phosphopeptide enrichment was performed on a KingFisher™ Apex instrument (ThermoFisher) as described before (Leutert et al, 2019). In total, 500 μg of peptides were used as input material. For each sample, a small aliquot was used for full proteome analysis. To enable post-enrichment TMT labeling, the phosphopeptides were eluted with 0.2% dimethylamine (Sigma-Aldrich) in 80% acetonitrile. Peptides were labeled with TMT16plex (Thompson et al, 2019) Isobaric Label Reagent (ThermoFisher) according to the manufacturer's instructions. Briefly, 0.8 mg reagent was dissolved in 42 μl acetonitrile (100%), and 4 μl of stock was added and incubated for 1 h at room temperature followed by quenching with 5% hydroxylamine for 15 min at room temperature. Samples were combined and for further sample clean up an OASIS® HLB μElution Plate (Waters) was used. The TMT-labeled phosphoproteome and full proteome were fractionated by high-pH reversed-phase chromatography carried out on an Agilent 1200 Infinity high-performance liquid chromatography system equipped with a Gemini C18 column (3 μm, 110 Å, $100 \times 1.0$ mm, Phenomenex). In total, 48 fractions were collected along with the LC separation that were subsequently pooled into 12 fractions. Pooled fractions were dried under vacuum centrifugation and reconstituted in 10 μL 1% formic acid, 4% acetonitrile and then stored at −80 °C until LC-MS analysis.

### Data acquisition

An UltiMate 3000 RSLC nano LC system (Dionex) fitted with a trapping cartridge (μ-Precolumn C18 PepMap 100, 5 μm, 300 μm i.d. × 5 mm, 100 Å) and an analytical column (nanoEase™ M/Z HSS T3 column 75 μm × 250 mm C18, 1.8 μm, 100 Å, Waters) was coupled to a Orbitrap Fusion™ Lumos™ Tribrid™ Mass Spectrometer (Thermo-Fisher). Peptides were concentrated on the trapping column with a constant flow of 0.05% trifluoroacetic acid at 30 μL/min for 6 min. Subsequently, peptides were eluted via the analytical column using a binary solution system at a constant flow rate of 0.3 μL/min. Solvent A consists of 0.1% formic acid in water with 3% DMSO and solvent B of 0.1% formic acid in acetonitrile with 3% DMSO. For the phospho-proteome, the percentage of solvent B was increased as follows: from 2% to 4% in 4 min, to 8% in 2 min, to 25% in 64 min, to 40% in 12 min, to 80% in 4 min, followed by re-equilibration back to 2% B in 4 min. For the full proteome analysis, the steps were as follows: from 2 to 8% in 4 min, to 28% in 104 min, to 40% in 4 min, to 80% in 4 min, followed by re-equilibration back to 2% B in 4 min.

The peptides were introduced into the Fusion Lumos via a Pico-Tip Emitter 360 μm OD × 20 μm ID; 10 μm tip (CoAnn technologies) and an applied spray voltage of 2.4 kV. The capillary temperature was set at 275 °C. A full mass scan was acquired with mass range 375–1650 $m/z$ for the phosphoproteome (375–1500 $m/z$ for the full proteome), in profile mode in the Orbitrap with resolution of 120,000. The filling time was set at a maximum of 50 ms for the full proteome with a limitation of $4 \times 10^5$ ions. Data-dependent acquisition (DDA) was performed with the resolution of the Orbitrap set to 30,000, with a fill time of 110 ms for the phosphoproteome (94 ms for the full proteome) and a limitation of $1 \times 10^5$ ions. A normalized collision energy of 34 was applied. MS2 data was acquired in profile mode. Fixed first mass was set 110 $m/z$.

### MS database search and data analysis

Fragpipe v20 with MSFragger v3.8 (Kong et al, 2017) was used to process the acquired data, which was searched against the *Homo sapiens* Uniprot proteome database (UP000005640, ID9606, 20594 entries, release October 2022) with common contaminants and reversed sequences included. The following modifications were considered as fixed modification: Carbamidomethyl (C) and TMT16 (K). As variable modifications: Acetyl (Protein N-term), Oxidation (M) and TMT16 (N-term), for the phosphoproteome specifically phosphorylation on STY. For the MS1 and MS2 scans, a mass error tolerance of 20 ppm was set. Further parameters were: trypsin as protease with an allowance of maximum two missed cleavages; minimum peptide length of seven amino acids; at least two unique peptides were required for protein identification. The false discovery rate on peptide and protein level was set to 0.01.

The raw output files of FragPipe (psm.tsv for phospho data and protein.tsv files for input data, (Kong et al, 2017)) were processed using the R programming language (ISBN 3-900051-07-0). Only peptide spectral matches (PSMs) with a phosphorylation probability greater 0.75 and proteins with at least 2 unique peptides were considered for the analysis. Phosphorylated amino acids were marked with a * in the amino acid sequences behind the phosphorylated amino acid, labeled with a 1, 2, or 3 for the number of phosphorylation sites in the peptide and concatenated with the protein ID in order to create a unique ID for each phosphopeptide. Raw TMT reporter ion intensities were summed for all PSMs with the same phosphopeptide ID.

Transformed summed TMT reporter ion intensities were first cleaned for batch effects using the "removeBatchEffects" function of the limma package (Ritchie et al, 2015) and further normalized using the vsn package (variance stabilization normalization (Huber et al, 2002)). Missing values were imputed with "knn" method using the Msnbase package (Gatto and Lilley, 2012). Proteins were tested for differential expression using the limma package. The replicate information was added as a factor in the design matrix given as an argument to the "lmFit" function of limma. Also, imputed values were given a weight of 0.05 in the "lmFit" function.

Phosphopeptides compatible with GCN2-dependent phosphorylation were defined as those satisfying all the following requirements: (1) fold change >1.2 and $P$ value < 0.05 in the comparison "control -Q vs +Q", (2) fold change < 0.8 and $P$ value < 0.05 in the comparison "−Q GCN2 KO vs control", (3) $P$ value > 0.05 in the comparison "GCN2 KO -Q vs +Q".

### Statistics and reproducibility

Statistical analysis was performed using GraphPad Prism (10.0.0) and Microsoft Excel (16.16.27). Statistical significance was determined using two-tailed, unpaired $t$ test, one-way or two-way ANOVA followed by post hoc tests, as specified in the corresponding figure legends. Data were assumed to be homoschedastic and normally distributed, although this was not formally tested. Sample sizes were not predetermined using statistical methods, but selected based on standard practice in the field. All experiments were repeated at least twice with independent sets of biological samples. All quantification and statistical analyses were derived from at least three independent experiments. No data were excluded from the analyses. The experiments were not randomized. The investigators were not blinded to allocation during experiments and outcome assessment.

## Data availability

Raw mass spectrometry proteomics data have been deposited to the ProteomeXchange Consortium via the PRIDE (Perez-Riverol et al, 2022) partner repository (https://proteomecentral.proteomexchange.org/cgi/GetDataset?ID=PXD055740). Corresponding processed mass spectrometry proteomics data are available as Dataset EV1. All other data are available from the corresponding author upon request.

The source data of this paper are collected in the following database record: biostudies:S-SCDT-10_1038-S44318-025-00505-1.

## Peer review information

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

## Acknowledgements

We are grateful to Jennifer Schwarz, Frank Stein and all other staff of the Proteomics Core Facility at the EMBL in Heidelberg for the phosphoproteomics analysis. We would also like to thank all members of the Teleman lab for helpful suggestions and insightful discussions.

## Author contributions

**Gianluca Figlia**: Conceptualization; Data curation; Formal analysis; Supervision; Validation; Investigation; Visualization; Methodology; Writing—original draft; Project administration; Writing—review and editing. **Sandra Müller**: Investigation. **Fabiola Garcia-Cortizo**: Conceptualization; Investigation. **Marilena Neff**: Investigation. **Glynis Klinke**: Data curation; Formal analysis; Investigation; Methodology. **Gernot Poschet**: Conceptualization; Formal analysis; Supervision; Investigation; Methodology; Project administration. **Aurelio A Teleman**: Conceptualization; Supervision; Funding acquisition; Visualization; Writing—original draft; Project administration; Writing—review and editing.

Source data underlying figure panels in this paper may have individual authorship assigned. Where available, figure panel/source data authorship is listed in the following database record: biostudies:S-SCDT-10_1038-S44318-025-00505-1.

## Funding

## Disclosure and competing interests statement

The authors declare no competing interests.

# Expanded View Figures

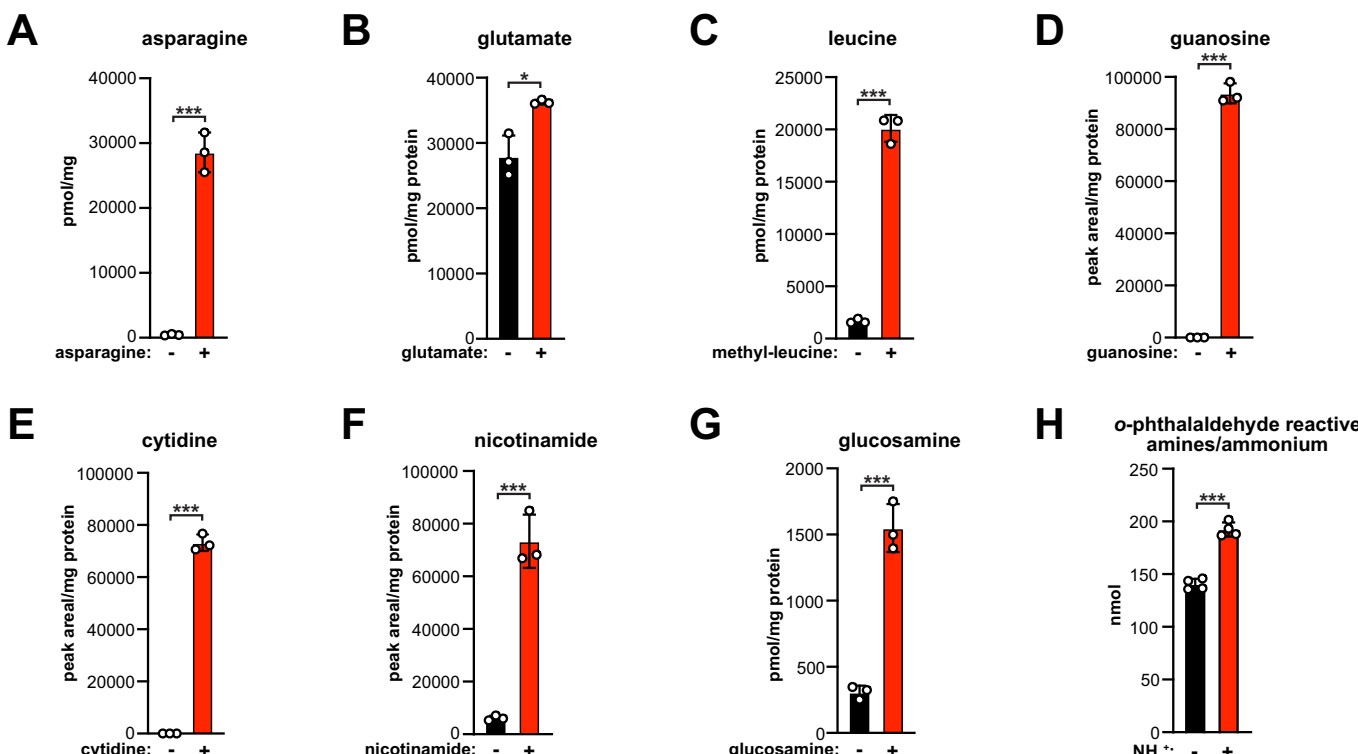

**Figure EV1. Measurement of intracellular downstream glutamine metabolites and metabolite precursors upon exogenous addition.**

(**A–H**) Exogenous addition of the metabolites in Fig. 1B results in a corresponding increase in their intracellular levels in control HEK293T cells. In the case of methyl-leucine, intracellular levels of non-esterified leucine were determined. Ammonium levels in (**H**) were measured as *o*-phthalaldehyde-reactivity. All metabolites were added for 1 h at 4 mM to cells cultured in regular DMEM. Data are expressed as either pmol amino acid per mg protein (**A–C**, **G**), peak area normalized per mg protein (**D–F**) or nmol per sample (**H**). Bar height: average, error bars: standard deviation, $n = 3$ (**A–G**) or 4 (**H**) biological replicates. Two-tailed unpaired *t* test. **P* < 0.05, ****P* < 0.001.

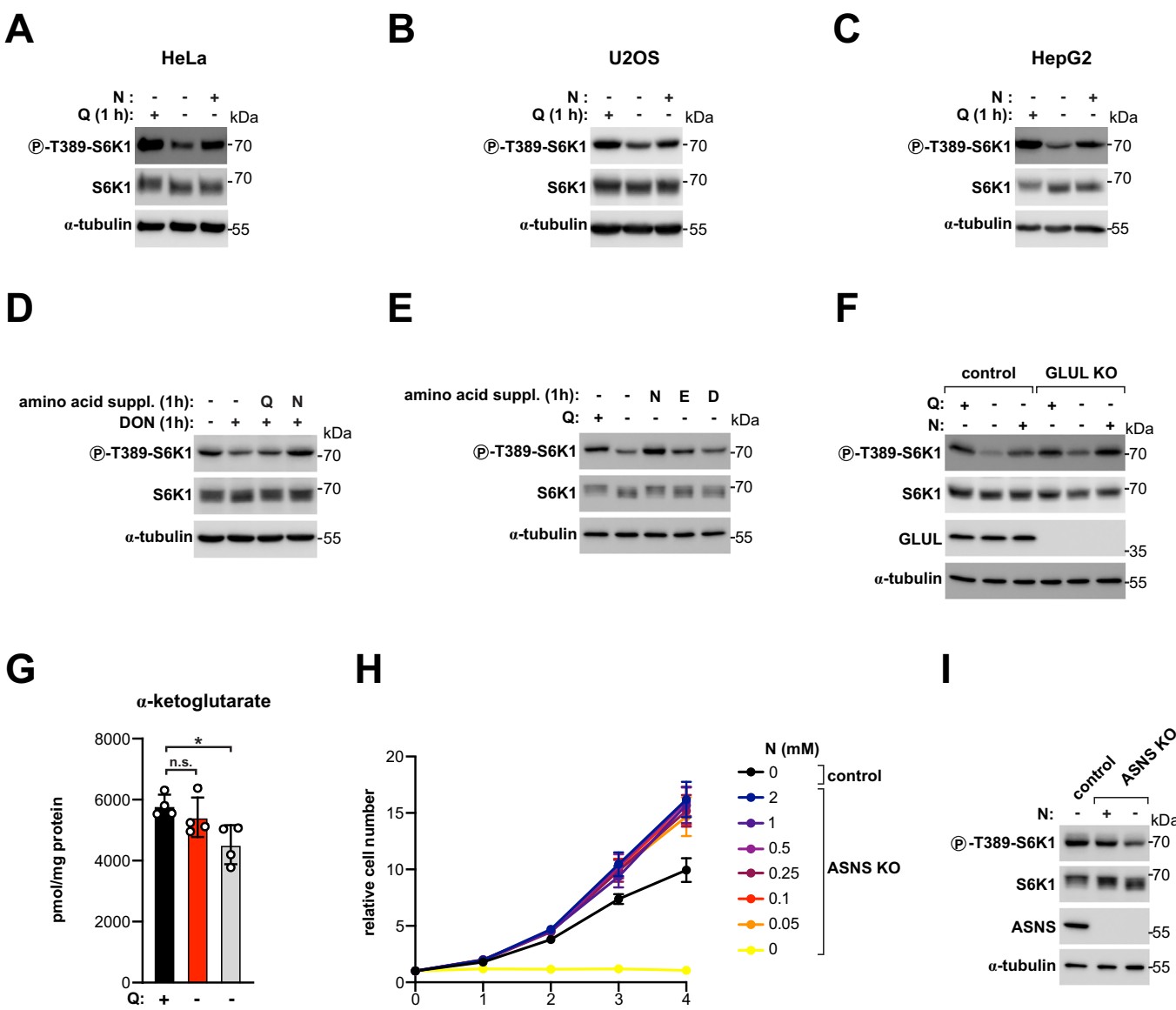

◀  **Figure EV2. Asparagine is both necessary and sufficient to maintain mTORC1 activity downstream of glutamine metabolism.**

(A–C) Asparagine rescues mTORC1 inhibition upon acute glutamine depletion in multiple cell lines. Immunoblot from HeLa (A), U2OS (B) and HepG2 (C) cells incubated for 1h in regular DMEM, glutamine-free DMEM or glutamine-free DMEM supplemented with 250 µM asparagine. Q: glutamine, N: asparagine. Representative of 5 biological replicates. The same samples were loaded on different gels and blotted in parallel with the indicated antibodies. Protein concentration was controlled by blotting α-tubulin on a separate gel. (D) Supplementation with asparagine rescues mTORC1 activity upon inhibition of glutamine metabolism with DON. Immunoblot from HEK293T cells treated for 1h with DON (100 µM) and incubated in regular DMEM or DMEM supplemented with additional glutamine (Q, 4 mM) or with asparagine (N, 4 mM). Representative of 2 biological replicates. The same samples were loaded on different gels and blotted in parallel with the indicated antibodies. Protein concentration was controlled by blotting α-tubulin on a separate gel. (E) Neither glutamate nor aspartate rescue mTORC1 inhibition upon glutamine removal. Immunoblot from HEK293T cells incubated for 1h in regular DMEM, glutamine-free DMEM or glutamine-free DMEM supplemented with 4 mM asparagine (N), glutamate (E) or aspartate (D). Representative of 3 biological replicates. The same samples were loaded on different gels and blotted in parallel with the indicated antibodies. Protein concentration was controlled by blotting α-tubulin on a separate gel. (F) Asparagine supplementation rescues mTORC1 inhibition in glutamine synthetase (GLUL)-knockout cells starved of glutamine. Immunoblot from control and GLUL KO HEK293T cells incubated for 1h in regular DMEM, glutamine-free DMEM or glutamine-free DMEM supplemented with 250 µM asparagine. Q: glutamine, N: asparagine. Representative of 3 biological replicates. The same samples were loaded on different gels and blotted in parallel with the indicated antibodies. Protein concentration was controlled by blotting α-tubulin on a separate gel. (G) Neither glutamine depletion nor asparagine supplementation affect α-ketoglutarate levels. α-ketoglutarate was measured in control HEK293T cells incubated for 1h in regular DMEM, glutamine-free DMEM or glutamine-free DMEM supplemented with 250 µM asparagine. Q: glutamine, N: asparagine. Data are expressed as pmol α-ketoglutarate per mg protein. Bar height: average, error bars: standard deviation, $n = 4$ biological replicates. One-way ANOVA and Tukey's post hoc test. *$p < 0.05$, n.s.: non significant. (H) ASNS KO cells require asparagine supplementation to proliferate. Relative cell numbers for control HEK293T or ASNS KO cells cultured over four days in regular DMEM (which does not contain asparagine) or in DMEM supplemented with the indicated asparagine (N) concentrations. Cell numbers on the day of seeding (day 0) are set to 1. Circle: average, error bars: standard deviation, $n = 5$ technical replicates. (I) Removal of asparagine inhibits mTORC1. Immunoblot from control and ASNS KO HEK293T cells incubated for 1h in DMEM supplemented with 250 µM asparagine (N) or in regular DMEM (which does not contain asparagine). Representative of 3 biological replicates. The same samples were loaded on different gels and blotted in parallel with the indicated antibodies. Protein concentration was controlled by blotting α-tubulin on a separate gel.

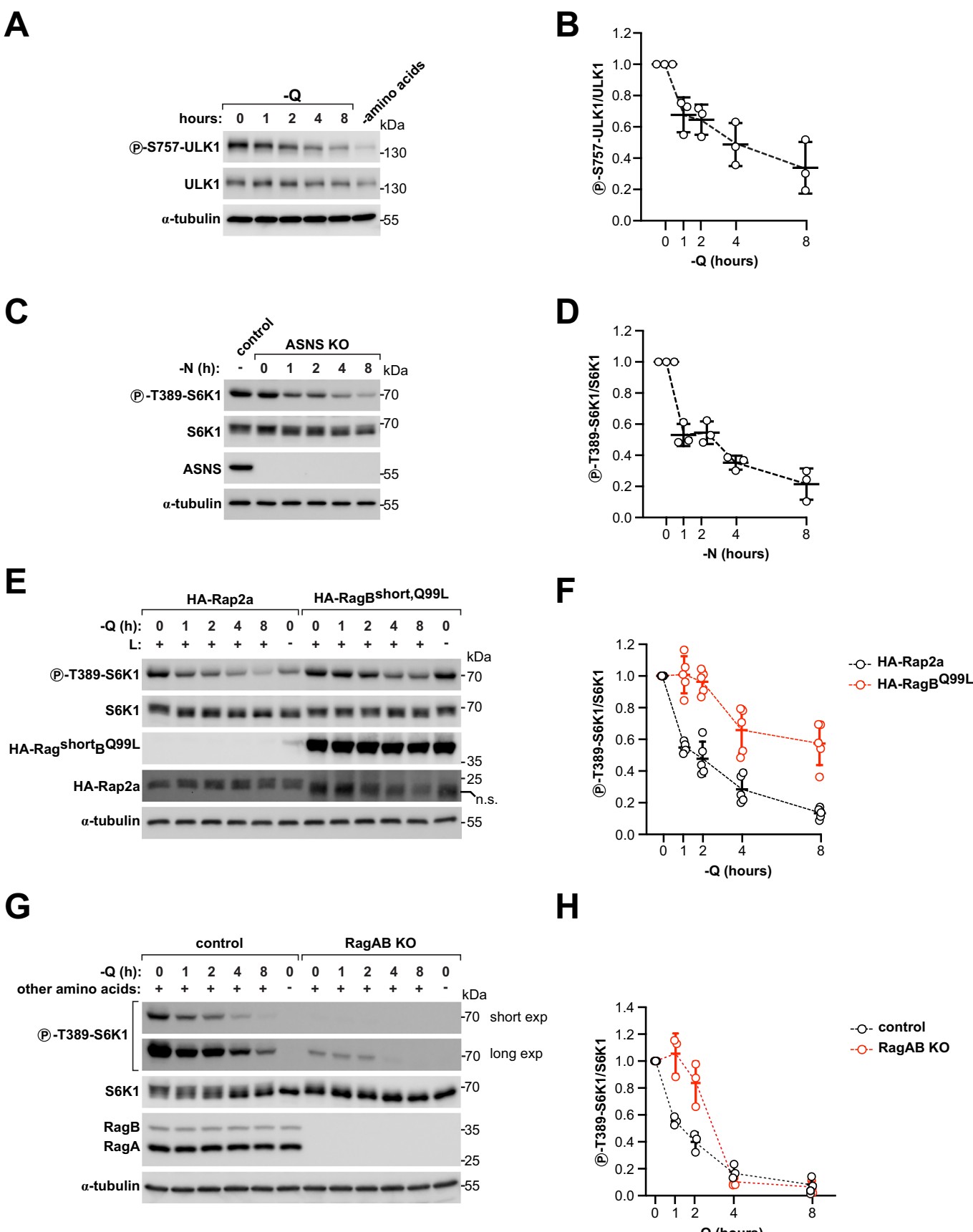

**Figure EV3. Glutamine is sensed through both Rag-dependent and Rag-independent mechanisms.**

(A, B) Glutamine removal causes dephosphorylation of the mTORC1 substrate ULK1. Immunoblot from control HEK293T cells incubated in regular or glutamine-free DMEM for the indicated time points. As a control, cells were incubated in amino acid-free DMEM for 8 h. Representative example (A) of 3 biological replicates quantified in (B). The same samples were loaded on different gels and blotted in parallel with the indicated antibodies. Protein concentration was controlled by blotting α-tubulin on a separate gel. Glutamine-rich condition is set to 1. Q: glutamine, line: average, error bars: standard deviation. (C, D) Asparagine removal inactivates mTORC1 progressively over time. Immunoblot from control and ASNS KO HEK293T cells incubated in DMEM supplemented with 250 μM asparagine (N) or in regular DMEM (which does not contain asparagine) for the indicated time points. Representative example (C) of 3 biological replicates quantified in (D). The same samples were loaded on different gels and blotted in parallel with the indicated antibodies. Protein concentration was controlled by blotting α-tubulin on a separate gel. Asparagine-rich condition is set to 1. N: asparagine, line: average, error bars: standard deviation. (E, F) Expression of constitutively active RagB[short] prevents acute (1–2 h) but not late mTORC1 inhibition after glutamine removal. Immunoblot from HEK293T stably transfected with a control protein (HA-Rap2a) or with HA-tagged constitutively active RagB[short] (Q99L) and incubated in regular or glutamine-free DMEM for the indicated time points. As a control, cells were incubated in leucine-free DMEM for 2 h. Q: glutamine, L: leucine. n.s.: non-specific band. Representative example (C) of 5 biological replicates quantified in (D). The same samples were loaded on different gels and blotted in parallel with the indicated antibodies. Protein concentration was controlled by blotting α-tubulin on a separate gel. Glutamine-rich condition for each genotype is set to 1. Q: glutamine, line: average, error bars: standard deviation. (G, H) Double deletion of RagA and RagB prevents acute (1–2 h) but not late mTORC1 inhibition after glutamine removal. Immunoblot from control and RagAB-knockout HEK293T cells incubated in regular or glutamine-free DMEM for the indicated time points. As a control, cells were incubated in amino acid-free DMEM for 8 h. Representative example (G) of 3 biological replicates quantified in (H). The same samples were loaded on different gels and blotted in parallel with the indicated antibodies. Protein concentration was controlled by blotting α-tubulin on a separate gel. Glutamine-rich condition for each genotype is set to 1. Q: glutamine, exp: exposure, line: average, error bars: standard deviation.

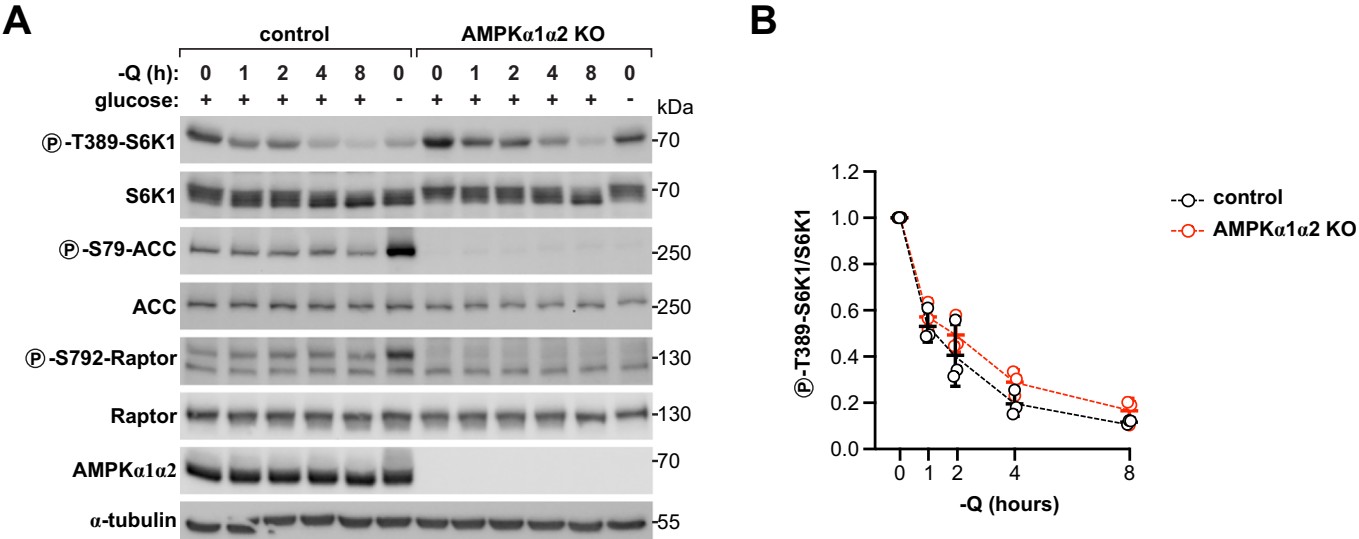

**Figure EV4.  Glutamine is not sensed through AMPK.**

(A, B) Inactivation of mTORC1 upon glutamine removal is independent of AMPK. Immunoblot from control or AMPKα1/α2-knockout HEK293T cells incubated in regular or glutamine-free DMEM for the indicated time points. As a control, cells were incubated in glucose-free DMEM for 2 h. Q: glutamine. Representative example (A) of 3 biological replicates quantified in (B). The same samples were loaded on different gels and blotted in parallel with the indicated antibodies. Protein concentration was controlled by blotting α-tubulin on a separate gel. Glutamine-rich condition for each genotype is set to 1. Line: average, error bars: standard deviation.

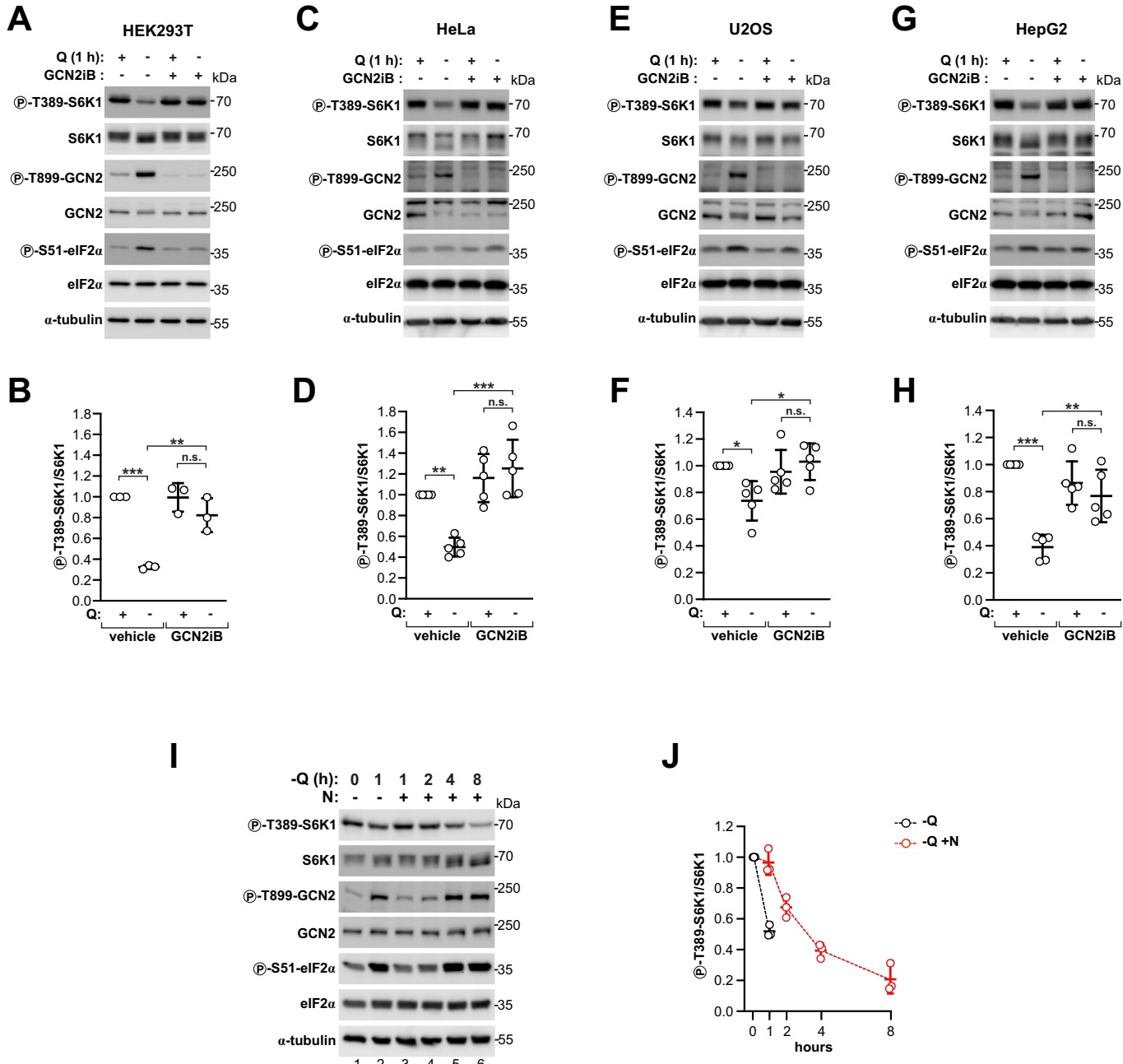

**Figure EV5. Glutamine is sensed through GCN2 in multiple cell lines.**

(A–H) mTORC1 inhibition upon acute glutamine depletion for 1 h requires GCN2. Immunoblot from HEK293T (**A**, **B**), HeLa (**C**, **D**), U2OS (**E**, **F**) and HepG2 (**G**, **H**) cells treated with vehicle (DMSO) or GCN2iB (1 µM) and incubated in regular or glutamine-free DMEM for 1 h. Representative example of 3 (**A**) or 5 (**C**, **E**, **G**) biological replicates quantified in (**B**, **D**, **F**, **H**). The same samples were loaded on different gels and blotted in parallel with the indicated antibodies. Protein concentration was controlled by blotting α-tubulin on a separate gel. Vehicle-treated unstarved cells are set to 1. Q: glutamine, line: average, error bars: standard deviation. One-way ANOVA and Tukey's post hoc test. *$P < 0.05$, **$P < 0.01$, ***$P < 0.001$, n.s.: non significant. (**I**, **J**) Asparagine supplementation rescues GCN2 activation and mTORC1 inhibition acutely, but not during prolonged glutamine depletion. Immunoblot from control HEK293T cells incubated in regular or glutamine-free DMEM for the indicated time points and supplemented with 250 µM asparagine. Representative example (**I**) of 3 biological replicates quantified in (**J**). The same samples were loaded on different gels and blotted in parallel with the indicated antibodies. Protein concentration was controlled by blotting α-tubulin on a separate gel. Glutamine-rich condition is set to 1 and relative phosphorylation of S6K1 upon glutamine starvation for 1 h or glutamine starvation and supplementation of asparagine from one to eight hours is shown. Q: glutamine, N: asparagine, line: average, error bars: standard deviation.

**A**

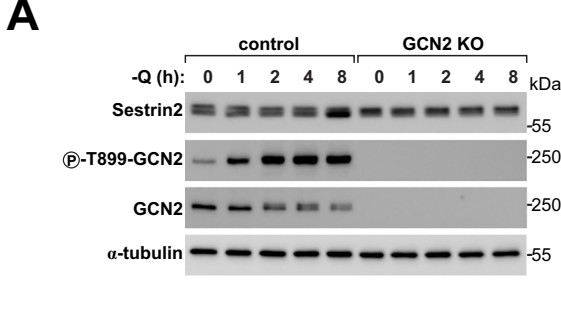

**B**

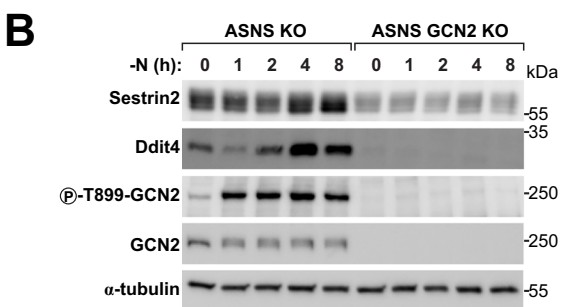

**C**

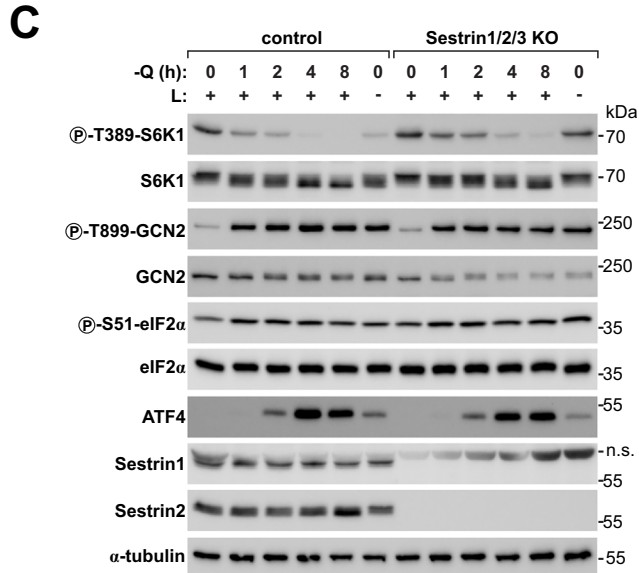

**D**

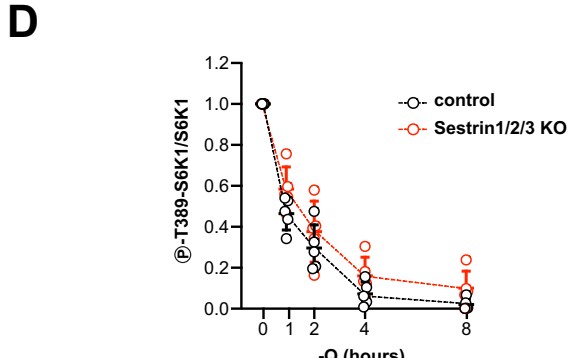

**E**

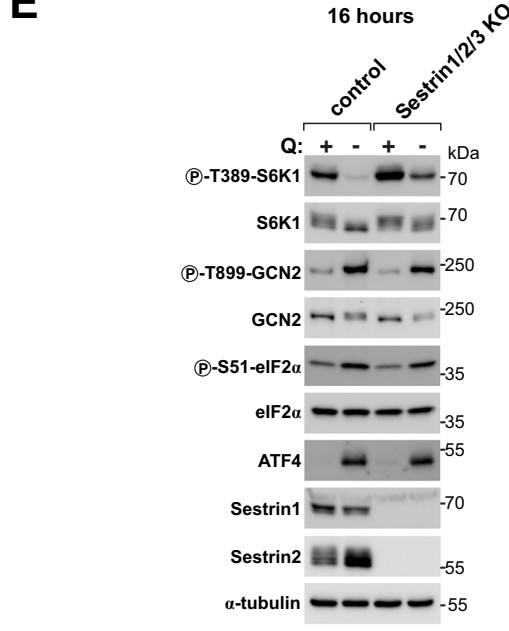

**F**

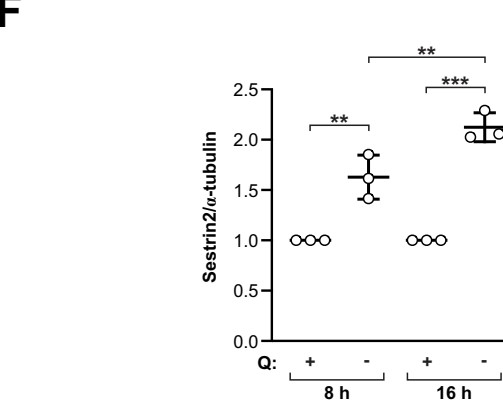

**G**

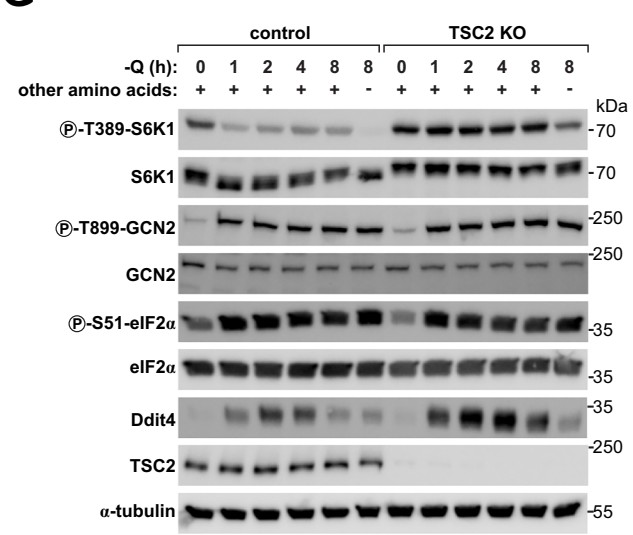

◀ **Figure EV6. Induction of Sestrin2 downstream of the ISR is involved in mTORC1 inhibition during overnight glutamine deprivation.**

(A) Sestrin2 protein levels increase at late time points after glutamine deprivation in a GCN2-dependent manner. Immunoblot from control or GCN2 KO HEK293T cells incubated in regular or glutamine-free DMEM for the indicated time points. Q: glutamine. Representative of 3 biological replicates. The same samples were loaded on different gels and blotted in parallel with the indicated antibodies. Protein concentration was controlled by blotting α-tubulin on a separate gel. (B) Sestrin2 and Ddit4 protein levels increase upon asparagine deprivation in a GCN2-dependent manner. Immunoblot from ASNS KO or ASNS GCN2 KO HEK293T cells incubated in DMEM supplemented with 250 μM asparagine (N) or in regular DMEM (which does not contain asparagine) for the indicated time points. Representative of 3 biological replicates. The same samples were loaded on different gels and blotted in parallel with the indicated antibodies. Protein concentration was controlled by blotting α-tubulin on a separate gel. (C, D) Sestrin1/2/3 do not significantly contribute to the inhibition of mTORC1 within the first 8 h of glutamine removal. Immunoblot from control and Sestrin1/2/3 KO HEK293T cells incubated in regular or glutamine-free DMEM for the indicated time points. As a control, cells were incubated in leucine-free DMEM for 2 h. Deletion of Sestrin3 was determined through sequencing of the corresponding genomic locus, given the poor quality of the Sestrin3 antibodies tested. Q: glutamine, L: leucine. n.s.: non-specific band. Representative experiment (B) of 5 biological replicates quantified in (C). The same samples were loaded on different gels and blotted in parallel with the indicated antibodies. Protein concentration was controlled by blotting α-tubulin on a separate gel. Glutamine-rich condition is set to 1 for all genotypes. Line: average, error bars: standard deviation. (E, F) Sestrin1/2/3 contribute towards maintaining mTORC1 repression upon prolonged removal of glutamine. Immunoblot from control HEK293T and Sestrin1/2/3 KO cells incubated in regular or glutamine-free DMEM overnight (16 h) (D) and quantification of Sestrin2 induction at 16 h and at 8 h in the experiment in (A). The same samples were loaded on different gels and blotted in parallel with the indicated antibodies. Protein concentration was controlled by blotting α-tubulin on a separate gel. Q: glutamine. Representative of 3 biological replicates. Glutamine-rich condition for each time point is set to 1. Circle: average, error bars: standard deviation. One-way ANOVA and Tukey's post hoc test. **$P < 0.01$, ***$P < 0.001$. (G) mTORC1 inhibition upon glutamine depletion requires the TSC complex. Immunoblot from control or TSC2 KO MEF cells incubated in regular or glutamine-free DMEM for the indicated time points. As a control, cells were treated with amino acid-free DMEM for 8 h. Q: glutamine. Representative example of 3 biological replicates. The same samples were loaded on different gels and blotted in parallel with the indicated antibodies. Protein concentration was controlled by blotting α-tubulin on a separate gel.

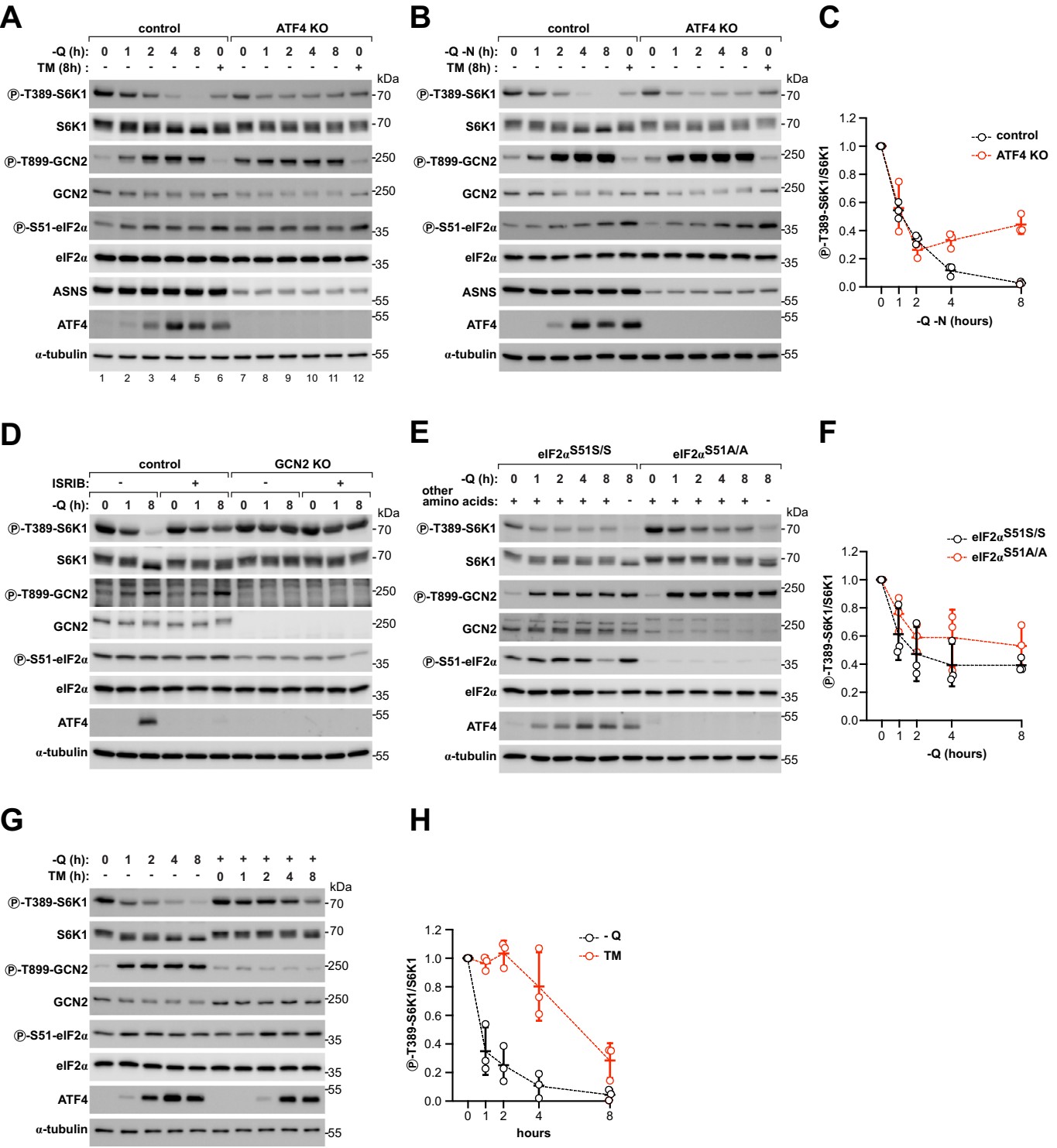

**Figure EV7. mTORC1 inhibition upon acute glutamine depletion does not require the ISR.**

(A) Late but not acute mTORC1 inhibition upon glutamine deprivation requires ATF4. Immunoblot from control or ATF4 KO HEK293T cells incubated in regular (without asparagine) or glutamine-free DMEM for the indicated time points. As a control, cells were treated with tunicamycin (TM, 1 μg/ml) for 8 h. Q: glutamine. Representative of 3 biological replicates. The same samples were loaded on different gels and blotted in parallel with the indicated antibodies. Protein concentration was controlled by blotting α-tubulin on a separate gel. (B, C) Late but not acute mTORC1 inhibition upon combined glutamine and asparagine deprivation requires ATF4. Immunoblot from control or ATF4 KO HEK293T cells incubated in regular DMEM supplemented with 250 μM asparagine (N) or in glutamine-free DMEM (which does not contain asparagine) for the indicated time points. As a control, cells were treated with tunicamycin (TM, 1 μg/ml) for 8 h. Q: glutamine. Representative example (B) of 3 biological replicates quantified in (C). The same samples were loaded on different gels and blotted in parallel with the indicated antibodies. Protein concentration was controlled by blotting α-tubulin on a separate gel. Glutamine- and asparagine-rich condition for each genotype is set to 1. Line: average, error bars: standard deviation. (D) mTORC1 activity upon glutamine depletion in control or GCN2 KO cells treated with ISRIB. Immunoblot from control or GCN2 KO HEK293T cells treated with vehicle (DMSO) or 200 nM ISRIB and incubated in regular or glutamine-free DMEM for the indicated time points. Q: glutamine. Representative example of 3 biological replicates. The same samples were loaded on different gels and blotted in parallel with the indicated antibodies. Protein concentration was controlled by blotting α-tubulin on a separate gel. (E, F) Acute mTORC1 inhibition upon glutamine removal does not require eIF2α phosphorylation. Immunoblot from control (eIF2α$^{S51S/S}$) or knock-in MEF cells harboring a mutation that abolishes eIF2a phosphorylation (eIF2α$^{S51A/A}$). Cells were incubated in regular or glutamine-free DMEM for the indicated time points. As a control, cells were treated with amino acid-free DMEM for 8 h. Q: glutamine. Representative example (E) of 3 biological replicates quantified in (F). The same samples were loaded on different gels and blotted in parallel with the indicated antibodies. Protein concentration was controlled by blotting α-tubulin on a separate gel. Glutamine-rich condition for each genotype is set to 1. Line: average, error bars: standard deviation. (G, H) Induction of the ISR by tunicamycin does not inhibit mTORC1 acutely. Immunoblot from control HEK293T cells incubated in regular or glutamine-free DMEM for the indicated time points, or treated with vehicle (DMSO) or tunicamycin (TM, 1 μl/ml) for the indicated time points. Q: glutamine. Representative example (G) of 3 biological replicates quantified in (H). The same samples were loaded on different gels and blotted in parallel with the indicated antibodies. Protein concentration was controlled by blotting α-tubulin on a separate gel. 0 h treatment conditions are set to 1. Line: average, error bars: standard deviation.

## A

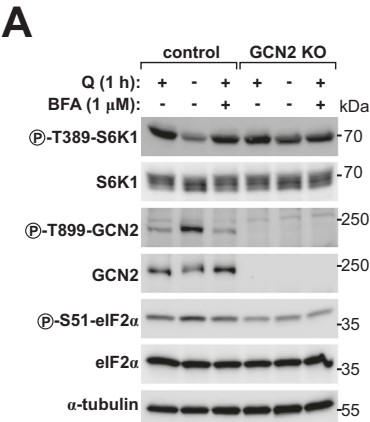

## B

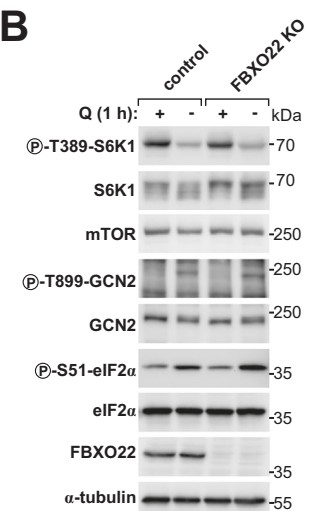

## C

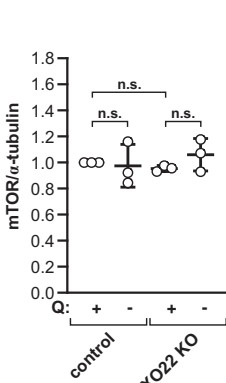

## D

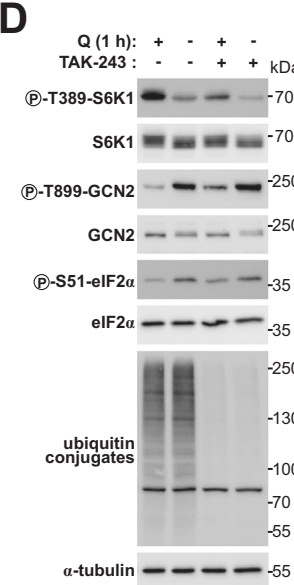

**Figure EV8. Neither ARF1 nor FBXO22 are involved in mTORC1 inhibition upon acute glutamine starvation.**

(A) ARF1 inhibition does not affect mTORC1 activity. Immunoblot from control or GCN2 KO HEK293T cells treated with vehicle (DMSO) or 1 μM Brefeldin A (BFA) for 1 h or starved of glutamine in glutamine-free DMEM for 1 h. Q: glutamine. Representative of 3 biological replicates. The same samples were loaded on different gels and blotted in parallel with the indicated antibodies. Protein concentration was controlled by blotting α-tubulin on a separate gel. (B, C) Inhibition of mTORC1 in response to acute glutamine removal for 1 h does not require FBXO22. Immunoblot from control or FBXO22 KO HEK293T cells incubated in regular or glutamine-free DMEM for 1 h. Q: glutamine. Representative example (B) of 3 biological replicates quantified in (C). The same samples were loaded on different gels and blotted in parallel with the indicated antibodies. Protein concentration was controlled by blotting α-tubulin on a separate gel. mTOR levels in control unstarved cells are set to 1. Q: glutamine, line: average, error bars: standard deviation. One-way ANOVA and Tukey's post hoc test. n.s.: non significant. (D) Inhibition of mTORC1 in response to glutamine removal for 1 h does not require any protein ubiquitination. Immunoblot from control HEK293T cells treated with vehicle (DMSO) or the ubiquitin activating enzyme (UAE) inhibitor TAK-243 (1 μM) and incubated in regular or glutamine-free DMEM for 1 h. Q: glutamine. Representative of 3 biological replicates. The same samples were loaded on different gels and blotted in parallel with the indicated antibodies. Protein concentration was controlled by blotting α-tubulin on a separate gel.

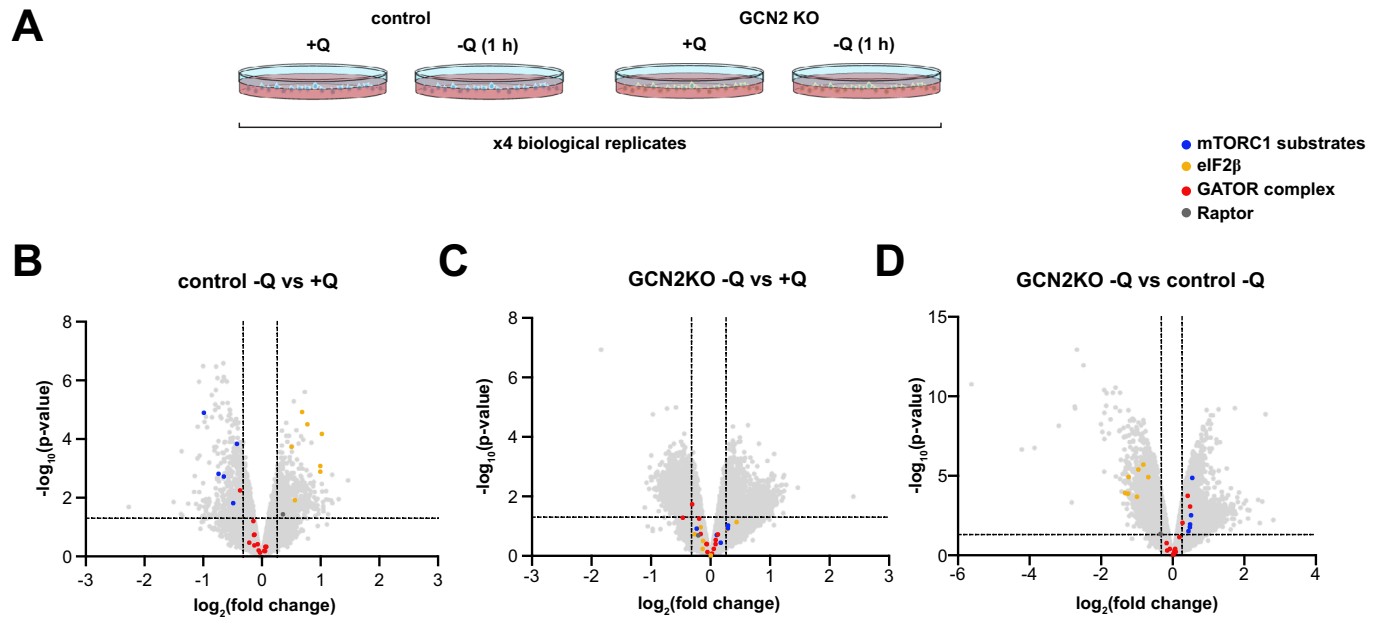

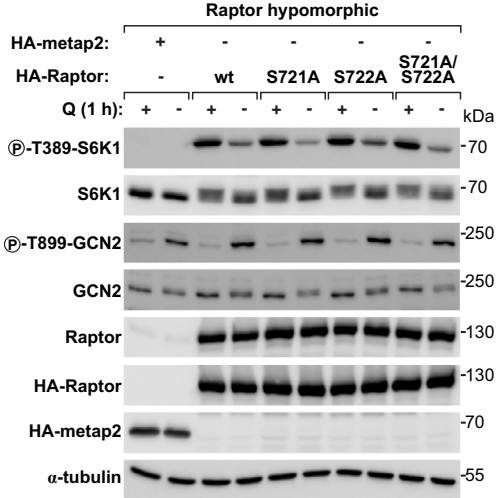

**Figure EV9. The GATOR-Rag machinery is not a direct substrate of GCN2.**

(A–D) Phosphoproteomics analysis of control HEK293T and GCN2 KO cells incubated in regular and glutamine-free DMEM for 1 h, as shown in the scheme in (A). (B–D) Volcano plots comparing the abundance of phosphorylated peptides between the indicated conditions. eIF2β: phospho-eIF2β (S105, T111, S105/T111). mTORC1 substrates: phospho-4EBP1 (S65/T68, S65/T70, T68/T70), phospho-4EBP2 (S65/T70), phospho-S6 (S235/S236). GATOR complex: phospho-DEPDC5 (S503, S1530, S445), phospho-Mios (S766), phospho-Wdr24 (S594/S598), phospho-Wdr59 (S564, S603), phospho-Szt2 (S719, S1642, S1644/1645, S1651, S1656/S1657). Raptor: phospho-Raptor (S722). Dashed lines correspond to $p = 0.05$ ($x$ axis) and ±1.2 fold change ($y$ axis), $n = 4$ biological replicates. Q: glutamine. (E) Inhibition of mTORC1 upon 1 h of glutamine removal does not require phosphorylation of Raptor at Ser721 or Ser722. Immunoblot from control HEK293T cells or Raptor-hypomorphic HEK293T cells that are almost completely lacking Raptor expression and thereby have very low mTORC1 activity. Cells were stably transfected with a HA-tagged control protein (HA-metap2), HA-tagged wild-type Raptor or alanine point mutants abolishing the phosphorylations at S721 and S722, either individually or combined. Cells were incubated in regular or glutamine-free DMEM for 1 h. Q: glutamine. Representative of 3 biological replicates. The same samples were loaded on different gels and blotted in parallel with the indicated antibodies. Protein concentration was controlled by blotting α-tubulin on a separate gel.

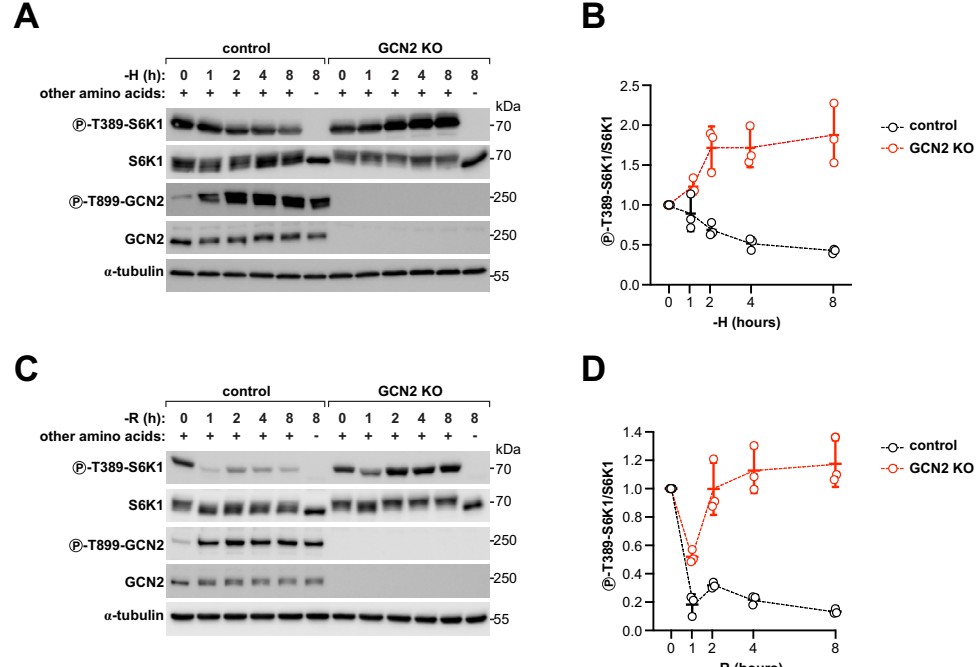

**Figure EV10. Response of mTORC1 to arginine or histidine deprivation.**

(A, B) Histidine deprivation causes progressive mTORC1 inhibition in an entirely GCN2-dependent manner. Immunoblot from control or GCN2 KO HEK293T cells incubated in regular or histidine-free DMEM for the indicated time points. As a control, cells were incubated in amino acid-free DMEM for 8 h. H: arginine. Representative example (A) of 3 biological replicates quantified in (B). The same samples were loaded on different gels and blotted in parallel with the indicated antibodies. Protein concentration was controlled by blotting α-tubulin on a separate gel. Histidine-replete condition for each genotype is set to 1. Line: average, error bars: standard deviation. (C, D) GCN2 is required for complete and persistent mTORC1 inhibition upon arginine deprivation. Immunoblot from control or GCN2 KO HEK293T cells incubated in regular or arginine-free DMEM for the indicated time points. As a control, cells were incubated in amino acid-free DMEM for 8 h. R: arginine. Representative example (C) of 3 biological replicates quantified in (D). The same samples were loaded on different gels and blotted in parallel with the indicated antibodies. Protein concentration was controlled by blotting α-tubulin on a separate gel. Arginine-replete condition for each genotype is set to 1. Line: average, error bars: standard deviation.

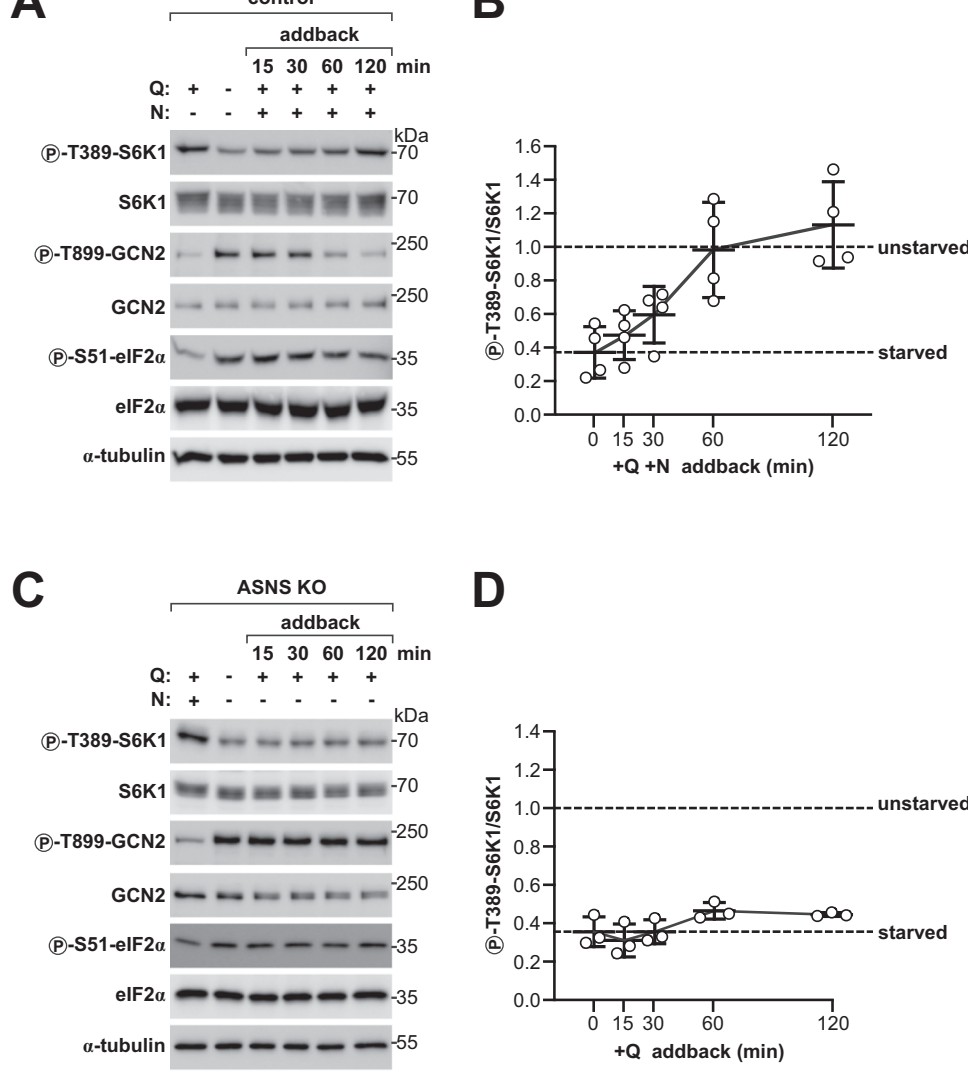

**Figure EV11. Metabolism to asparagine is required for mTORC1 reactivation upon addback of glutamine.**

(A, B) mTORC1 is reactivated slowly upon glutamine addback also when asparagine is additionally supplemented. Immunoblot from control HEK293T cells treated with glutamine removal and re-addition of glutamine together with 25 μM asparagine for the indicated time points. Q: glutamine, N: asparagine. Representative examples (A) of 3 biological replicates quantified in (B). The same samples were loaded on different gels and blotted in parallel with the indicated antibodies. Protein concentration was controlled by blotting α-tubulin on a separate gel. Unstarved cells are set to 1. Line: average, error bars: standard deviation. (C, D) mTORC1 reactivation upon glutamine addback requires its metabolism to asparagine. ASNS KO HEK293T cells were incubated in regular DMEM supplemented with 25 μM asparagine for 1 h, or starved of glutamine- and asparagine for 1 h and then re-stimulated with glutamine for the indicated time points. Q: glutamine, N: asparagine. Representative example (C) of 3 biological replicates quantified in (D). The same samples were loaded on different gels and blotted in parallel with the indicated antibodies. Protein concentration was controlled by blotting α-tubulin on a separate gel. Unstarved cells are set to 1. Line: average, error bars: standard deviation.

