## [Peer Review File · The EMBO Journal]

mTORC1 senses glutamine and other amino acids through GCN2

Gianluca Figlia, Sandra Müller, Fabiola Garcia-Cortizo, Marilena Neff, Glynis Klinke, Gernot Poschet and Aurelio A. Teleman

Corresponding authors: Aurelio Teleman (a.teleman@dkfz.de) , Gianluca Figlia (g.figlia@dkfz-heidelberg.de)

Review Timeline:

Submission Date:	29th Apr 25
Editorial Decision:	19th May 25
Additional Correspondence:	22nd May 25
Revision Received:	30th May 25
Editorial Decision:	26th Jun 25
Revision Received:	27th Jun 25
Accepted:	30th Jun 25

Editor: Daniel Klimmeck

Transaction Report:

Please note that the manuscript was previously reviewed at another journal. As EMBO Press has a transfer agreement (including the identities of the referees) with that journal, revision was invited based on the reports from that previous external submission.

Reviewer #1 (Remarks to the Author):

All my comments and concerns have been adequately addressed since the previous review and I would recommend this manuscript for publication.

Reviewer #2 (Remarks to the Author):

1) Fig. 1b. Because Gln starvation for 1 hour doesn't significantly decrease mTORC1 activity, it's hard to really see drastic changes here. It also appears that Glu, nicotinamide, and NH₄⁺ rescues (at least to some extent) mTORC1 activity in Gln deprived conditions. In these conditions Asn is not in the media. Could just the addition of Asn activate mTORC1 in Gln starvation conditions? Asn has previously been shown by many groups to regulate mTORC1. It could be helpful to look at other mTORC1 substrates in Fig. 1b-c to see a more dramatic effect.

2) It's not clear how the authors can conclude that both Rag-dependent and Rag-independent mechanisms are involved in glutamine sensing from Fig. 2c, Sup. Fig. 3e-h.

3) There are other pathways that regulate mTORC1 besides GCN2 and AMPK. It's not clear why these were the only two pathways looked at.

4) Fig. 2a. doesn't seem surprising since amino acid starvation activates GCN2. Fig. 2e. Does GCN2 KO cells block the decrease in mTORC1 activity when starving cells of other single amino acids (Leu, Arg, Met, Asn)?

5) Sup. Fig. 5e. There doesn't appear to be a change in mTORC1 activity after Gln starvation when looking at the Western.

6) Sup. Fig. 6e. The authors state in the text (page 11): "Sestrin-triple knockout cells did show resistance to mTORC1 inhibition after overnight glutamine removal." From the figure it doesn't appear to show full resistance to mTORC1 inhibition.

7) Fig. 3d. The Western doesn't appear to match the quantifications looking at mTORC1 activity.

8) Reviewer 2 - Point 10: The most crucial question is how does asparagine specifically regulate (molecular mechanism) GCN2, compared to the removal of other amino acids?

Reviewer Response - As we show in Figure 5, our results indicate that the GCN2-dependent mechanism we discover here is not specific for asparagine (or glutamine). Instead, it relays the depletion of virtually any amino acid to mTORC1. The particular dependence of glutamine sensing on asparagine in the first hour after glutamine removal reflects the fact that asparagine is already basally very low in cells (Figure 1d) and that it is the only amino acid that is entirely dependent on glutamine for its synthesis. Thus, when glutamine is removed, cells experience limitation of asparagine well before the levels of other amino acids become limiting (Figure 1d-e). However, removal of almost any amino acid leads to GCN2 activation (Figure 5). To avoid this potential misunderstanding, we have now changed the title of the manuscript as well as multiple parts of the text. This appears confusing because the title has glutamine in it, and the experiments throughout the manuscript focuses on glutamine and asparagine. It appears that amino acid starvation in general initiates these pathways through GCN2.

Reviewer #3 (Remarks to the Author):

In this resubmission, the authors provided additional evidence that the Asn sensing by mTORC1 occurs in several other cell lines. They propose that this is likely due to low levels of Asn, although it is not entirely clear if this is true for most cells or whether the expression levels of ASNS also play a role in most cases. This would be interesting to address given that asparaginase is effectively used in ALL but it is not clear whether this treatment may also be effective in particular cancer cells (where mTORC1 may be sensitive to Q or N limitation).

The studies propose that mTORC1 inhibition during Q starvation occurs via sequential GCN2-dependent mechanisms whereby acute Q limitation is mediated via a Rag-dependent and eIF2a/ISR-independent mechanism, whereas the prolonged response occurs via GCN2/ATF4/Ddit4/Sestrin2 mechanism. While they examined Sestrin2 as a Leu sensor, it is not clear if this is true for the other amino acid sensors in terms of addback responses. They can either try to address this further or limit their conclusion to Leu sensing.

Overall, the studies have some interesting findings and clarify some issues in amino acid sensing by mTORC1 via GCN2. The key mechanisms underlying Asn sensing via GCN2 or the acute GCN2 signals mediating Q starvation remain unclear.

Reviewer #2 (Remarks to the Author):

1) Fig. 1b. Because Gln starvation for 1 hour doesn't significantly decrease mTORC1 activity, it's hard to really see drastic changes here. It also appears that Glu, nicotinamide, and NH₄⁺ rescues (at least to some extent) mTORC1 activity in Gln deprived conditions. In these conditions Asn is not in the media. Could just the addition of Asn activate mTORC1 in Gln starvation conditions? Asn has previously been shown by many groups to regulate mTORC1. It could be helpful to look at other mTORC1 substrates in Fig. 1b-c to see a more dramatic effect.

The main point of the paper is that glutamine regulates mTORC1 activity via multiple mechanisms that act sequentially and additively (Fig. 4h). Each of these mechanisms contributes partially to mTORC1 inhibition upon Gln starvation, and some of them take several hours to kick in because they require transcriptional regulation (e.g. activation of ATF4 and expression of Ddit4). We dissect these sequential mechanisms by looking at various timepoints of Gln removal, and Figure 1 focuses entirely on the 1-hour response, which, by the nature of the biological system as described above, is partial. That said, we disagree that the response is not significant – as we show in the new panel in Figure 1c, mTORC1 activity drops to 60%, this drop is statistically significant and can be rescued only with asparagine, among the metabolites tested. Quantifications of the other immunoblots in this figure have now also been added (Figure 1e and Figure 1i).

2) It's not clear how the authors can conclude that both Rag-dependent and Rag-independent mechanisms are involved in glutamine sensing from Fig. 2c, Sup. Fig. 3e-h.

In the figures the Reviewer is referring to, we did three different manipulations that either pin Rag activity to be maximally high (DEPDC5 KO cells in Fig. 2c-d or cells with a constitutively active Rag complex in Supp. Fig. 3e-f) or to be completely off (Rag knockout cells in Suppl. Fig. 3g-h). In all three cases one can see that the cells still respond partially to Gln removal, meaning that part of the response is Rag-independent, but part of the response is gone, meaning that part of the response is Rag-dependent. This can easily be seen by comparing the red curves (mutant genotypes) to the control black curves in the quantifications and, we believe, unequivocally shows that both Rag-dependent and Rag-independent mechanisms are at play.

3) There are other pathways that regulate mTORC1 besides GCN2 and AMPK. It's not clear why these were the only two pathways looked at.

We do not think that the reviewer's comment is relevant. Although many other pathways converge on mTORC1, the Rag GTPases, GCN2 and AMPK are the only ones we are aware of which could be directly modulated by glutamine abundance. Moreover, we clearly show in Fig. 2e-f that in GCN2 knockout cells, all the response of mTORC1 to glutamine removal is gone. This means no other pathway is contributing.

4) Fig. 2a. doesn't seem surprising since amino acid starvation activates GCN2. Fig. 2e. Does GCN2 KO cells block the decrease in mTORC1 activity when starving cells of other single amino acids (Leu, Arg, Met, Asn)?

Yes - this is clearly shown in Figure 5a-d and explicitly discussed at length in the Results text. GCN2 KO cells show only partial inhibition of mTORC1 activity when starved of Leu, Arg, Met (and also Ser), and no inhibition at all when starved of all the other amino acids individually.

5) Sup. Fig. 5e. There doesn't appear to be a change in mTORC1 activity after Gln starvation when looking at the Western.

The pS6K band in the 2nd lane is less than in the 1st lane, and this is quantified for 5 biological replicates in Suppl. Fig. 5f showing a statistically significant drop. In lanes 3-4, in the presence of GCN2 inhibitor, pS6K does not drop, which is precisely the point of the figure.

6) Sup. Fig. 6e. The authors state in the text (page 11): "Sestrin-triple knockout cells did show resistance to mTORC1 inhibition after overnight glutamine removal." From the figure it doesn't appear to show full resistance to mTORC1 inhibition.

That is correct - the resistance is partial, in agreement with the main point of our paper which is that there are several mechanisms that contribute additively to mTORC1 regulation (Fig. 4h).

7) Fig. 3d. The Western doesn't appear to match the quantifications looking at mTORC1 activity.

The point of this figure is to show that loss of Ddit4 affects the response of mTORC1 to Gln removal during the later time-points of 2h, 4h and 8h. The quantification in Fig. 3e is from 6 biological replicates, one of which is shown in Fig. 3d. Nonetheless, also in the western blot in Fig. 3d one can clearly see that the pS6K band at 4h of Gln removal in the Ddit4 KO cells (lane 10) is higher than in the control cells (lane 4), as well as at 8 hours (lanes 11 versus 5).

8) Reviewer 2 - Point 10: The most crucial question is how does asparagine specifically regulate (molecular mechanism) GCN2, compared to the removal of other amino acids?

Reviewer Response - As we show in Figure 5, our results indicate that the GCN2-dependent mechanism we discover here is not specific for asparagine (or glutamine). Instead, it relays the depletion of virtually any amino acid to mTORC1. The particular dependence of glutamine sensing on asparagine in the first hour after glutamine removal reflects the fact that asparagine is already basally very low in cells (Figure 1d) and that it is the only amino acid that is entirely dependent on glutamine for its synthesis. Thus, when glutamine is removed, cells experience limitation of asparagine well before the levels of other amino acids become limiting (Figure 1d-e). However, removal of almost any amino acid leads to GCN2 activation (Figure 5). To avoid this potential misunderstanding, we have now changed the title of the manuscript as well as multiple parts of the text. This appears confusing because the title has glutamine in it, and the experiments throughout the manuscript focuses on glutamine and asparagine. It appears that amino acid starvation in general initiates these pathways through GCN2.

Yes - indeed - our findings are more broadly applicable to how mTORC1 senses all amino acids, and not only glutamine. We have changed the title to make it more clear that our results are broadly applicable.

Reviewer #3 (Remarks to the Author):

In this resubmission, the authors provided additional evidence that the Asn sensing by mTORC1 occurs in several other cell lines. They propose that this is likely due to low levels of Asn, although it is not entirely clear if this is true for most cells or whether the expression levels of ASNS also play a role in most cases. This would be interesting to address given that asparaginase is effectively used in ALL but it is not clear whether this treatment may also be effective in particular cancer cells (where mTORC1 may be sensitive to Q or N limitation).

As we described in our previous response to the reviewer, we see that Gln is sensed via Asn in all the cell lines that we tested, both cancer cell lines and non-cancer cell lines. Hence this seems to be a general conclusion that applies to all cells. In ALL cells, which have low endogenous Asn synthesis, we would expect this to be the case even more-so.

The studies propose that mTORC1 inhibition during Q starvation occurs via sequential GCN2-dependent mechanisms whereby acute Q limitation is mediated via a Rag-dependent and eIF2a/ISR-independent mechanism, whereas the prolonged response occurs via GCN2/ATF4/Ddit4/Sestrin2 mechanism. While they examined Sestrin2 as a Leu sensor, it is not clear if this is true for

the other amino acid sensors in terms of addback responses. They can either try to address this further or limit their conclusion to Leu sensing.

It is unfortunately not clear what conclusion the reviewer is referring to by 'it is not clear if this is true'. Maybe the reviewer refers to the results shown in Figure 6, where we show that mTORC1 inhibition by GCN2 is released more slowly than mTORC1 inhibition through the dedicated leucine sensors Sestrin1/2/3. If this is the case, we have now adjusted the final section of the introduction and added "dedicated sensors Sestrin1/2/3" in the title of the paragraph corresponding to Figure 6 to make more clear that we tested this aspect only for leucine and its sensors. If the reviewer is referring to the contribution of GCN2 to sensing other amino acids, this is clearly shown in Fig. 5a-d and Suppl. Fig. 5i-j.

Overall, the studies have some interesting findings and clarify some issues in amino acid sensing by mTORC1 via GCN2. The key mechanisms underlying Asn sensing via GCN2 or the acute GCN2 signals mediating Q starvation remain unclear.

The mechanisms how GCN2 senses amino acid deficiency, including Asn deficiency, have been broadly studied - this is known to occur via sensing of uncharged tRNAs and by sensing ribosome collisions. While the exact mechanism how GCN2 acutely inhibits mTORC1 still needs to be elucidated, we provide nonetheless an important mechanistic insight by showing that this is mediated by the Rag GTPases downstream of the GATOR complex.

Dear Aurelio,

Thank you again for the submission of your amended manuscript (EMBOJ-2025-121219-T) to The EMBO Journal. Please accept my apologies for the protraction due to delayed expert input. We have carefully assessed your manuscript and the point-by-point response provided to the referee concerns that were raised during peer-review at a different journal. In addition, and as mentioned before, we decided to involve the previous referee #1 as an arbitrating expert to evaluate the revised version of your work, with respect to technical robustness, conceptual advance and overall suitability for publication in The EMBO Journal.

As you will see from his/her comment enclosed below, the arbitrating advisor is broadly in favour of the work stating the interest and value of your results and s/he is supportive of publication at The EMBO Journal.

We are thus pleased to inform you that we can offer to swiftly move forward towards acceptance of this work at The EMBO Journal.

We now need you to take care of a number of minor issues related to formatting and data annotation, which I will share shortly in a separate message, together with additional changes and requests by our production team for Source Data provision.

Please submit a revised version of the manuscript using the link enclosed below, addressing the advisor's comments.

As you might have seen on our web page, every paper at the EMBO Journal now includes a 'Synopsis', displayed on the html and freely accessible to all readers. The synopsis includes a 'model' figure as well as 2-5 one-short-sentence bullet points that summarize the article. I would appreciate if you could provide this figure and the bullet points.

Thank you again for giving us the chance to consider your manuscript for The EMBO Journal, I look forward to hearing from you and receiving your final revised version of the manuscript.

Kind regards,

Daniel

EMBOJ-2025-121219-T

Referee #1, additional arbitrating comment:

'I am of the opinion that the results are novel, and the conclusions described within the manuscript are wholly justified by the data.

The mechanism underlying some of those results are unexplained, but the experiments conducted are to a high quality and the resulting observations are likely valid. All possible current understanding of how these pathways operate are unable to fully explain the carefully produced results seen within the manuscript.

In terms as to whether it is suitable for EMBO J. I would say the quality of the data and the novelty of the findings are of sufficient quality for publication in EMBO J. The mechanistic detail is lacking, but, as the manuscript does not present a model of how these observations may be explained (as there is no existing framework, given that experiments have been conducted which exclude facile models based on prior data), I feel this is fully justified. There may be another two papers' worth of systems/molecular cell biology/metabolic data necessary to fully elucidate the underlying cause of their results.

I, personally, would rather read a high-quality paper that ends with "this is new biology that requires further investigation" than a paper which fails to report novelty or include currently inexplicable results.'

Dear Aurelio,

Further to below message, please find the mentioned additional formatting requests on your study enclosed below.

Please let us know any time should there be questions related.

Best regards,
Daniel

Daniel Klimmeck, PhD
d.klimmeck@embojournal.org

>> Please provide the main manuscript text as .docx file.

>> Author Contributions: Remove the author contributions information from the manuscript text. Note that CRediT has replaced the traditional author contributions section as of now because it offers a systematic machine-readable author contributions format that allows for more effective research assessment. and use the free text boxes beneath each contributing author's name to add specific details on the author's contribution.

More information is available in our guide to authors.
<https://www.embopress.org/page/journal/14602075/authorguide>

>> Adjust the title of the 'Competing Interests' section to 'Disclosure and Competing Interests Statement'.

>> Provide a completed Author Checklist.

>> Section order should be corrected as follows: title page with complete author information, abstract, keywords, introduction, results, discussion, methods, data availability section, acknowledgements, disclosure and competing interests statement, references, main figure legends, tables, expanded figure legends.

Methods should be after Discussion; References should be placed before the figure legends

>> Figures in separate files: figures should be uploaded as individual, high resolution figure files. The supplementary figures should be renamed "Figure EV1" - EV11 and also uploaded as separate files. Their legends should be placed after the main figure legends and under the heading "Expanded View Figure Legends".

>> References: please adjust reference format to EMBO Journal format, 10 authors et al. . dois should be removed

>> Figure callouts: please recheck callouts for Fig 3H and Suppl Table 2 in the main text.

>> Please add a Reagents and Tools table to the Methods section, as a separate file using the existing template in the Guide For Authors, listing key reagents, experimental models, software and relevant equipment.

>> Dataset EV legends: the supplementary tables should be renamed "Dataset EV1" and "Dataset EV2", and both files should

have legends added in a separate tab/worksheet, with the titles and short descriptions.

>> Please provide source data for the study as to the separate request e-mail. Source data should be uploaded as one (zipped) file per figure.

>> Funding: please enter all funding information into the list of funders in our online system and detail them in addition in the Acknowledgments section of the manuscript.

>> Data availability section: Please provide a URL for the PRIDE dataset and ensure privacy is released and the data is public.

>> Methods: Please specify if reblotting of Western nitrocellulose membranes was applied.

>> Data integrity checks: Please recheck annotation and display for Western Blot loading controls and samples in the following Figure panels:

Figure 2A, 2G
Figure 3D, 3F
Figure 4A, 4C, 4E
Figure 5G, 5I
Figure 6A, 6E
Supp. Figure 2F
Supp. Figure 3E, 3G
Supp. Figure 4A
Supp. Figure 5I
Supp. Figure 6G
Supp. Figure 7B, 7D, 7G
Supp. Figure 8A
Supp. Figure 9E
Supp. Figure 10A, 10C
Supp. Figure 11A, 11C

Comparisons between and stacked display of samples on/from different gels/blots are discouraged. If a 'representative' loading control is shown for multiple gels/blots, the intra-gel controls should be shown in the source data files and the figure legends should describe the data displayed accurately.

Please also consider below respective text from author guidelines regarding this issue.

Quantitative comparisons between samples on different gels/blots are discouraged, even if the samples derive from one experiment, as confounding factors reduce comparability. If unavoidable, the figure legend must state that the samples derive from the same experiment and that gels/blots were processed in parallel. Loading controls must be run on the same gel.

If a 'representative' loading control is shown for multiple gels/blots, the intra-gel controls should be shown in the source data files and the figure legends should describe the data displayed accurately. Quantitative comparisons require that the experimental setup produces data within the linear signal range, unless a titration is added for standardisation. Where experimental limitations prevent reliable quantitative comparisons, we encourage qualitative interpretation instead.

>> Consider additional changes and comments from our production team as indicated below:

- DAS:

1. Please note that the specific URL for PXD055740 dataset is not provided in the data availability statement.

- Figure legends:

Please note that the exact p values are not provided in the legends of figures 1F, G, E, I; 4G

The authors addressed the editorial issues.

Dear Dr Teleman,

As discussed I hereby invite you for an additional minor revision of your manuscript files.

Please

** explicitly state in the Methods that '... The same samples were loaded on different gels and blotted in parallel with the indicated antibodies. Protein concentration was controlled by blotting a-tubulin on a separate gel. Sample volumes were controlled by subsequent Ponceau stain.'

** state in the Figure legends 'Protein concentration was controlled by blotting a-tubulin on a separate gel.'

** group/indicate blots deriving from the same gel in the SD with dashed lines.

Please let me us know if there are any questions related.

Best regards,

Daniel Klimmeck

Daniel Klimmeck, PhD
Senior Editor | The EMBO Journal
d.klimmeck@embojournal.org

The authors addressed the remaining editorial issues.

Dear Aurelio,

Thank you for submitting the revised version of your manuscript. I have now evaluated your amended manuscript and concluded that the remaining minor concerns have been sufficiently addressed.

I am thus pleased to inform you that your manuscript has been accepted for publication in the EMBO Journal.

On a different note, I would like to alert you that EMBO Press offers a format for a video-synopsis of work published with us, which essentially is a short, author-generated film explaining the core findings in hand drawings, and, as we believe, can be very useful to increase visibility of the work. Please see the following link for representative examples and their integration into the article web page:

<https://www.embopress.org/doi/full/10.15252/embojournal.2019103932>

Best regards,

Daniel

Daniel Klimmeck, PhD
Senior Editor
The EMBO Journal
EMBO
Postfach 1022-40
Meyerhofstrasse 1
D-69117 Heidelberg
contact@embojournal.org